# EVALUATING GFLOWNET FROM PARTIAL EPISODES FOR STABLE AND FLEXIBLE POLICY-BASED TRAINING

**Puhua Niu**
Texas A&M University
College Station, TX 77843, USA
`niupuhua.123@tamu.edu`

**Shili Wu**
Texas A&M University
College Station, TX 77843, USA
`shiliwu@tamu.edu`

**Xiaoning Qian**
Texas A&M University
College Station, TX 77843, USA
Brookhaven National Laboratory
Upton, NY 11973, USA
`xqian@tamu.edu`

## ABSTRACT

Generative Flow Networks (GFlowNets) were developed to learn policies for efficiently sampling combinatorial candidates by interpreting their generative processes as trajectories in directed acyclic graphs. The value-based training follows the Q-learning framework. The objective is to enforce the balance over partial episodes between the flows of the learned policy and the estimated flows of the desired policy, implicitly encouraging policy divergence minimization. The policy-based training operates under the actor-critic framework. It alternates between estimating the policy divergence and updating the policy, but reliable estimation of the divergence under directed acyclic graphs remains a major challenge. This work bridges the two perspectives by showing that flow balance also yields a principled policy evaluator that measures the divergence, and an evaluation balance objective over partial episodes is proposed for learning the evaluator. As demonstrated on both synthetic and real-world tasks, evaluation balance not only strengthens the reliability of policy-based training but also broadens its flexibility by seamlessly supporting parameterized backward policies and enabling the integration of offline data-collection techniques. Our code is available at github.com/niupuhua1234/Sub-EB.

## 1 INTRODUCTION

Generative Flow Networks (GFlowNets) are generative models on combinatorial spaces $\mathcal{X}$, such as graphs formed by organizing nodes and edges in a particular way, or strings composed of alphabets in a specific ordering. GFlowNets aim at sampling $x \in \mathcal{X}$ with probability $\propto R(x)$, where $R(x)$ is a non-negative score function. The task is challenging as $|\mathcal{X}|$ can be too large to compute the normalization constant $Z^* := \sum_{x \in \mathcal{X}} R(x)$ and the distribution modes can be highly isolated due to the combinatorial nature for efficient exploration. In GFlowNets (Bengio et al., 2021; 2023), generating or sampling $x \in \mathcal{X}$ is decomposed into incremental trajectories (episodes) that start from a null state, pass through intermediate states, and end at $x$ as the desired terminating state. These trajectories $\tau \in \mathcal{T}$ can be viewed as the paths along a Directed Acyclic Graph (DAG) with state $s \in \mathcal{S}$ and edge $(s \rightarrow s') \in \mathcal{E}$. Positive measures (unnormalized probability) of trajectories are viewed as the amount of *flows* along the DAG, and $R(x)$ is the total flow of trajectories ending at $x$, so that sampled trajectories will end at $x$ with the probability $\propto R(x)$.

The core of the GFlowNet training problem can be understood as minimizing the discrepancy of the forward trajectory distribution $P_F(\tau)$ induced by a forward policy $\pi_F(s'|s)$ towards the backward trajectory distribution $P_B(\tau) := P_B(\tau|x)R(x)/Z^*$ induced by a given backward policy $\pi_B(s|s')$ and $R(x)$ (Bengio et al., 2023; Malkin et al., 2022b). This is motivated by the fact that in real-world

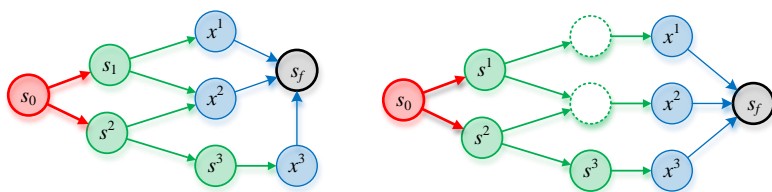

Figure 1: A graphical illustration of a DAG (left) and its graded version (right). Dotted circles represent dummy states, added during the conversion to a graded DAG.

applications, sequential generation must be done in a forward manner. Besides, the marginalization $P^*(x) = \sum_{\tau|x} P_B(\tau)$ trivially holds for all $x \in \mathcal{X}$, so any backward policy can be used to define a target backward trajectory distribution. Since evaluating $Z^*$ and thereby the normalized distribution $P_B(\tau)$ is considered intractable, directly optimizing some distributional divergence of $P_F(\tau)$ and $P_B(\tau)$ is not feasible. To circumvent this, value-based methods reformulate the problem of distributional matching as a flow-matching problem and leverage the balance conditions of flow values to derive training objectives. The basic condition $\mathbb{E}_{P_\mathcal{D}(s)}[\log \pi_F(s'|s)F(s)] = \mathbb{E}_{P_\mathcal{D}(s)}[\log \pi_B(s|s')F(s')]$ means that the edge flow value $F(s \to s') = \pi_F(s'|s)F(s)$ matches the target edge flow value in the reverse direction, where $P_\mathcal{D}(s)$ is the marginal state distribution induced by the data-collection policy $\pi_\mathcal{D}(s'|s)$. This basic condition leads to various training objectives that implicitly encourage the alignment between forward and backward distributions. These objectives range from edge-wise formulations to subtrajectory-wise variants, providing diverse estimations of target flows and quantifications of flow imbalances (Bengio et al., 2023; Madan et al., 2023; Malkin et al., 2022a). Alternatively, policy-based methods (Malkin et al., 2022b; Niu et al., 2024; Zimmermann et al., 2022) introduce an evaluation function $V(s)$ of the forward policy $\pi_F(\cdot|s)$, which approximates the Kullback-Leibler (KL) divergence between the distribution of forward subtrajectories (partial episodes) starting from state $s$ and that of backward subtrajectories ending at $s$. Then $V(s)$ is used to update the forward policy $\pi_F(\cdot|s)$. While policy updating based on $V(s)$ has been well-studied, how to reliably learn $V(s)$ remains an open problem.

In light of the success of value-based methods that leverage flow balance conditions and the analogy between the state flow function $F(s)$ as the total amount of trajectory flows passing state $s$, and the evaluation function $V(s)$ as a measurement of distributional divergence at state $s$, it is worthwhile to investigate the relationship between the two quantities. It is known that the optimal forward policy and state flow function can be uniquely determined by the flow balance condition $\mathbb{E}_{\pi_F}[\log F(s)\pi_F(s'|s)] = \mathbb{E}_{\pi_F}[\log F(s')\pi_B(s|s')]$. We find that, for any fixed $\pi_F$, the solution of the state flow to this condition coincides with the exact KL divergence starting from state $s$, that is, the true evaluation function of $\pi_F$. This paves the way to derive balance-based objectives to learn $V(s)$ reliably. Our contributions are as follows:

- We establish the connection between balance conditions with respect to (w.r.t.) the state flow function $F$ and the evaluation function $V$. Fixing $\pi_F$, the expected subtrajectory balancing conditions of $\log F$ directly lead to a sufficient condition for $V$, which we call the Subtrajectory Evaluation Balance (Sub-EB) condition.
- We introduce the Sub-EB objective for learning parameterized $V$. Based on the objective and its backward variant, we derive new policy-based methods that naturally accommodate parameterized $\pi_B$ and data-collection policy $\pi_\mathcal{D} \neq \pi_F$, enabling stable and flexible GFlowNet training.
- Experimental results on both simulated and real-world tasks, including hypergrid modeling, biological sequence design, molecular graph design, and Bayesian network structure learning, have demonstrated the effectiveness of Sub-EB for policy-based training.

## 2 PRELIMINARIES

We restrict DAGs of GFlowNets to be *graded* as any DAG can be equivalently converted to be graded by adding dummy non-terminating states (Malkin et al., 2022b). In a DAG $\mathcal{G} := (\mathcal{S}, \mathcal{E})$, element $s \in \mathcal{S}$ denotes a state, and element $(s \to s') \in \mathcal{E}(\subseteq \mathcal{S} \times \mathcal{S})$ denotes a directed edge. We

define the index set of time horizons as $[H] := \{0, \ldots, H\}$. Being *acyclic* means the state space $\mathcal{S}$ can be partitioned into disjoint subspaces: $\mathcal{S}_0, \ldots, \mathcal{S}_{H+1}$, where each element of $\mathcal{S}_h$ is denoted as $s_h$ for $h \in [H+1]$. Being *graded* further implies that actions are only allowed from $\mathcal{S}_h$ to $\mathcal{S}_{h+1}$. The initial and final states are unique, that is, $S_0 = \{s_0\}$ and $S_{H+1} = \{s_f\}$. The terminating state set, $S_H := \mathcal{X}$ is associated with a score function $R : \mathcal{X} \to \mathbb{R}^+$. Furthermore, the (complete) trajectory set is defined as $\mathcal{T} := \{\tau = (s_0 \to \cdots \to s_f) | \forall (s \to s') \in \tau : (s \to s') \in \mathcal{E}\}$. For $i < j \in [H+1]$, $\tau_{i:j}, \tau_{i:}$ and $\tau_{:j}$ denote subtrajectories (partial episodes) from $s_i$ to $s_j$, from $s_i$ to $s_f$, and from $s_0$ to $s_j$, respectively. Denoting the natural $\sigma$-algebra on $\mathcal{T}$ as $\sigma(\mathcal{T})$, $P(\Gamma) := \sum_{\tau \in \Gamma} P(\tau)$ for any event $\Gamma \in \sigma(\mathcal{T})$. An exemplary DAG and its *graded* version are shown in Fig. 1.

GFlowNet training aims to align the forward trajectory distribution $P_F(\tau) := \prod_{i=0}^{H} \pi_F(s_{h+1}|s_h)$ with the backward trajectory distribution $P_B(\tau) := \prod_{i=0}^{H} \pi_B(s_h|s_{h+1})$, where $\pi_F$ and $\pi_B$ are forward and backward *Markovian* policies (transition distributions), respectively, and are assumed to satisfy $P_F(\tau) > 0$ and $P_B(\tau) > 0$ for any $\tau \in \mathcal{T}$. Accordingly, $P_F(\tau_{i:j}|s_i) = \prod_{h=i}^{j-1} \pi_F(s_{h+1}|s_h)$, and $P_B(\tau_{i:j}|s_j) = \prod_{h=i}^{j-1} \pi_B(s_h|s_{h+1})$. Except that $\pi_B(x|s_f) := R(x)/Z^*$ for any $x \in \mathcal{X}$, $\pi_B(s|s')$ can be arbitrary for intermediate edges, as $P_B(x) = \sum_{\tau|x} P_B(\tau) = R(x)/Z^*$ always holds. Therefore, the optimal forward policy $\pi_F^*$, satisfying $P_F^*(\tau) = P_B(\tau)$ for all $\tau \in \mathcal{T}$, induces its marginal $P_{F^*}(x) = R(x)/Z^*$, thereby achieving the ultimate goal of GFlowNet. Moreover, under the *Markovian* assumption (Bengio et al., 2023), the optimal forward policy $\pi_F^*$ corresponding to $\pi_B$ is unique.

## 2.1 VALUE-BASED TRAINING

Traditional Variational Inference (VI) approaches that minimize the KL divergence $\mathcal{L}_{KL}(\theta) := D_{\mathrm{KL}}(P_F(\tau; \theta) \| P_B(\tau))$ is infeasible as $Z^*$ is considered intractable (Malkin et al., 2022b). To circumvent the intractable $Z^*$, value-based methods introduce an *unnormalized* and *Markovian* distribution $F(\tau) := P_F(\tau)Z$, called (trajectory) *flow* (Bengio et al., 2023), together with a total *flow* $Z$. Accordingly, for any event $\Gamma \in \sigma(\mathcal{T})$, its flow $F(\Gamma) := \sum_{\tau \in \Gamma} P(\tau)$. In particular, the edge flow and state flow are $F(s \to s') := \sum_{\tau \ni (s \to s')} F(\tau)$ and $F(s) := \sum_{s'} F(s \to s')$. Furthermore, $F(s_0) = F(s_f) = Z$, and $\pi_F(s'|s) = F(s \to s')/F(s)$. Analogously, the optimal flow is $F^*(\tau) := P_B(\tau)Z^* = P_B(\tau|x)R(x)$. Therefore, the optimality condition of GFlowNets is converted to $\forall \tau \in \mathcal{T} : F(\tau) = F^*(\tau)$. Thanks to the *Markovian* assumption, the condition can be relaxed and reformulated to the Sub-Trajectory Balance (Sub-TB) condition (Madan et al., 2023). It requires $\forall i < j \in [H+1]$:

$$\mathbb{E}_{P_{\mathcal{D}}(\tau_{i:j})} \left[ \log \left( F(s_i) P_F(\tau_{i:j}|s_i) \right) \right] = \mathbb{E}_{P_{\mathcal{D}}(\tau_{i:j})} \left[ \log \left( F(s_j) P_B(\tau_{i:j}|s_j) \right) \right], \quad (1)$$

where $F(s_f) P_B(x|s_f) := R(x)$ for $x \in \mathcal{X}$, and $P_{\mathcal{D}}(\tau_{i:j}) = \sum_{\tau \ni \tau_{i:j}} P_{\mathcal{D}}(\tau)$ is induced by the offline data-collection policy $\pi_{\mathcal{D}}$. It is assumed that $P_{\mathcal{D}}(\tau) > 0$ for any $\tau \in \mathcal{T}$ (Malkin et al., 2022b). Without the assumption, a stricter alternative is to enforce flow balance for each individual subtrajectory, as in Madan et al. (2023). The condition above can be interpreted as requiring the flow value of a subtrajectory to match the target flow value in the reverse direction, which represents a (approximated) desired flow value.

Leveraging the balance condition, value-based methods typically use the Sub-TB objective to optimize the parameterized policy $\pi_F(s'|s; \theta)$ and state flow $F(s; \theta)$ toward their optimal solutions $\pi_F^*(s'|s)$ and $F^*(s)$. The objective is defined as follows:

$$\mathcal{L}_F(\theta) := \mathbb{E}_{P_{\mathcal{D}}(\tau)} \left[ \sum_{\tau_{i:j}} w_{i,j} \left( \delta_F(\tau_{i:j}; \theta) \right)^2 \right], \quad \delta_F(\tau_{i:j}; \theta) := \log \frac{P_F(\tau_{i:j}|s_i; \theta) F(s_i; \theta)}{P_B(\tau_{i:j}|s_j; \theta) F(s_j; \theta)}, \quad (2)$$

where $w_{i,j}$ denotes the non-zero weight coefficient for subtrajectory $\tau_{i:j}$. For practical gradient-based optimization, $\mathcal{L}_F(\theta)$ are approximated by $\widehat{\mathcal{L}}_F(\theta) := \frac{1}{K} \sum_{\tau \in \mathcal{D}} [\sum_{\tau_{i:j} \in \tau} w_{i,j} \delta_F(\tau_{i:j}; \theta)]$, where $\mathcal{D} := \{\tau^k : \tau^k \sim P_{\mathcal{D}}(\tau)\}_{k=1}^{K}$ is the set of samples.

## 2.2 POLICY-BASED TRAINING

Policy-based training methods for GFlowNets (Niu et al., 2024) resemble the policy gradient algorithm in Reinforcement Learning (RL) and aim to minimize the KL divergence $\mathcal{L}_{KL}(\theta)$ as in traditional VI approaches. The methods follow the **actor-critic** framework (Sutton & Barto, 2018; Haarnoja et al., 2018).

**Critic** In each training iteration, the critic (true evaluation function) $V^\dagger$ of actor $\pi_F$ is first computed to capture the policy gaps in terms of KL divergences over subtrajectories, thereby facilitating the computation of the overall trajectory divergence. For any $h \in [H]$, $V^\dagger(s_h; \theta)$ is defined as:[1]

$$\mathbb{E}_{P_F(\tau_{h:}|s_h;\theta)}\left[\log \frac{P_B(\tau_{h:})Z^*}{P_F(\tau_{h:}|s_h;\theta)}\right] = \log F^*(s_h) - D_{\text{KL}}(P_F(\tau_{h:}|s_h;\theta)\|P_B(\tau_{h:}|s_h)). \tag{3}$$

The equality can be easily verified (Niu et al., 2024) as shown in (19) in the Appendix. It should be noted that the explicit computation of $Z^*$ is not required in the above definition as $\pi_B(x|s_f)Z^* = R(x)$. Analytical computation of $V^\dagger$ is usually infeasible as $|\mathcal{T}|$ can be enormous, so $V^\dagger$ is modeled by an evaluation function $V$ with learnable parameters. We will discuss the learning objective of $V$ in the next section.

**Actor** To minimize $\mathcal{L}_{KL}(\theta)$, it is noted that $\nabla_\theta V^\dagger(s_0; \theta) = -\nabla_\theta \mathcal{L}_{KL}(\theta)$ since $F^*(s_0)(= Z^*)$ is a constant. Consequently, $-V^\dagger(s_0; \theta)$ can be a surrogate to the KL divergence. Further applying the policy gradient theorems in RL (Agarwal et al., 2021), $\nabla_\theta V^\dagger(s_0; \theta)$ can be simplified and computed in terms of $V$ as follows:

$$\mathbb{E}_{P_F(\tau)}\left[\sum_{h=0}^{H} A^\gamma(s_h \to s_{h+1})\nabla_\theta \log \pi_F(s_{h+1}|s_h;\theta)\right]. \tag{4}$$

Here, $A^\gamma(s_h \to s_{h+1}) := \sum_{i=h}^{H} \gamma^{i-h}\delta_V(s_i \to s_{i+1})$, and $\delta_V$ is defined in (8) of Section 3.1. The hyperparameter $\gamma \in [0, 1]$ enables the **variance-bias** trade-off during stochastic gradient descent (Schulman et al., 2016; 2017), analogous to $w$ in the Sub-TB method.

## 3 SUBTRAJECTORY EVALUATION BALANCE

As discussed in the previous sections, the evaluation function $V$ plays a central role in policy-based methods. In this section, we first introduce the Sub-EB condition that characterizes the relationship between $\pi_F$ and $V$, which closely resembles the Sub-TB conditions between $\pi_F$ and $F$. We then present the Sub-EB objective for learning $V$. Further, we extend these results with corresponding conditions and objectives for $\pi_B$ and the backward evaluation function $W$.

### 3.1 THE BALANCE CONDITION AND OBJECTIVE FOR FORWARD POLICIES

Given a forward policy $\pi_F$ such that $P_F(\tau) > 0$ for all $\tau \in \mathcal{T}$, the Subtrajectory Evaluation Balance (Sub-EB) condition for the associated evaluation function $V$ can be written as $\forall i < j \in [H + 1]$:

$$\mathbb{E}_{P_F(\tau_{i:j})}\left[\log\left(P_F(\tau_{i:j}|s_i)\exp V(s_i)\right)\right] = \mathbb{E}_{P_F(\tau_{i:j})}\left[\log\left(P_B(\tau_{i:j}|s_j)\exp V(s_j)\right)\right], \tag{5}$$

where $P_B(x|s_f)\exp V(s_f) := R(x)$. Denoting the entropy of a distribution as $\mathcal{H}(\cdot)$, the condition can be reformulated as: $\mathbb{E}_{P_F(s_i,s_j)}[V(s_i) - V(s_j)] = \mathbb{E}_{P_F(\tau_{i:j})}\left[\log \frac{P_B(\tau_{i:j}|s_j)}{P_F(\tau_{i:j}|s_i)}\right] = \mathcal{H}(P_F(s_i)) - \mathcal{H}(P_F(s_j)) - \mathbb{E}_{P_F(s_j)}[D_{\text{KL}}(P_F(\tau_{i:j}|s_j)\|P_B(\tau_{i:j}|s_j))]$. Therefore, this condition intuitively requires that the difference of the learned divergences at $s_i$ and $s_j$ matches the true divergence over subtrajectories between the two states.

**Theorem 3.1.** *Suppose $V$ is an arbitrary evaluation function over $\mathcal{S}$, and $F^*$ is the optimal flow induced by a backward policy $\pi_B$. Given a forward policy $\pi_F$,*

$$\forall h \in [H] : V(s_h) = \log F^*(s_h) - D_{\text{KL}}(P_F(\tau_{h:}|s_h)\|P_B(\tau_{h:}|s_h)), \tag{6}$$

*if and only if $V$ satisfies the Sub-EB condition (5).*

The corresponding proof can be found in Appendix A.1.

**Theorem 3.2.** *Suppose $F$ is an arbitrary state flow function over $\mathcal{S}$, $\pi_F$ is an arbitrary forward policy, and $(F^*, \pi_F^*)$ is the optimal flow and forward policy induced by a backward policy $\pi_B$. Then,*

$$\forall h \in [H] : \log F(s_h) = \log F^*(s_h) - D_{\text{KL}}(P_F^*(\tau_{h:}|s_h)\|P_B(\tau_{h:}|s_h)) \tag{7}$$

*and $\pi_F = \pi_F^*$ if and only if $(F, \pi_F)$ satisfies the Sub-TB condition (1).*

---

[1]Niu et al. (2024) consider an additional total flow estimator $Z(\theta)$ to scale down the magnitude of $V^\dagger$. For notational compactness, we defer the discussion of integrating $Z$ into our method to Appendix A.4.

The corresponding proofs can be found in Appendix A.2. While the sufficiency and necessity of the Sub-TB condition have been studied in prior work (Bengio et al., 2023; Malkin et al., 2022a), Theorem 3.2 offers an alternative perspective that more clearly elucidates the connection between the flow function and the evaluation function. It should be noted that the minimum of the KL term above is zero, as both $P_F(\tau)$ and $P_B(\tau)$ are *Markovian*. Moreover, by Proposition 23 in Bengio et al. (2023), the trajectory flow $F(\tau) = P_F(\tau)Z$ that achieves the zero KL term is unique.

Leveraging the balancing condition, we define the Sub-EB objective for optimizing a parameterized evaluation function $V(\,\cdot\,;\phi)$ as:

$$\mathcal{L}_V(\phi) := \mathbb{E}_{P_F(\tau)}\Big[\sum_{\tau_{i:j}} w_{i,j}\,(\delta_V(\tau_{i:j};\phi))^2\Big],\quad \delta_V(\tau_{i:j};\phi) = \log\frac{P_B(\tau_{i:j}|s_j;\phi)\exp V(s_j;\phi)}{P_F(\tau_{i:j}|s_i)\exp V(s_i;\phi)},\quad (8)$$

where $P_B(x|s_f)\exp V(s_f;\phi) := R(x)$, and $w_{i,j}$ is the weight for subtrajectory $\tau_{i:j}$, as in the Sub-TB objective. While the traditional $\lambda$-Temporal-Difference (TD) objective (52) detailed in Appendix A.5 (Niu et al., 2024; Schulman et al., 2016) focuses on learning $V(s_h;\phi)$ only from events starting at step $h$ and edge-wise mismatches $\delta_V(s \to s')$, the Sub-EB objective incorporates events both before and after $h$ and leverages subtrajectory-wise mismatches, yielding more balanced learning of $V(s_h;\phi)$. Moreover, Sub-EB allows freely weighting schemes, whereas the scheme of $\lambda$-TD is restricted to the $\lambda$-decay form. A detailed comparison between our Sub-EB and $\lambda$-TD objectives is provided in Appendix A.5.

It should be noted that the Sub-EB objective is specifically designed for estimating $V^\dagger$ of $\pi_F$. In each optimization iteration, its gradient w.r.t. $\phi$ is computed to update $V(\,\cdot\,;\phi)$, while $\theta$ is frozen. In contrast, the Sub-TB objective (2) is used to jointly update $\pi_F(\,\cdot\,|\,\cdot\,;\theta)$ and $\log F(\,\cdot\,;\theta)$.[2] We summarize the workflow of our policy-based method for GFlowNet training in Algorithm 1, where $\widehat{\mathcal{L}}_V(\phi) := \frac{1}{K}\sum_{\tau\in\mathcal{D}}[\sum_{\tau_{i:j}\in\tau} w_{i,j}\delta_V(\tau_{i:j};\phi)]$ and $\widehat{\nabla}_\theta V^\dagger(s_0;\theta) := \frac{1}{K}\sum_{\tau\in\mathcal{D}}[\sum_{h=0}^{H} A^\gamma(s_h \to s_{h+1})\nabla_\theta \log \pi_F(s_{h+1}|s_h;\theta)]$.

### 3.2 PARAMETERIZED BACKWARD POLICY

In this and the following sections, we further discuss two key advantages introduced by the Sub-EB condition, which provide the improved flexibility of policy-based training. The $\lambda$-TD objective (52) requires $\pi_B$ to remain fixed throughout optimization, as $V^\lambda$ is treated as constant w.r.t. $\phi$. To address this limitation, Niu et al. (2024) proposed a two-phase algorithm, where each training iteration includes a forward phase and a backward phase. In the forward phase, we sample $\mathcal{D} \sim P_F(\tau)$, update $V$ based on $\mathcal{D}$, and update $\pi_F$ based on $V$ and $\mathcal{D}$. In the backward phase, we sample $\mathcal{D}' \sim P_B(\tau|x)P_F(x)$, update the backward evaluation function $W$ that approximates the true backward evaluation function $W^\dagger$ of $\pi_B$, and update $\pi_B$ based on $W$. The definitions of $W^\dagger$ and $W$ will be detailed in Section 3.3. In parallel, Gritsaev et al.; Jang et al. (2024) introduced an additional objective for $\pi_B$ within the forward phase. When applied to the policy-based framework, their approach first samples a batch $\mathcal{D} \sim P_F(\tau)$ and updates $\pi_B$ by maximizing its log-likelihood, $\sum_{\tau\in\mathcal{D}} \log P_B(\tau)$. Then, with $\pi_B$ held fixed, $V$ and $\pi_F$ are updated using the same batch $\mathcal{D}$, following the procedure described above.

In contrast, both the Sub-TB (2) and Sub-EB (8) objectives allow for updating parameterized $\pi_B$ without introducing a separate backward phase or an additional objective. To be more specific, $\pi_B$ is jointly updated with $V$ under the Sub-EB objective, and with $(\pi_F, \log F)$ under the Sub-TB objective. This leads to a more streamlined and efficient training process while enabling the backward policy to adapt dynamically during optimization. Moreover, beyond basic policy-gradient methods, the Sub-EB objective also enables a parameterized $\pi_B$ within more advanced policy-based methods, such as the Trust-Region Policy Optimization (TRPO) method (Niu et al., 2024), where $\pi_B$ was previously required to remain fixed under the $\lambda$-TD objective.

### 3.3 OFFLINE POLICY-BASED TRAINING

Both the single-phase and two-phase algorithms by Niu et al. (2024) operate in an **online** manner, where we can not use a data-collection policy $\pi_\mathcal{D} \neq \pi_F$ during training. To overcome this limita-

---

[2]Here, $\theta$ and $\phi$ are introduced as separate parameter sets. $\theta$ corresponds to functions updated jointly with $\pi_F$, while $\phi$ corresponds to functions that are not.

**Algorithm 1** Online Policy-based Workflow

**Require:** $\pi_F(\cdot|\cdot;\theta)$, $\pi_B(\cdot|\cdot;\phi)$, $V(\cdot;\phi)$, batch size $K$, number of total iterations $N$
  **for** $n = 1, \ldots, N$ **do**
    $\mathcal{D} \leftarrow \{\tau^k : \tau^k \sim P_F(\tau)\}_{k=1}^K$
    Based on $\mathcal{D}$, update $\phi$ by $\nabla_\phi \widehat{\mathcal{L}}_V(\phi)$
    Based on $\mathcal{D}$ and $V$, update $\theta$ by $-\widehat{\nabla}_\theta V^\dagger(s_0; \theta)$
  **end for**
  **return** $\pi_F(\cdot|\cdot;\theta), \pi_B(\cdot|\cdot;\phi), V(\cdot;\phi)$

**Algorithm 2** Offline Policy-based Workflow

**Require:** $\pi_B(\cdot|\cdot;\phi)$, $\pi_F(\cdot|\cdot;\theta)$, $W(\cdot;\theta)$, offline policy $\pi_\mathcal{D}$, $K$, $N$
  **for** $n = 1, \ldots, N$ **do**
    $\mathcal{D}^\top \leftarrow \{x^k : x^k \in \tau^k, \tau^k \sim P_\mathcal{D}(\tau)\}_{k=1}^K$
    $\mathcal{D} \leftarrow \{\tau^k : x^k \in \mathcal{D}^\top, \tau^k \sim P_B(\tau|x^k)\}_{k=1}^K$
    Based on $\mathcal{D}$, update $\theta$ by $\nabla_\theta \widehat{\mathcal{L}}_W(\theta)$
    Based on $\mathcal{D}$ and $W$, update $\phi$ by $-\widehat{\nabla}_\phi \mathbb{E}_{P_\mathcal{D}(x)}[W^\dagger(x; \phi)]$
  **end for**
  **return** $\pi_B(\cdot|\cdot;\phi), \pi_F(\cdot|\cdot;\theta), W(\cdot;\theta)$

tion, we introduce an **offline** policy-based method made possible by the flexibility of the Sub-EB objective. We define $W^\dagger(s_0) := \log F(s_0)$. For any $h \in [H] \setminus \{0\}$, $W^\dagger(s_h; \phi)$ is defined as:

$$\mathbb{E}_{P_B(\tau_{:h}|s_h;\phi)}\left[\log \frac{P_F(\tau_{:h}|s_0)F(s_0)}{P_B(\tau_{:h}|s_h;\phi)}\right] = \log F(s_h) - D_{\mathrm{KL}}(P_B(\tau_{:h}|s_h;\phi)\|P_F(\tau_{:h}|s_h)). \quad (9)$$

The equality can be easily verified, as shown in (42) in the Appendix. It is known that minimizing $\mathbb{E}_{P_\mathcal{D}(x)}[\mathcal{L}_{KL}(\phi|x)] := \mathbb{E}_{P_\mathcal{D}(x)}[D_{\mathrm{KL}}(P_B(\tau|x;\phi)\|P_F(\tau|x))]$ also reduces the discrepancy between $P_B(\tau)$ and $P_F(\tau)$ (Niu et al., 2024). Since $\nabla_\phi \log F(x) = 0$ and $-\nabla_\phi \mathbb{E}_{P_\mathcal{D}(x)}[W^\dagger(x;\phi)] = \nabla_\phi \mathbb{E}_{P_\mathcal{D}(x)}[\mathcal{L}_{KL}(\phi|x)]$, it follows that $-\mathbb{E}_{P_\mathcal{D}(x)}[W^\dagger(x;\phi)]$ can be a surrogate to the expected KL divergence. In analogy to the forward case, we use a parameterized evaluation function $W$ to approximate $W^\dagger$, and $\nabla_\phi \mathbb{E}_{P_\mathcal{D}(x)}[W^\dagger(x;\phi)]$ is computed as follows:

$$\mathbb{E}_{P_B^\mathcal{D}(\tau)}\left[\sum_{h=0}^{H-1} A^\gamma(s_h \leftarrow s_{h+1})\nabla_\phi \log \pi_B(s_h|s_{h+1};\phi)\right] \quad (10)$$

where $A^\gamma(s_h \leftarrow s_{h+1}) := \sum_{i=0}^h \gamma^{h-i}\delta_W(s_i \rightarrow s_{i+1})$, $P_B^\mathcal{D}(\tau) := P_B(\tau|x)P_\mathcal{D}(x)$, and the definition of $\delta_W$ is given in (13).

Given a backward policy $\pi_B$, the Sub-EB condition for $W$ can be written as $\forall i < j \in [H+1]$:

$$\mathbb{E}_{P_B^\mathcal{D}(\tau_{i:j})}\left[\log\left(P_F(\tau_{i:j}|s_i)\exp W(s_i)\right)\right] = \mathbb{E}_{P_B^\mathcal{D}(\tau_{i:j})}\left[\log\left(P_B(\tau_{i:j}|s_j)\exp W(s_j)\right)\right], \quad (11)$$

where $P_B(x|s_f)\exp W(s_f) := R(x)$. The condition can be reformulated as $\mathbb{E}_{P_B^\mathcal{D}(s_i,s_j)}[W(s_i) - W(s_j)] = \mathbb{E}_{P_B^\mathcal{D}(\tau_{i:j})}\left[\log \frac{P_B(\tau_{i:j}|s_j)}{P_F(\tau_{i:j}|s_i)}\right]$, requiring learned divergence difference at $s_i$ and $s_j$ to match the true divergence between them.

**Theorem 3.3.** *Suppose $W$ is an arbitrary evaluation function over $\mathcal{S} \setminus \{s_f\}$, and $F$ is the flow associated with a forward policy $\pi_F$. Given a backward policy $\pi_B$,*

$$\forall h \in [H-1] : W(s_{h+1}) = \log F(s_{h+1}) - D_{\mathrm{KL}}(P_B(\tau_{:h+1}|s_{h+1})\|P_F(\tau_{:h+1}|s_{h+1})), \quad (12)$$

*and $W(x) = \log R(x)$ if and only if $W$ satisfies the backward Sub-EB condition (11).*

The corresponding proof can be found in Appendix A.3. Based on the backward Sub-EB condition, we present the backward Sub-EB objective for $W(\cdot;\theta)$ as follows:

$$\mathcal{L}_W(\theta) := \mathbb{E}_{P_B^\mathcal{D}(\tau)}\left[\sum_{\tau_{i:j}} w_{i,j}\left(\delta_W(\tau_{i:j};\theta)\right)^2\right], \quad \delta_W(\tau_{i:j};\theta) = \log \frac{P_F(\tau_{i:j}|s_i;\theta)\exp W(s_i;\theta)}{P_B(\tau_{i:j}|s_j)\exp W(s_j;\theta)}, \quad (13)$$

where $P_B(x|s_f)\exp W(s_f;\theta) := R(x)$. When $\pi_B$ and $\pi_F$ are at their optima, the KL term in the expression of $W$ is zero. Consequently, $F(x) = R(x)$ and $P_F(x) = F(x)/\sum_x F(x) = R(x)/Z^*$, thereby fulfilling the goal of GFlowNet training. The workflow of our offline policy-based method for GFlowNet training is presented in Algorithm 2, where $\widehat{\mathcal{L}}_W(\theta) := \frac{1}{K}\sum_{\tau \in \mathcal{D}}[\sum_{\tau_{i:j} \in \tau} w_{i,j}\delta_W(\tau_{i:j};\theta)]$ and $\widehat{\nabla}_\phi \mathbb{E}_{P_\mathcal{D}(x)}[W^\dagger(x;\phi)] := \frac{1}{K}\sum_{\tau \in \mathcal{D}}[\sum_{h=0}^{H-1} A^\gamma(s_h \leftarrow s_{h+1})\nabla_\phi \log \pi_B(s_h|s_{h+1}|\phi)]$.

## 4 RELATED WORKS

**Value-based GFlowNet Training**   Existing works on value-based GFlowNet training can be categorized into two directions. The first one focuses on designing training objectives to characterize target flow values, thereby improving the estimation of flow imbalance. The Detailed Balance (DB) objective (Bengio et al., 2023) aims to reduce the logarithmic mismatch between the forward edge flow $\pi_F(s'|s)F(s)$ and the backward target edge flow $\pi_B(s|s')F(s')$, respectively. The target term $F(s')P_B(s|s')$ provides biased but low-variance feedback, as $F$ encodes the learned partial knowledge about the task. The Trajectory Balance (TB) objective (Malkin et al., 2022a) optimizes the logarithmic mismatch between the forward trajectory flow $P_F(\tau)Z$ and the true backward trajectory flow $P_B(\tau|x)R(x) = F^*(\tau)$, which yields unbiased but high-variance feedback. The Sub-TB objective (Madan et al., 2023) generalizes both DB and TB by minimizing the flow logarithm mismatch of subtrajectories of varying lengths. By reweighting subtrajectories according to their lengths, it enables a **bias-variance** trade-off, thereby achieving better performance. Building on Sub-TB objectives, various improvements are also proposed. Kim et al. (2023a) introduced temperature-conditional objectives, whose goal turns to make $P_F(x) \propto R^\beta(x)$ with positive scalar $\beta < 1$. Taking $R$ to the exponent $\beta$ reduces its sharpness, making it easier to match. As $P_F(x)$ does not match $R(x)$ in this case, this representation is specifically useful when the focus is solely on mode seeking. For all the objectives mentioned above, $\pi_B(s|s')$ can be chosen freely for any intermediate edges $(s \to s')$ and only $\pi(x|s_f)$ is fixed to $R(x)$. Therefore, target flow values of intermediate edges or subtrajectories do not directly reflect the ground-truth knowledge about the reward function $\mathcal{R}$. However, under the special cases where the covered object space of $\mathcal{R}$ can be extended from $\mathcal{X}$ to $\mathcal{S}$, Pan et al. (2023) and Jang et al. (2023) improved the formulation of the DB and Sub-TB objectives, propagating partial knowledge of $\mathcal{R}$ directly to intermediate edges. Due to the similarity of the Sub-TB objective and our Sub-EB objective, these improvements can also be easily adapted to Sub-EB to facilitate learning $V$.

A key characteristic of value-based methods is that $\pi_\mathcal{D}$ can be off-policy, that is, different from $\pi_F$, leading to numerous efforts at designing $\pi_\mathcal{D}$ (Kim et al., 2023b; Rector-Brooks et al., 2023). The goal is to effectively identify edges that precede highly-rewarded terminating states (exploration) while allowing revisiting the edges already found to yield high reward (exploitation). The most widely used approach is $\alpha$-greedy design that mixes $\pi_F$ with a uniform policy by a factor $\alpha$. These efforts, however, fail to achieve **deep exploration**, which requires considering not only immediate information gain but also the long-term consequences of a transition (edge) in future learning (Osband et al., 2019). Although there have been many theoretical advances in efficient exploration design from an RL perspective (Azar et al., 2017; Jin et al., 2018; Ménard et al., 2021), their expensive computational cost limits their applicability in practical problems.

**Policy-based GFlowNet Training**   As mentioned above, designing a data-collection policy that is both computationally efficient and capable of **deep exploration** remains a significant challenge. Moreover, what is truly needed for training efficiency is identifying the edges of high flow-imbalance rather than the edges that lead to high terminating rewards. However, the flow imbalance is closely related to $\pi_F$ and changes whenever $\pi_F$ is updated. Empirical evidence shows that on-policy training, meaning $\pi_\mathcal{D} = \pi_F$, can result in faster convergence under many conditions (Atanackovic & Bengio, 2024). Accordingly, policy-based GFlowNet methods (Malkin et al., 2022b; Niu et al., 2024; Zimmermann et al., 2022) conduct on-policy training, which typically corresponds to optimizing the KL divergence between $P_F(\tau)$ and the unnormalized distribution $P_B(\tau|x)R(x)(= P_B(\tau)Z^*)$, which has gradient equivalence to the divergence between $P_F(\tau)$ and $P_B(\tau)$. Removing the need to design a data-collection policy, the main challenge of policy-based methods shifts to robust estimation of the divergence and its gradients, that is, balancing the trade-off between variance and bias of the estimators. Malkin et al. (2022b) and Silva et al. (2024) construct estimators empirically and require fixed $\pi_B$. During gradient-based optimization, these estimators are computed solely from the training data sampled in the current iteration. These estimators typically exhibit low bias but high variance. From the theory of policy gradients in RL, Niu et al. (2024) proposed estimators on a parameterized evaluation function $V$, which enables leveraging sampled data from all previous iterations. These estimators generally have high bias but low variance. Combining empirical and parameterized approaches, Niu et al. (2024) introduced a hyperparameter to control **variance-bias** trade-off explicitly, resulting in significant performance gains. As the effectiveness of these policy-based methods critically depends on how $V$ is learned, our paper is a subsequent work towards this challenge.

**GFlowNet training and RL** Value-based methods such as DB and Sub-TB, as well as RL-inspired variants of DB, including Munchausen DQN (denoted as Q-Much) (Tiapkin et al., 2024) and Rectified Flow Iteration (RFI) (He et al., 2025), follow the Q-learning framework (Haarnoja et al., 2017). In contrast, policy-based methods, based on policy gradients, operate within the actor–critic framework (Sutton & Barto, 2018; Haarnoja et al., 2018). A detailed discussion of the theoretical distinctions between these two frameworks is provided in Appendix A.6. To further illustrate how the Sub-EB objective and policy gradients operate within the actor-critic framework, we derive the basic soft actor–critic algorithm (Alg. 3) for GFlowNet, together with a modified variant (Alg. 4), in Appendix A.7.

## 5 EXPERIMENTS

We consider tree policy-based methods: the empirical method of Silva et al. (2024), which uses the Control Variate (CV) technique for variance reduction during gradient estimation, and two policy-gradient methods (Niu et al., 2024) that learn $V$ by either the $\lambda$-TD objective or the proposed Sub-EB objective. We refer to these methods as CV, RL, and Sub-EB, respectively. Since Sub-TB (Madan et al., 2023) (Tiapkin et al., 2024) is closely related to Sub-EB, we also include it as representative baselines for value-based methods. Among RL-inspired value-based variants, we consider Q-Much (Tiapkin et al., 2024) and RFI (He et al., 2025). However, due to potential computational scalability issues with RFI (see Appendix A.6), we include only Q-Much in our benchmarking experiments. To design the data-collection policy $\pi_{\mathcal{D}}$ in the Sub-TB method, we follow a common choice, where $\pi_{\mathcal{D}}$ is equal to $\pi_F$ with probability $(1-\alpha)$ and a uniform policy with probability $\alpha$ (Shen et al., 2023; Rector-Brooks et al., 2023). The common settings of $\pi_B$ can be either a uniform policy or a parameterized policy. By default, we adopt the uniform policy as the parameterized policy does not carry ground-truth information about the reward function. We also augmented Sub-TB and the offline Sub-EB (Algorithm 2) with the local search technique (Kim et al., 2023b) for designing $P_{\mathcal{D}}$, to explicitly promote the exploration of high-reward terminating states during trajectory sampling. These variants are denoted as Sub-TB-B and Sub-EB-B. As explained in Section 4 and confirmed by the following experiment results, off-policy techniques that explicitly encourage exploration may not benefit distribution modeling. However, when the focus is on discovering modes of terminating states, these techniques can be valuable. For Sub-EB, Sub-TB and their variants, the weight $w_{i,j}$ for $i < j \in [H+1]$ is set to $\lambda^{j-i} / \sum_{i<j\in[H+1]} \lambda^{j-i}$ following Madan et al. (2023).

Instead of the average $l_1$-distance adopted in the literature, we chose total variation $D_{\mathrm{TV}}$ and Jensen–Shannon divergence $D_{\mathrm{JSD}}$ between $P_F(x)$ and $P^*(x)$ as the metrics for performance comparison between competing methods. The rationale and formal definitions are detailed in Appendix B. We have conducted three sets of experiments. The first set is conducted in simulated environments, referred to as 'Hypergrids'. The second set focuses on biological and molecular sequence design tasks using real-world datasets. The third set involves real-world applications of GFlowNet in Bayesian Network (BN) structure learning. More experimental details, such as model configurations and hyperparameter choices, can be found in Appendix B. Parts of our implementation code are adapted from the *torchgfn* package (Lahlou et al., 2023).

**Hypergrids** Hypergrid experiments are widely used for testing GFlowNet performance (Malkin et al., 2022b; Niu et al., 2024). Here, states are the coordinate tuples of an $D$-dimensional hypercubic grid of height $H$. The detailed description of the generative process is provided in Appendix B.1. We perform experiments on $256 \times 256$, $128 \times 128 \times 128$ and $64 \times 64 \times 64$ grids. We use dynamic programming (Malkin et al., 2022a) to explicitly compute $P_F(x)$ of learned $\pi_F$, and compute the exact $D_{\mathrm{TV}}$ and $D_{\mathrm{JSD}}$ between $P_F(x)$ and $P^*(x)$. Experimental results are depicted in Fig. 2 for $D_{\mathrm{TV}}$ and Fig. 4 in Appendix B.1 for $D_{\mathrm{JSD}}$ respectively.

On the $256 \times 256$ grid, it can be observed that replacing the $\lambda$-TD objective for learning $V$ with the proposed Sub-EB objective significantly improves the stability and convergence rate of the policy-gradient method. While the final performances of the two policy-gradient methods (RL and Sub-EB) are close, they both outperform Sub-TB and CV. These experimental results strongly support the effectiveness of the Sub-EB objective in enabling more reliable learning of the evaluation function $V$, leading to improved stability and faster convergence during policy-based GFlowNet training. The empirical gradient estimator constructed solely from the current training batch is not adequate for reliably guiding policy-based training. On the $128 \times 128 \times 128$ grid, the stability and convergence

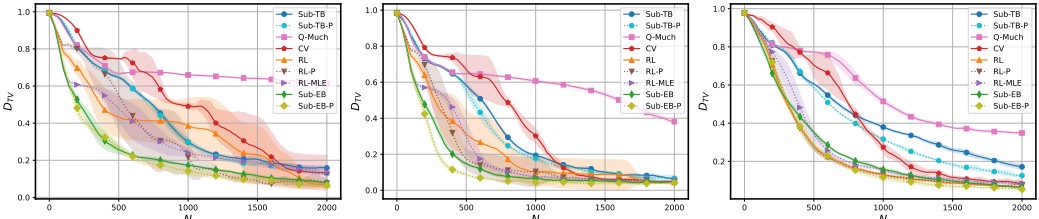

Figure 2: Plots of the means and standard deviations (represented by the shaded area) of $D_{\text{TV}}$ for different training methods with parameterized $\pi_B$ and uniform $\pi_B$ on the $256 \times 256$ (left) and $128 \times 128$ (middle) and $64 \times 64 \times 64$ (right) grids, based on five randomly started runs for each method. **By default**, metric values are recorded every 20 iterations over $N = 2000$ training iterations and smoothed by a sliding window of length 5 for all plotted curves in this paper.

rate of Sub-EB are again much better than those of RL. While CV, RL, Sub-EB and Sub-TB achieve similar final performance, both RL and Sub-EB outperform Sub-TB and CV in terms of convergence rate. These findings further validate the effectiveness of our Sub-EB objective. Finally, on the $64 \times 64 \times 64$ grid, both RL and Sub-EB outperform Sub-TB and CV, but the behavior of RL and Sub-EB is very close. This can be ascribed to the fact that the stability of RL is good enough for this experiment, so the advantages brought by the Sub-TB objectives is not obvious. Besides, the grid height $N$ can have more effect on the modeling difficulty than grid dimension $D$ since hypergrids are homogeneous w.r.t. each dimension, and the minimum distance between modes only depends on grid height $N$ (Niu et al., 2024). The training batch size is set to 128 by default, whereas a batch size of 16 and a grid height of 20 are used for hypergrid experiments in Tiapkin et al. (2024). To ensure a fair comparison and assess the impact of these differences, we additionally evaluate the performance of Sub-TB and Q-Much on $128 \times 128$ and $20 \times 20$ grids. Detailed discussions and results are provided in Appendix B.1 and Fig. 5.

**Ablation study on $\pi_B$ and $\lambda$**   To demonstrate that the Sub-EB objective naturally accommodates a parameterized $\pi_B$. We compare the performance of the different methods with parameterized and uniform $\pi_B$ on $256 \times 256$, $128 \times 128$, and $64 \times 64 \times 64$ grids. We use Sub-TB-P and Sub-EB-P to denote the corresponding methods with parameterized $\pi_B$. Furthermore, RL-P and RL-MLE denote two variants of RL that adopt the two-phase algorithm by Niu et al. (2024) and the approach by Gritsaev et al. for parameterized $\pi_B$, respectively. As CV requires fixed $\pi_B$, it does not support a parameterized $\pi_B$ and is therefore excluded from this ablation study. As shown in Fig. 2 and Fig. 4 in Appendix B.1, Sub-EB-P achieves the best performance and training stability among all the considered methods. This confirms that the Sub-EB objective well accommodates backward policies, which are parameterized and updated jointly with evaluation functions. For the $128 \times 128$ grid, we also conduct an ablation study on the hyperparameter $\lambda$ of the Sub-EB weights $w_{i,j}(= \lambda^{j-i} / \sum_{i < j \in [H+1]} \lambda^{j-i})$. Detailed discussions and results are in Appendix B.1 and Fig. 6.

**Sequence design**   In this set of experiments, we use GFlowNets to generate biological and molecular sequences of length $D$, which are composed of $M$ building blocks (Shen et al., 2023). We use nucleotide sequence datasets (SIX6 and PHO4), and molecular sequence datasets (QM9 and sEH ) from Shen et al. (2023). The detailed description of the generative process and experimental results are provided in Appendix B.2. Experimental results of RL, Sub-TB, Sub-EB, Sub-TB-B and Sub-EB-B on mode discovery and distribution modeling tasks validate that the proposed Sub-EB objective enables the integration of offline sampling techniques into policy-based methods.

**BN structure learning**   In this experiment set, we focus on real-world studies of Bayesian Network (BN) structure learning (Malkin et al., 2022b; Niu et al., 2024). Here, the object space $\mathcal{X}$ corresponds to the space of BN structures. The detailed description of the generative process is provided in Appendix B.3. We consider three cases with 5, 10, and 15 nodes, where the sizes of $\mathcal{X}$ are approximately $2.95 \times 10^4$, $4.18 \times 10^{18}$, and $2.38 \times 10^{35}$, respectively. For the large-scale cases, either $P_F(x)$ or $P^*(x)$ is computationally infeasible. Instead, we report the average reward of the top 100 unique graphs that are discovered during the training process. Since effective distribution modeling performance implies not only optimality but also diversity of generated candidates, we

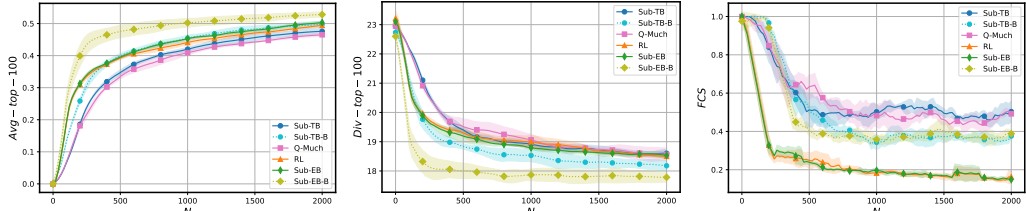

Figure 3: Plots of the mean and standard deviation values (represented by the shaded area) of average reward (left), diversity (right) and FCS (right) of the top 100 unique candidate graphs over 10 nodes, based on five randomly started runs for each method.

also compute the mean pairwise Hamming distance among these 100 graphs as a measure of diversity. It should be noted that **neither excessively high nor excessively low diversity is desirable**: the former corresponds to near-random generation, while the latter indicates that generation gets stuck in a limited set of structures. As exact computation of $D_{\mathrm{TV}}$ is infeasible, we use Flow Consistency in Sub-graphs (FCS) (Silva et al., 2025) as the unbiased estimation of $D_{\mathrm{TV}}$. For FCS, we randomly sampled 32 batches of terminal states, each of size 128, to construct the Monte Carlo estimator.

We present the discussion for the two large-scale cases below and defer the small-scale case to Appendix B.3. The corresponding experimental results are presented in Fig. 3 and Fig. 12 in the Appendix B.3. Among RL, Sub-EB, Sub-TB, and Q-Much, the results indicate that Sub-EB achieves the highest average reward. The four methods attain similar diversity. Notably, RL and Sub-EB converge faster than Sub-TB and Q-Much, and only RL and Sub-EB obtain strong distribution-modeling performance as measured by the FCS metric. Taking together all these findings, we can conclude that all methods achieve appropriate distribution modeling, and Sub-EB performs the best. This supports that Sub-EB not only enables reliable policy-based training but also scales effectively to large combinatorial spaces, providing both high-quality and diverse solutions. For the two variants, Sub-TB-B and Sub-EB-B, it can be observed that Sub-EB-B achieves the highest average reward among all six methods, accompanied by a more noticeable drop in diversity. Given that the local search component explicitly prioritizes high-reward regions of the solution space, such a trade-off—significant reward improvement at the expense of reduced diversity—is expected. In contrast, Sub-TB-B does achieve a higher average reward compared to its non-augmented counterpart (Sub-TB) with a moderate decrease in diversity. However, the trade-off becomes much less pronounced. Without the local search technique, Sub-EB already achieves a comparable average reward and higher diversity than Sub-TB-B. Overall, Sub-EB-B proves to be more effective than Sub-TB-B, aligning well with our expectations of the optimality-diversity trade-off. This finding further supports the superiority of the policy-based methods and validates that the Sub-EB objective enables the integration of offline techniques within policy-based frameworks.

**Molecular graph design** In this set of experiments, we consider the molecular graph design task (with $|\mathcal{X}| \approx 10^{16}$) described in Bengio et al. (2021). The sequence design task based on the sEH dataset (with $|\mathcal{X}| \approx 3.4 \times 10^7$) (Shen et al., 2023) is simplified from these tasks. A detailed description of the generative process and the corresponding experimental results for these graph design tasks is provided in the Appendix B.4. The experimental results demonstrate that Sub-EB achieves strong overall performance on large-scale molecular graph design, providing higher average rewards, faster convergence, and competitive diversity compared to RL, Sub-TB and Q-Much.

## 6 DISCUSSION AND CONCLUSION

In this work, we have established the connection between the state flow function $F(s)$ and the evaluation function $V(s)$. Built upon that, a new objective, called Sub-EB, is proposed for learning the evaluation function $V(s)$. Through four sets of experiments, we provide empirical evidence that the new Sub-EB objective enables more stable and flexible learning of $V$ than the $\lambda$-TD objective, thereby improving the performance of the policy-based methods for GFlowNet training. In principle, the Sub-EB objective allows flexible choices of weight coefficients and goes beyond policy-gradient methods. Further investigation into designing optimal weight coefficients and integrating the Sub-EB objective into more advanced policy-based methods is left for future work.

ACKNOWLEDGEMENTS

This work was supported in part by the U.S. National Science Foundation (NSF) grants SHF-2215573 and IIS-2212419. Portions of this research were conducted with the advanced computing resources provided by Texas A&M High Performance Research Computing.

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

## LLM Usage Disclosure

LLMs were used only for text refinement (grammar and style). All scientific content was developed and verified by the authors.

## A  Theoretical analyses

### A.1  Proof of Theorem 3.1

*Proof.* We first prove the **sufficiency** of the Sub-EB condition (5). Assume that $V$ is an evaluation function over $\mathcal{S}$ that satisfies the Sub-EB condition for a given forward policy $\pi_F$. Then, for any $h \in [H]$:

$$\mathbb{E}_{P_F(s_h)\pi_F(s_{h+1}|s_h)}\left[\log \pi_B(s_h|s_{h+1}) + V(s_{h+1}) - \log \pi_F(s_{h+1}|s_h) - V(s_h)\right] = 0 \tag{14}$$

$$\Downarrow$$

$$\mathbb{E}_{\pi_F(s_{h+1}|s_h)}\left[\log \pi_B(s_h|s_{h+1}) + V(s_{h+1}) - \log \pi_F(s_{h+1}|s_h) - V(s_h)\right] = 0 \tag{15}$$

$$\Downarrow$$

$$V(s_h) = \mathbb{E}_{\pi_F(s_{h+1}|s_h)}\left[\log \frac{\pi_B(s_h|s_{h+1})}{\pi_F(s_{h+1}|s_h)} + V(s_{h+1})\right]. \tag{16}$$

Here, the second equation holds by our assumption that any valid $\pi_F$ introduces a trajectory distribution $P_F(\tau)$ that is strictly positive for any $\tau \in \mathcal{T}$. Consequently, the corresponding state probability $P_F(s)$ is strictly positive for any $s \in \mathcal{S}$.

Based on (16), the previous definition $\pi_B(x|s_f)\exp V(s_f) := R(x)$, we have:

$$V(x) = \mathbb{E}_{\pi_F(s_f|x)}\left[\log \frac{R(x)}{\pi_F(s_f|x)}\right]$$

$$= \mathbb{E}_{\pi_F(s_f|x)}\left[\log \frac{F^*(x \to s_f)}{\pi_F(s_f|x)}\right]$$

$$= \mathbb{E}_{P_F(x \to s_f|x)}\left[\log \frac{P_B(x \to s_f)Z^*}{P_F(x \to s_f|x)}\right]$$

$$= \mathbb{E}_{P_F(x \to s_f|x)}\left[\log \frac{P_B(x \to s_f|x)}{P_F(x \to s_f|x)}\right] + \log P_B(x)Z^*$$

$$= \log F^*(x) - D_{\mathrm{KL}}(P_F(\tau_{H:}|s_H)\|P_B(\tau_{H:}|s_H)),$$

$$V(s_{H-1}) = \mathbb{E}_{\pi_F(x|s_{H-1})}\left[\log \frac{\pi_B(s_{H-1}|x)}{\pi_F(x|s_{H-1})} + \mathbb{E}_{P_F(x \to s_f|x)}\left[\log \frac{P_B(x \to s_f)Z^*}{P_F(x \to s_f|x)}\right]\right]$$

$$= \mathbb{E}_{P_F(\tau_{H-1:}|s_{H-1})}\left[\log \frac{P_B(\tau_{H-1:})Z^*}{P_F(\tau_{H-1:}|s_{H-1})}\right]$$

$$= \mathbb{E}_{P_F(\tau_{H-1:}|s_{H-1})}\left[\log \frac{P_B(\tau_{H-1:}|s_{H-1})}{P_F(\tau_{H-1:}|s_{H-1})}\right] + \log P_B(s_{H-1})Z^*$$

$$= \log F^*(s_{H-1}) - D_{\mathrm{KL}}(P_F(\tau_{H-1:}|s_{H-1})\|P_B(\tau_{H-1:}|s_{H-1})), \tag{17}$$

$$\vdots$$

$$V(s_h) = \mathbb{E}_{\pi_F(s_{h+1}|s_h)}\left[\log \frac{\pi_B(s_h|s_{h+1})}{\pi_F(s_{h+1}|s_h)} + \mathbb{E}_{P_F(\tau_{h+1:}|s_{h+1})}\left[\log \frac{P_B(\tau_{h+1:})Z^*}{P_F(\tau_{h+1:}|s_{h+1})}\right]\right]$$

$$= \mathbb{E}_{P_F(\tau_{h:}|s_h)}\left[\log \frac{P_B(\tau_{h:})Z^*}{P_F(\tau_{h:}|s_h)}\right] \tag{18}$$

$$= \mathbb{E}_{P_F(\tau_{h:}|s_h)}\left[\log \frac{P_B(\tau_{h:}|s_h)}{P_F(\tau_{h:}|s_h)}\right] + \log P_B(s_h)Z^*$$

$$= \log F^*(s_h) - D_{\mathrm{KL}}(P_F(\tau_{h:}|s_h)\|P_B(\tau_{h:}|s_h)), \tag{19}$$

$$\vdots$$

$$V(s_0) = \mathbb{E}_{P_F(\tau|s_h)} \left[ \log \frac{P_B(\tau)Z^*}{P_F(\tau)} \right]$$

$$= \mathbb{E}_{P_F(\tau)} \left[ \log \frac{P_B(\tau)}{P_F(\tau)} \right] + \log Z^*$$

$$= \log F^*(s_0) - D_{\mathrm{KL}}(P_F(\tau) \| P_B(\tau)). \tag{20}$$

Now, we prove the **necessity** of the Sub-EB condition (5). Assume that $V$ is the evaluation function associated with a forward policy $\pi_F$. Then, $\forall h \in [H]$:

$$V(s_h) = \mathbb{E}_{P_F(\tau_{h:}|s_h)} \left[ \sum_{i=h}^{H} \log \frac{\pi_B(s_i|s_{i+1})}{\pi_F(s_{i+1}|s_i)} \right] + \log Z^*$$

$$= \mathbb{E}_{\pi_F(s_{h+1}|s_h)} \left[ \log \frac{\pi_B(s_h|s_{h+1})}{\pi_F(s_{h+1}|s_h)} + \mathbb{E}_{P_F(\tau_{h+1:}|s_{h+1})} \left[ \log \frac{P_B(\tau_{h+1:})Z^*}{P_F(\tau_{h+1:}|s_{h+1})} \right] \right]$$

$$= \mathbb{E}_{\pi_F(s_{h+1}|s_h)} \left[ \log \frac{\pi_B(s_h|s_{h+1})}{\pi_F(s_{h+1}|s_h)} + V(s_{h+1}) \right]. \tag{21}$$

Therefore,

$$\mathbb{E}_{P_F(s_h)\pi_F(s_{h+1}|s_h)} \left[ \log \pi_B(s_h|s_{h+1}) + V(s_{h+1}) - \log \pi_F(s_{h+1}|s_h) - V(s_h) \right] = 0 \tag{22}$$

$$\Downarrow$$

$$\sum_{l=i}^{j-1} \mathbb{E}_{P_F(s_l \to s_{l+1})} \left[ \log \pi_B(s_l|s_{l+1}) + V(s_{l+1}) - V(s_l) - \log \pi_F(s_{l+1}|s_l) \right] = 0 \tag{23}$$

$$\Downarrow$$

$$\mathbb{E}_{P_F(\tau_{i:j})} \left[ \sum_{l=i}^{j-1} \log \pi_B(s_l|s_{l+1}) + V(s_j) - V(s_i) - \sum_{l=i}^{j-1} \log \pi_F(s_{l+1}|s_l) \right] = 0 \tag{24}$$

$$\Downarrow$$

$$\mathbb{E}_{P_F(\tau_{i:j})} \left[ \log P_B(\tau_{i:j}|s_j) + V(s_j) - \log P_F(\tau_{i:j}|s_i) - V(s_i) \right] = 0 \tag{25}$$

for any $i < j \in [H+1]$. $\qquad \square$

## A.2 PROOF OF THEOREM 3.2

*Proof.* We first prove the **sufficiency** of the Sub-TB condition (1). Assume that $F$ is a state flow function over $\mathcal{S}$, and $\pi_F$ is a forward policy, such that they satisfy the Sub-TB condition. Then, for any $h \in [H]$:

$$\mathbb{E}_{P_\mathcal{D}(s_h \to s_{h+1})} \left[ \log \pi_B(s_h|s_{h+1}) F(s_{h+1}) - \log \pi_F(s_{h+1}|s_h) F(s_h) \right] = 0 \tag{26}$$

$$\Downarrow$$

$$\log \pi_B(s_h|s_{h+1}) F(s_{h+1}) - \log \pi_F(s_{h+1}|s_h) F(s_h) = 0 \tag{27}$$

$$\Downarrow$$

$$\log F(s_h) = \log \frac{\pi_B(s_h|s_{h+1})}{\pi_F(s_{h+1}|s_h)} + \log F(s_{h+1}). \tag{28}$$

Here, the second equation holds due to the following two reasons. First, by our assumption that the trajectory distribution $P_\mathcal{D}$, which is induced by $\pi_\mathcal{D}$, assigns positive probability to all trajectories in $\mathcal{T}$. Thus, the marginal probability $P_\mathcal{D}(s \to s')$ is strictly positive for any $(s \to s') \in \mathcal{E}$. Second, $\pi_\mathcal{D}$ is arbitrarily constructed and may differ from $\pi_F$. Let

$$\pi^\dagger(s'|s) := \frac{\pi_B(s|s')F(s')}{\sum_{s'} \pi_B(s|s')F(s')}. \tag{29}$$

Then, taking the expectation of (28) w.r.t. $\pi_F$, we have for any $h \in [H]$:

$$\log F(s_h) = \mathbb{E}_{\pi_F(s_{h+1}|s_h)} \left[ \log \frac{\pi_B(s_h|s_{h+1})}{\pi_F(s_{h+1}|s_h)} + \log F(s_{h+1}) \right] \tag{30}$$

$$= \mathbb{E}_{\pi_F(s_{h+1}|s_h)} \left[ \log \frac{\pi^\dagger(s_{h+1}|s_h)}{\pi_F(s_{h+1}|s_h)} \right] + \log \sum_{s_{h+1}} \pi_B(s_h|s_{h+1}) F(s_{h+1})$$

$$= -D_{\mathrm{KL}}(\pi_F(\cdot|s_h) \| \pi^\dagger(\cdot|s_h)) + \log \sum_{s_{h+1}} \pi_B(s_h|s_{h+1}) F(s_{h+1}). \tag{31}$$

Note that the second term in the last equality is independent of $\pi_F(\cdot|s_h)$. Then, given $\pi_B(s_h|\cdot)$ and $F(s_{h+1})$, $F(s_h)$ is maximized at $s_h$ when $\pi_F(\cdot|s_h) = \pi^\dagger(\cdot|s_h)$. Accordingly, given $\pi_B(s_{h+1}|\cdot)$ and $F(s_{h+2})$, $F(s_{h+1})$ is maximized at $s_{h+1}$ when $\pi_F(\cdot|s_{h+1}) = \pi^\dagger(\cdot|s_{h+1})$. Keeping doing this recursion from $h = 0$ to $h = H$, we get the conclusion that $F$ is maximized when $\pi_F = \pi^\dagger$ for any $(s \to s') \in \mathcal{E}$.

Taking the exponential of (27) and summing over $s_{h+1}$ yields $\sum_{s_{h+1}} \log \pi_B(s_h|s_{h+1}) F(s_{h+1}) = \sum_{s_{h+1}} \log F(s_h) \pi_F(s_{h+1}|s_h) = F(s_h)$. Inserting this back into (27), we arrive at $\pi_F = \pi^\dagger$. Achieving the maximum, we have $\log F(x) = \log \sum_{s_f} \pi(x|s_f) F(s_f) := \log R(x) = \log F^*(x)$, $\log F(s_{H-1}) = \log \sum_x \pi_B(s_{H-1}|x) F^*(x) = \log F^*(s_{H-1}), \ldots, \log F(s_0) = \log \sum_{s_1} \pi_B(s_0|s_1) F^*(s_1) = F^*(s_0)$. Therefore, $F(s) = F^*(s)$ for any $s \in \mathcal{S}$. Combining this with (27), we have $\pi_F = \pi_F^*$, where $\pi_F^*(s'|s) = \frac{\pi_B(s|s') F^*(s')}{F^*(s)}$ is the optimal forward policy induced by $\pi_B$. Moreover, based on (30) and a derivation analogous to that of $V$, we have, for Markovian $P_B$ and any $h \in [H]$:

$$\log F(s_h) := \log F^*(s_h) - D_{\mathrm{KL}}(P_F(\tau_{h:}|s_h) \| P_B(\tau_{h:}|s_h)). \tag{32}$$

Therefore, when $\pi_F = \pi_F^*$, we have:

$$\log F(s_h) = \log F^*(s_h) - D_{\mathrm{KL}}(P_F^*(\tau_{h:}|s_h) \| P_B(\tau_{h:}|s_h)). \tag{33}$$

Now we prove the **necessity** of the Sub-TB condition (1). Assume that $F$ and $\pi_F$ satisfy the equation (33) above. Then, the equation (32) must also hold for $F$ and $\pi_F$, and we have, for any $h \in [H]$:

$$\log F(s_h) = \mathbb{E}_{P_F(\tau_{h:}|s_h)} \left[ \log \frac{P_B(\tau_{h:}) Z^*}{P_F(\tau_{h:}|s_h)} \right]$$

$$= \mathbb{E}_{P_F(\tau_{h:}|s_h)} \left[ \sum_{i=h}^{H} \log \frac{\pi_B(s_i|s_{i+1})}{\pi_F(s_{i+1}|s_i)} \right] + \log Z^*$$

$$= \mathbb{E}_{\pi_F(s_{h+1}|s_h)} \left[ \log \frac{\pi_B(s_h|s_{h+1})}{\pi_F(s_{h+1}|s_h)} + \mathbb{E}_{P_F(\tau_{h+1:}|s_{h+1})} \left[ \log \frac{P_B(\tau_{h+1:}) Z^*}{P_F(\tau_{h+1:}|s_{h+1})} \right] \right]$$

$$= \mathbb{E}_{\pi_F(s_{h+1}|s_h)} \left[ \log \frac{\pi_B(s_h|s_{h+1})}{\pi_F(s_{h+1}|s_h)} + \log F(s_{h+1}) \right]. \tag{34}$$

We can rewrite the above equation as follows:

$$\log F(s_h) := -D_{\mathrm{KL}}(\pi_F(\cdot|s_h) \| \pi^\dagger(\cdot|s_h)) + \log \sum_{s_{h+1}} \pi_B(s_h|s_{h+1}) F(s_{h+1}), \quad \forall h \in [H]. \tag{35}$$

Since $\pi_F(\cdot|s_h)$ is independent of $\pi_B(s_h|\cdot)$ and $F(s_{h+1})$, the assumption that $\log F(s_h)$ is maximized at $s_h$ indicates $\pi_F(\cdot|s_h) = \pi^\dagger(\cdot|s_h)$. Therefore, we recover $\forall h \in [H]$:

$$\log \pi_F(s_{h+1}|s_h) F(s_h) = \log \pi_B(s_h|s_{h+1}) F(s_{h+1}), \tag{36}$$

and the Sub-TB condition can be easily derived from it. □

### A.3 Proof of Theorem 3.3

*Proof.* We first prove the **sufficiency** of the backward Sub-EB condition (11). Assume that $W$ is an evaluation function over $\mathcal{S} \backslash \{s_f\}$ that satisfies the backward Sub-EB condition for a given backward policy $\pi_B$. Then, for any $h \in [H - 1]$:

$$\mathbb{E}_{\pi_B(s_h|s_{h+1}) P_B(s_{h+1})} \left[ \log \pi_B(s_h|s_{h+1}) + W(s_{h+1}) - \log \pi_F(s_{h+1}|s_h) - W(s_h) \right] = 0 \tag{37}$$

$$\Downarrow$$

$$\mathbb{E}_{\pi_B(s_h|s_{h+1})}\left[\log\pi_B(s_h|s_{h+1}) + W(s_{h+1}) - \log\pi_F(s_{h+1}|s_h) - W(s_h)\right] = 0 \qquad (38)$$

$$\Downarrow$$

$$W(s_{h+1}) = \mathbb{E}_{\pi_B(s_h|s_{h+1})}\left[\log\frac{\pi_F(s_{h+1}|s_h)}{\pi_B(s_h|s_{h+1})} + W(s_h)\right]. \qquad (39)$$

Here, the second equation holds by our assumption that any valid $\pi_B$ and $R$ should introduce a a trajectory distribution $P_B(\tau)$ that is strictly positive for any $\tau \in \mathcal{T}$. Consequently, the corresponding state probability $P_B(s)$ is strictly positive for any $s \in \mathcal{S}$.

Based on (39), we have:

$$\begin{aligned}
W(s_1) &= \mathbb{E}_{\pi_B(s_0|s_1)}\left[\log\frac{\pi_F(s_1|s_0)}{\pi_B(s_0|s_1)}\right] + W(s_0) \\
&= \mathbb{E}_{P_B(s_0\to s_1|s_1)}\left[\log\frac{P_F(s_0\to s_1)}{P_B(s_0\to s_1|s_1)}\right] + \log F(s_0) \\
&= \mathbb{E}_{P_B(s_0\to s_1|s_1)}\left[\log\frac{P_F(s_0\to s_1|s_1)}{P_B(s_0\to s_1|s_1)}\right] + \log F(s_0)P_F(s_1) \\
&= \log F(s_1) - D_{\mathrm{KL}}(P_B(\tau_{:1}|s_1)\|P_F(\tau_{:1}|s_1)), \qquad (40)
\end{aligned}$$

$$\vdots$$

$$\begin{aligned}
W(s_{h+1}) &= \mathbb{E}_{\pi_B(s_h|s_{h+1})}\left[\log\frac{\pi_F(s_{h+1}|s_h)}{\pi_B(s_h|s_{h+1})} + \mathbb{E}_{P_B(\tau_{:h}|s_h)}\left[\log\frac{P_F(\tau_{:h})}{P_B(\tau_{:h}|s_h)}\right]\right] + W(s_0) \\
&= \mathbb{E}_{P_B(\tau_{:h+1}|s_{h+1})}\left[\log\frac{P_F(\tau_{:h+1})}{P_B(\tau_{:h+1}|s_{h+1})}\right] + \log F(s_0) \qquad (41) \\
&= \mathbb{E}_{P_B(\tau_{:h+1}|s_{h+1})}\left[\log\frac{P_F(\tau_{:h+1}|s_{h+1})}{P_B(\tau_{:h+1}|s_{h+1})}\right] + \log F(s_0)P_F(s_{h+1}) \\
&= \log F(s_{h+1}) - D_{\mathrm{KL}}(P_F(\tau_{:h+1}|s_{h+1})\|P_B(\tau_{:h}|s_{h+1})), \qquad (42)
\end{aligned}$$

$$\vdots$$

$$\begin{aligned}
W(x) &= \mathbb{E}_{\pi_B(s_{H-1}|x)}\left[\log\frac{\pi_F(x|s_{H-1})}{\pi_B(s_{H-1}|x)} + \mathbb{E}_{P_B(\tau_{:H-1}|s_{H-1})}\left[\log\frac{P_F(\tau_{:H-1})}{P_B(\tau_{:H-1}|s_{H-1})}\right]\right] + W(s_0) \\
&= \mathbb{E}_{P_B(\tau|x)}\left[\log\frac{P_F(\tau)}{P_B(\tau|x)}\right] + \log F(s_0) \\
&= \mathbb{E}_{P_B(\tau|x)}\left[\log\frac{P_F(\tau|x)}{P_B(\tau|x)}\right] + \log F(s_0)P_F(x) \\
&= \log F(x) - D_{\mathrm{KL}}(P_B(\tau|x)\|P_F(\tau|x)). \qquad (43)
\end{aligned}$$

In particular, when the backward Sub-EB condition is satisfied for $h = H$, we have

$$\mathbb{E}_{P_\mathcal{D}(x)}[\log\pi_F(s_f|x) + W(x) - \log\pi_B(x|s_f) - W(s_f)] = 0 \qquad (44)$$

$$\Downarrow$$

$$\mathbb{E}_{P_\mathcal{D}(x)}[W(x) - \log R(x)] = 0. \qquad (45)$$

Since $\pi_F$ is arbitrarily chosen provided that $P_\mathcal{D}(\tau) > 0$ for any $\tau \in \mathcal{T}$, it follows that $P_\mathcal{D}(x) > 0$ for any $x \in \mathcal{X}$ and $W(x) = \log R(x)$.

Now, we prove the **necessity** of backward the Sub-EB condition (11). Assume that $W$ is the evaluation function of a given forward policy $\pi_B$. Then, $\forall h \in [H-1]$:

$$\begin{aligned}
W(s_{h+1}) &= \mathbb{E}_{P_B(\tau_{:h+1}|s_{h+1})}\left[\sum_{i=0}^{h}\log\frac{\pi_F(s_{i+1}|s_i)}{\pi_B(s_i|s_{i+1})}\right] + W(s_0) \\
&= \mathbb{E}_{\pi_B(s_h|s_{h+1})}\left[\log\frac{\pi_F(s_{h+1}|s_h)}{\pi_B(s_h|s_{h+1})} + \mathbb{E}_{P_B(\tau_{:h}|s_h)}\left[\log\frac{P_F(\tau_{:h})F(s_0)}{P_B(\tau_{:h}|s_h)}\right]\right]
\end{aligned}$$

$$= \mathbb{E}_{\pi_B(s_h|s_{h+1})} \left[ \log \frac{\pi_F(s_{h+1}|s_h)}{\pi_B(s_h|s_{h+1})} + W(s_h) \right]. \tag{46}$$

and $W(x) = \log R(x)$. Therefore,

$$\mathbb{E}_{P_B^{\mathcal{D}}(s_{h+1})\pi_B(s_h|s_{h+1})} \left[ \log \pi_B(s_h|s_{h+1}) + W(s_{h+1}) - \log \pi_F(s_{h+1}|s_h) - W(s_h) \right] = 0 \tag{47}$$

$$\Downarrow$$

$$\sum_{l=i}^{j-1} \mathbb{E}_{P_B^{\mathcal{D}}(s_l \to s_{l+1})} \left[ \log \pi_B(s_l|s_{l+1}) + W(s_{l+1}) - W(s_l) - \log \pi_F(s_{l+1}|s_l) \right] = 0 \tag{48}$$

$$\Downarrow$$

$$\mathbb{E}_{P_B^{\mathcal{D}}(\tau_{i:j})} \left[ \sum_{l=i}^{j-1} \log \pi_B(s_l|s_{l+1}) + W(s_j) - W(s_i) - \sum_{l=i}^{j-1} \log \pi_F(s_{l+1}|s_l) \right] = 0 \tag{49}$$

$$\Downarrow$$

$$\mathbb{E}_{P_B^{\mathcal{D}}(\tau_{i:j})} \left[ \log P_B(\tau_{i:j}|s_j) + W(s_j) - \log P_F(\tau_{i:j}|s_i) - W(s_i) \right] = 0 \tag{50}$$

for any $i < j \in [H+1]$ $\qquad\qquad\square$

## A.4 Evaluation Function with Total Flow Estimator

When incorporating the total-flow estimator $Z$, the true evaluation function $V^\dagger$ still takes the form of (3), only differing by replacing $Z^*$ with $Z^*/Z$. In this case, it can be verified that $V^\dagger(s_0; \theta) = \log \frac{Z^*}{Z(\theta)} - D_{\mathrm{KL}}(P_F(\tau)\|P_B(\tau))$. Then, the gradient of $-\mathbb{E}_{\mu(s_0;\theta)}[V^\dagger(s_0;\theta)]$ is equal to that of $D_{\mathrm{KL}}(P_F(\tau;\theta)\|P_B(\tau)) + \frac{1}{2}(\log Z(\theta) - \log Z^*)^2$, where $\mu(s_0;\theta) := Z(\theta)/Z$ as $s_0$ is unique (Niu et al., 2024).

The Sub-EB condition and objective remain unchanged, except that $P_B(x|s_f) \exp V(s_f)$ is refined as $R(x)/Z$.

**Corollary A.1** (Corollary to Theorem 3.1). *Suppose $V$ is an arbitrary evaluation function over $\mathcal{S}$, and $F^*$ is the optimal flow induced by a backward policy $\pi_B$. Given a forward policy $\pi_F$,*

$$\forall h \in [H] : V(s_h) = \log \frac{F^*(s_h)}{Z} - D_{\mathrm{KL}}(P_F(\tau_{h:}|s_h)\|P_B(\tau_{h:}|s_h)), \tag{51}$$

*if and only if $V$ satisfies the Sub-EB condition (5).*

*Proof.* The proof can be done by replacing $R(x)$ in the proof of Theorem 3.1 by $R(x)/Z$ $\qquad\square$

Finally, in Algorithm 1, $\nabla_\theta \mathbb{E}_{\mu(s_0;\theta)}[V^\dagger(s_0;\theta)] = \mathbb{E}_{\mu(s_0)}[V^\dagger(s_0)\nabla_\theta \log Z(\theta) + \nabla_\theta V^\dagger(s_0;\theta)]$ is approximated by $\widehat{\nabla}_\theta \mathbb{E}_{\mu(s_0;\theta)}[V^\dagger(s_0;\theta)] := \frac{1}{K} \sum_{\tau \in \mathcal{D}} [\sum_{h=0}^{H} R(s_h \to s_{h+1})\nabla_\theta \log Z(\theta) + \sum_{h=0}^{H} A^\gamma(s_h \to s_{h+1})\nabla_\theta \log \pi_F(s_{h+1}|s_h;\theta)]$.

## A.5 Comparison between $\lambda$-TD and Sub-EB objectives

The traditional $\lambda$-TD objective for $V(\cdot; \phi)$ can be expressed as follows:

$$\mathbb{E}_{P_F(\tau)} \left[ \sum_{h=0}^{H} \left( V^\lambda(s_h) - V(s_h; \phi) \right)^2 \right],$$

$$V^\lambda(s_h) := V(s_h) + \sum_{i=h}^{H} \lambda^{i-h} \delta_V(s_i \to s_{i+1}), \tag{52}$$

where $V^\lambda$ is considered **constant** when computing gradients w.r.t. $\phi$, and $\delta_V(s_i \to s_{i+1})$ is equal to $\delta_V(\tau_{i,i+1})$ as defined in (8). Without consideration of gradient computation, the expression of the objective value can be simplified as:

$$\mathbb{E}_{P_F(\tau)} \left[ \sum_{h=0}^{H} \left( \sum_{i=h}^{H} \lambda^{i-h} \delta_V(s_i \to s_{i+1}) \right)^2 \right]. \tag{53}$$

In comparison, the expression of the Sub-EB objective value is:

$$\mathbb{E}_{P_F(\tau)} \left[ \sum_{\tau_{i:j} \,:\, i < j \in [H+1]} w_{i,j} (\delta_V(\tau_{i:j}))^2 \right]. \tag{54}$$

It can be observed that the $\lambda$-TD objective only considers the events that start at step $h$ for learning $V(s_h)$, and edges-wise mismatch $\delta_V(s_i \to s_{i+1})$. In contrast, the Sub-EB objective incorporates information from both events that start at $h$ and those that end at $h$ by considering subtrajectory-wise mismatches $\delta_V(\tau_{i:(\cdot)})$ that start at $i (\geq h)$ and $\delta_V(\tau_{(\cdot):j})$ that end at $j (\leq h)$. This results in a more balanced and reliable learning of $V(s_h)$. Besides, the form of $w$ can be freely chosen, while it must be $\lambda^{i-h}$ in the $\lambda$-TD objective (Schulman et al., 2016).

## A.6 GFLOWNET AND RL

RL methods can be roughly categorized into two main frameworks (Sutton & Barto, 2018): the first is the Q-learning framework, and the second is the actor–critic framework.

**Q-learning with Q-parameterization**  Under the Q-learning framework, the core idea, exemplified by the soft Q-learning algorithm, is to learn a parameterized function $Q$ that minimizes the **offline** Bellman objective for the transition environment $\mathcal{G}$ with edge reward $\log \pi_B(s|s')$ and $\log \pi_B(x|s_f) + V(s_f) := \log R(x)$ (Haarnoja et al., 2017). This objective function can be written as

$$\mathbb{E}_{P_{\mathcal{D}}(s \to s')}[(\delta_Q(s \to s'))^2], \quad \delta_Q(s \to s') := \log \pi_B(s|s') + V(s') - Q(s, s'), \tag{55}$$

with

$$V(s) := \log \sum_{s'} \exp Q(s, s'), \quad \pi_F(s'|s) := \frac{\exp Q(s, s')}{\exp V(s)}.$$

Any function $Q$ that achieves zero mismatch is guaranteed to equal the optimal soft Q-function $Q^*$ with the corresponding $V = V^*$ and $\pi_F = \pi_F^*$. Tiapkin et al. (2024) showed that if we treat $Q(s, s')$ as $\log F(s \to s')$ such that $V(s) = \log \sum_{s'} F(s \to s') = \log F(s)$, then the Bellman objective transforms into the DB objective. The distinction lies only in parameterization: instead of parameterizing $Q(s, s')$ directly and deriving $V$ and $\pi_F$ from it, one may parameterize $(\pi_F, V)$ jointly and define $Q(s, s') := \pi_F(s'|s) + \log V(s)$[3]. The authors further showed that the optimal solutions of the Bellman objective coincide with those of the DB objective from the perspective of Soft Q-learning. However, as acknowledged by the authors, this result applies only to the DB objective with a fixed $\pi_B$. To address this limitation, Deleu et al. (2024) established an equivalence between path-consistency learning (which incorporates soft Q-learning as a special case) and the Sub-TB objective (which incorporates DB as a special case) from a gradient-based perspective. Compared to Deleu et al. (2024), our Theorem 3.2 offers a more direct and explicit connection along this RL direction.

**Q-learning with V-parameterization**  In analogy to soft Q-learning, we may instead parameterize $V$ and define:

$$Q(s, s') := \log \pi_F(s'|s) + V(s), \quad \pi_F(s'|s) := \frac{\pi_B(s|s') \exp V(s')}{\sum_{s'} \pi_B(s|s') \exp V(s')}.$$

Then, we learn $V$ via the state-level Bellman objective (in a dynamic programming style):

$$\mathbb{E}_{P_{\mathcal{D}}(s)}[(\delta_E(s))^2], \quad \delta_E(s) := \log \sum_{s'} \pi_B(s|s') \exp V(s') - \log \sum_{s'} \exp Q(s, s'), \tag{56}$$

where $\log \sum_{s'} \exp Q(s, s') = \log \sum_{s'} \pi_F(s'|s) \exp V(s) = V(s)$. Notably, RFI (He et al., 2025) follows this formulation, differing in that it parameterizes $F(s)$ as $\exp V(s)$ and multiplies $F(s)$ by a rectified scalar $g(s)$. However, this definition of $\pi_F$ can lead to computational scalability issues

---

[3]To be specific, we have $\delta_Q(s' \to s') = \log \pi_B(s|s') \exp V(s') - \log \pi_F(s'|s) \exp V(s) = \log \pi_B(s|s') F(s') - \log \pi_F(s'|s) F(s)$, coinciding with the formulation in the DB objective.

when sampling trajectories. Let $Ch(s) : \{s' \in \mathcal{S} \mid (s \to s') \in \mathcal{E}\}$ denote the set of child states of state $s$. As $\pi_\mathcal{D}$ is equal to or design based on $\pi_F$, making a transition from $\pi_\mathcal{D}(\cdot \mid s)$ at state $s$ requires $|Ch(s)|$ separate passes through $V$ (a neural network with scalar outputs) to obtain the value of $V(s')$ for every $s' \in Ch(s)$. In contrast, $\pi_F(\cdot \mid s)$ can be readily obtained from $Q$ (a neural network with vector outputs), where a single pass at $s$ yields $Q(s, s')$ for every $s' \in Ch(s)$. In tasks like BN structure learning, where $|Ch(s)|$ can be larger than $10^2$, methods that only parameterize $V$ can be much slower than those that jointly parameterize $(V, \pi_F)$ or directly parameterize $Q$.

**Actor-Critic with V-parameterization**  Our work presented here, together with Niu et al. (2024), is grounded in on the theory of policy-gradient (Agarwal et al., 2021), which operates under the actor–critic framework (Sutton & Barto, 2018; Haarnoja et al., 2018). In this framework, both $\pi_F$ and $V$ are parameterized. The central idea of the soft actor-critic algorithm is to learn $V$ that minimizes (though not necessarily solves) the **online** Bellman objective:

$$\mathbb{E}_{P_F(s \to s')}[(\delta_V(s \to s'))^2], \quad \delta_V(s \to s') = \log \pi_F(s'|s) + V(s) - Q(s, s'), \quad (57)$$

with

$$Q(s, s') := \log \pi_B(s|s') + V(s').$$

The expectation $\mathbb{E}_{P_F(s \to s')}[\ldots]$ above can be generalized into $\mathbb{E}_{P_\mathcal{D}(s)}[\mathbb{E}_{\pi_F(s'|s)}[\ldots]]$, but the inner online expectation must still be maintained. Notably, the online Bellman objective can be viewed as a special case of the proposed Sub-EB objective. **At this point, the two RL directions begin to diverge** (Schulman et al., 2017). When $V$ achieves the optimal solution of the objective, we have:

$$V(s) = \mathbb{E}_{\pi_F(s'|s)}[\log \pi_B(s|s') + V(s') - \log \pi_F(s'|s)]. \quad (58)$$

Therefore, according to the definition in (3), $V(s_h) = V^\dagger(s_h)$. Moreover,

$$V(s) = \mathbb{E}_{\pi_F(s'|s)}[Q(s, s') - \log \pi_F(s'|s)]$$

$$= \mathbb{E}_{\pi_F(s'|s)} \left[ \frac{\exp Q(s, s')}{\log \sum_{s'} \exp Q(s, s')} - \log \pi_F(s'|s) + \log \sum_{s'} \exp Q(s, s') \right]$$

$$= -D_{KL}\left(\pi_F(\cdot|s) \| \pi_Q(\cdot|s)\right) + \log \sum_{s'} \exp Q(s, s'), \quad \pi_Q(s'|s) := \frac{\exp Q(s, s')}{\sum_{s'} \exp Q(s, s')}. \quad (59)$$

This clarifies why $V$ ( and $V^\dagger$) serves as a critic: it evaluates how far the actor $\pi_F$ is from the local optimal policy $\pi_Q$. Here $\pi_Q$ is only locally optimal as $Q(s, s')$ may still deviate from the globally optimal point $Q^*(s, s')$. The divergence is then minimized w.r.t. $\pi_F$ using $V$, so that $V^\dagger$ of $\pi_F$ moves closer to the optimal one $V^*$. This can be achieved either by simply setting $\pi_F(\cdot|s)$ to $\pi_Q(\cdot|s)$ for all sampled states or by applying a policy-gradient update along sampled trajectories.

**Actor-Critic with Q-parameterization**  Under the actor-critic framework, we may instead parameterize $Q$ and $\pi_F$, and define:

$$V(s) := \mathbb{E}_{\pi_F(s'|s)}[Q(s, s') - \log \pi_F(s'|s)].$$

Then, the corresponding learning objective for $Q$ is written as follows:

$$\mathbb{E}_{P_F(s \to s')}[(\delta_S(s \to s'))^2], \quad \delta_S(s \to s') := \log \pi_B(s|s') + V(s') - Q(s, s'). \quad (60)$$

The expectation $\mathbb{E}_{P_F(s \to s')}[\ldots]$ above can be generalized into $\mathbb{E}_{P_\mathcal{D}(s \to s')}[\ldots]$, while the online expectation $\mathbb{E}_{\pi_F(s'|s)}[\ldots]$ in the definition of $V(s')$ is still maintained. When $Q$ achieves the optimal solution of the objective, we have:

$$V(s) = \mathbb{E}_{\pi_F(s'|s)}[\log \pi_B(s|s') + V(s') - \log \pi_F(s'|s)]. \quad (61)$$

Therefore, $V = V^\dagger$ according to the definition in (3).

---

**Algorithm 3** Basic Soft Actor-Critic Workflow

---

**Require:** $\pi_F, \pi_B, V$, batch size $K$, number of total iterations $N$
  **for** $n = 1, \ldots, N$ **do**
    $\mathcal{D} \leftarrow \{\tau^k | \tau^k \sim P_\mathcal{D}(\tau)\}_{k=1}^K$
    Based on $\mathcal{D}$, set $V(s)$ to $Q(s, s') - \log \pi_F(s'|s)$ for any $s(\neq s_f) \in \mathcal{D}$.
    Based on $\mathcal{D}$ and $V$, set $\pi_F(\cdot|s)$ to $\pi_Q(\cdot|s)$ for any $s(\neq s_f) \in \mathcal{D}$.
  **end for**
  **return** $\pi_F, V$

---

## A.7 SOFT ACTOR-CRITIC FOR GFLOWNET

The basic soft actor-critic algorithm tailored to GFlowNet training is presented in Alg. 3. When all states are visited through sampling in Alg. 3 and the online Bellman objective of $V$ reaches zero, meaning $\forall s \in \mathcal{S} : D_{KL}(\pi_F(\cdot|s), \pi_Q(\cdot|s)) = 0$, and $V = V^\dagger$, we show below that policy $\pi_F$ and $V$ will converge to optimal quantities $\pi_F^*$ and $V^*$. Starting at terminating states, we have:

$$
\begin{aligned}
V(x) &= -D_{KL}(\pi_F(\cdot|x)\|\pi_Q(\cdot|x)) + Q(x, s_f) \\
&= Q(x, s_f) := \log R(x)
\end{aligned}
\tag{62}
$$

$$
\begin{aligned}
Q(s_{H-1}, x) &:= \log \pi_B(s_{H-1}|x) + V(x) \\
&= \log \frac{F^*(s_{H-1} \to x)}{F^*(x)} + \log R(x) = \log F^*(s_{H-1} \to x),
\end{aligned}
\tag{63}
$$

where we use the definition of $\pi_B$. Next, we have:

$$
\begin{aligned}
V(s_{H-1}) &:= -D_{KL}\left(\pi_F(\cdot|s_{H-1})\|\pi_Q(\cdot|s_{H-1})\right) + \log \sum_x \exp Q(s_{H-1}, x) \\
&= \log \sum_x F^*(s_{H-1}, x) = \log F^*(s_{H-1}),
\end{aligned}
\tag{64}
$$

$$
\begin{aligned}
\pi_F(x|s_{H-1}) = \pi_Q(x|s_{H-1}) &:= \frac{\exp Q(s_{H-1}, x)}{\sum_x \exp Q(s_{H-1}, x)} \\
&= \frac{F^*(s_{H-1} \to x)}{F^*(s_{H-1})} := \pi_F^*(x|s_{H-1}).
\end{aligned}
\tag{65}
$$

Continuing this recursion, we will arrive at $V(s) = \log F^*(s)$, $Q(s \to s') = \log F^*(s \to s')$, and $\pi_F(s'|s) = \pi_F^*(s'|s)$ for all states and edges.

---

**Algorithm 4** Soft Actor-Critic Workflow with V-Parameterization

---

**Require:** $\pi_F(\cdot|\cdot;\theta), \pi_B(\cdot|\cdot), V(\cdot;\phi)$, batch size $K$, number of total iterations $N$
  **for** $n = 1, \ldots, N$ **do**
    $\mathcal{D} \leftarrow \{\tau^k | \tau^k \sim P_{\mathcal{D}}(\tau)\}_{k=1}^K$
    Based on $\mathcal{D}$, update $\phi$ by its gradients w.r.t. $\frac{1}{K} \sum_{\tau \in \mathcal{D}} \left[\sum_{s \in \tau} \mathbb{E}_{\pi_F(s'|s)} \left[\delta_V(s \to s'; \phi)^2\right]\right]$[4]
    Based on $\mathcal{D}$ and $V$, update $\theta$ by its gradients w.r.t. $\frac{1}{K} \sum_{\tau \in \mathcal{D}} \sum_{s \in \tau} D_{KL}(\pi_F(\cdot|s;\theta)\|\pi_Q(\cdot|s))$
  **end for**
  **return** $\pi_F(\cdot|\cdot;\theta), V(\cdot;\phi)$

---

**Algorithm 5** Soft Actor-Critic Workflow with Q-Parameterization

---

**Require:** $\pi_F(\cdot|\cdot;\theta), \pi_B(\cdot|\cdot), Q(\cdot,\cdot;\phi)$, batch size $K$, number of total iterations $N$
  **for** $n = 1, \ldots, N$ **do**
    $\mathcal{D} \leftarrow \{\tau^k | \tau^k \sim P_{\mathcal{D}}(\tau)\}_{k=1}^K$
    $\forall s \in \mathcal{D} : V(s) \leftarrow \mathbb{E}_{\pi_F(s'|s)}[Q(s, s') - \log \pi_F(s'|s)]$
    Based on $\mathcal{D}$, update $\phi$ by its gradients w.r.t. $\frac{1}{K} \sum_{\tau \in \mathcal{D}} \left[\sum_{(s \to s') \in \tau} \left[\delta_S(s \to s'; \phi)^2\right]\right]$.
    Based on $\mathcal{D}$ and $Q$, update $\theta$ by its gradients w.r.t. $\frac{1}{K} \sum_{\tau \in \mathcal{D}} \sum_{s \in \tau} D_{KL}(\pi_F(\cdot|s;\theta)\|\pi_Q(\cdot|s))$
  **end for**
  **return** $\pi_F(\cdot|\cdot;\theta), Q(\cdot,\cdot;\phi)$

---

For practical implementation, we may modify Alg. 3 into Alg. 4 or Alg. 5 for Q-parameterization, where the basic optimization operator is replaced with gradient-based updates. Compared to Alg. 4, Alg. 1 operates strictly in an on-policy manner, using online trajectories sampled from $P_F(\tau)$. This design enables policy optimization based on the estimated gradients of global divergence

---

[4]We can further approximate the expectation $\mathbb{E}_{\pi_F(s'|s)}[\ldots]$ by $\frac{1}{L} \sum_{s' \in \mathcal{D}(s)}[\ldots]$, where $\mathcal{D}(s) = \{(s')^k | (s')^k \sim \pi_F(s'|s)\}_{l=1}^L$.

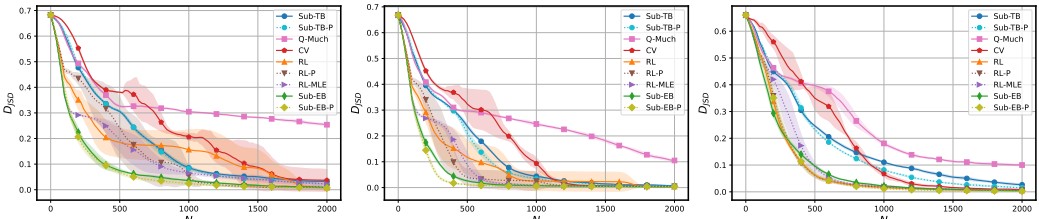

Figure 4: Plots of the means and standard deviations (represented by the shaded area) of $D_{\mathrm{JSD}}$ for different training methods on the $256 \times 256$ (left) $128 \times 128$ (middle) and $64 \times 64 \times 64$ (right) grids, based on five randomly started runs for each method.

$D_{\mathrm{KL}}(P_F(\tau;\theta)\|P_B(\tau))$ through policy-gradient theory, rather than the gradients induced by local divergence $D_{\mathrm{KL}}(\pi_F(\cdot|s;\theta)\|\pi_Q(\cdot|s))$. Meanwhile, it generalizes $\delta_V(s \to s';\phi)$ to $\delta_V(\tau_{i:j};\phi)$. To relax the on-policy restriction of standard policy-gradient methods, Alg. 2 is introduced to enable the use of an offline sampler $P_{\mathcal{D}}$.

## B EXPERIMENTAL SETTINGS AND RESULTS

**Hyperparameters**  For both the original policy-based method and the proposed one with the Sub-TB objective (RL and Sub-EB), we set the hyperparameter $\gamma$ to 0.99 based on the ablation study results reported in Niu et al. (2024). For the data-collection policy $\pi_{\mathcal{D}}$ of Sub-TB, the hyperparameter $\alpha$ starts at $1.0$ and decays exponentially at a rate of $0.99$, where the decay rate is also determined based on the results of the ablation study in Niu et al. (2024). In the Sub-TB objective, the hyperparameter $\lambda$ is set to 0.9 following the ablation study by Madan et al. (2023). For the Sub-EB objective, $\lambda$ is set to be 0.9 selected from $\{0.1, 0.2, \dots, 0.9, 0.99\}$ based on the ablation study results shown in Fig. 6. All experiments are conducted on the Grace cluster, one of the main Texas A&M High Performance Research Computing systems. Each node is equipped with an Intel Xeon Cascade Lake CPU. For our experiments, we requested a single node, 20 GB of RAM, and no GPU was used. The running times of the considered methods across experiments are reported in Appendix B.5.

**Optimization**  The Adam optimizer is used for optimization. The sample batch size is set to 128 for each optimization iteration following Niu et al. (2024). The learning rates of $\pi_F(\cdot|\cdot;\theta)$ and $\log F(\cdot;\theta)$ are equal to $1 \times 10^{-3}$, which is selected from $\{5 \times 10^{-3}, 1 \times 10^{-3}, 5 \times 10^{-4}, 1 \times 10^{-4}\}$ based on the performance of Sub-TB on the $256 \times 256$ grid. The learning rate of $V(\cdot;\phi)$ is set to $5 \times 10^{-3}$, which is selected from $\{10^{-2}, 5 \times 10^{-3}, 10^{-3}, 5 \times 10^{-4}, 10^{-4}\}$ based on the performance of RL on the $256 \times 256$ grid. In all experiments, each training method is run five times, initialized from five different random seeds.

**Model architecture**  The forward policy $\pi_F(\cdot|\cdot;\theta)$ and evaluation function $V(\cdot;\phi)$ are each parameterized by a neural network with four hidden layers, with a hidden dimension of 256 in each layer. The backward policy $\pi_B(\cdot|\cdot)$ is a uniform distribution over valid transitions (edges). In hypergrid, sequence design, and molecular graph design experiments, coordinate tuples, integer sequences, and integer matrices are transformed using K-hot encoding before entering the neural networks. In BN structure learning, adjacency matrices are used directly as input into neural networks without encoding.

**Peformance metrics**  The first one is the total variation $D_{\mathrm{TV}}$ between $P_F(x)$ and $P^*(x)$, which is defined as: $D_{\mathrm{TV}}(P_F(x), P^*(x)) = \frac{1}{2}\sum_{x \in \mathcal{X}}|P_F(x) - P^*(x)|$. An alternative performance metric adopted in literature is the average $l_1$-distance, which is defined as $|\mathcal{X}|^{-1}\sum_x |P^*(x) - P_F(x)|$. The reason that we use $D_{\mathrm{TV}}$ instead is as follows. The design space $|\mathcal{X}|$ is usually large ($> 10^4$) and $\sum_x |P^*(x) - P_F(x)| \leq 2$, resulting in the average $l_1$-distance being heavily scaled down by $|\mathcal{X}|$. We also evaluate different methods using the Jensen–Shannon divergence $D_{\mathrm{JSD}}$ as the second metric, which can be written as: $D_{\mathrm{JSD}}(P_F(x), P^*(x)) = \frac{1}{2}D_{\mathrm{KL}}(P_F(x), P_M(x)) + \frac{1}{2}D_{\mathrm{KL}}(P^*(x), P_M(x))$, where $P_M := \frac{1}{2}(P_F + P^*)$.

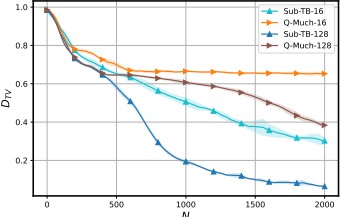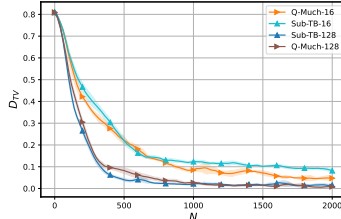

Figure 5: Plots of the means and standard deviations (represented by the shaded area) of $D_{\text{TV}}$ (right) for different training methods on the $128 \times 128$ (left) $20 \times 20$ (right) grids, based on five randomly started runs for Sub-TB-16 Sub-TB-128, Q-much-16 and Q-much 128. Here, "16" and "128" denote the training batch sizes.

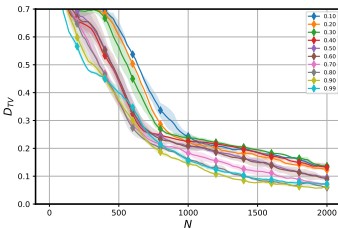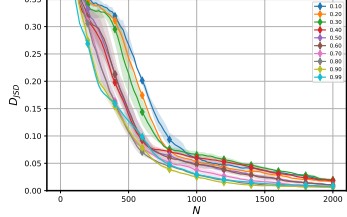

Figure 6: Plots of the means and standard deviations (represented by the shaded area) of $D_{\text{TV}}$ (left) and $D_{\text{JSD}}$ (right) for Sub-EB runs on the $128 \times 128$ grid. The hyperparameter $\lambda$ of Sub-EB ranges from 0.1 to 0.99. The results are based on five Sub-EB runs for each setup.

## B.1 HYPERGRIDS

The generative process of hypergrid experiments is defined as follows. For a grid with height $H$ and width $D$, the state space excluding the final state, $\mathcal{S}\backslash\{s_f\}$, consists of all $D$-dimensional coordinate vectors of the form $\{s = ([s]_1, \dots, [s]_d, \dots, [s]_D) \mid [s]_d \in \{0, \dots, H-1\}\}$. The generating process begins at the initial state $s_0 = (0, \dots, 0)$ and ends in the final state $s_f$, which we denote as $(-1, \dots, -1)$. From any state $s \in \mathcal{S}\backslash\{s_f\}$, there are $D+1$ valid transitions (edges): (1) for each $d \in \{0, \dots, D\}$, the $d$-th coordinate can be incremented by one, leading to a new state $s' = ([s]_1, \dots, [s]_d + 1, \dots, [s]_D)$; (2) if $s \in \mathcal{X}$, the process can be stopped by taking transition $(s \to s_f)$, returning $s$ as the terminating coordinate tuple $x$. By the definitions above, the GFlowNet DAG $\mathcal{G}$ is **not** *graded*, meaning every state (excluding $s_f$) can be returned as a terminating state. The reward function associated with terminating states is defined as:

$$R(x) = R_0 + R_1 \prod_{d=1}^{D} \mathbb{I}\left[\left|\frac{[s]_d}{N-1} - 0.5\right| \in (0.25, 0.5]\right] + R_2 \prod_{d=1}^{D} \mathbb{I}\left[\left|\frac{[s]_d}{N-1} - 0.5\right| \in (0.3, 0.4]\right],$$

where $R_0 = 10^{-2}$, $R_1 = 0.5$ and $R_2 = 2$ in our experiments.

**Additional experiment results**   As discussed in our Appendix A.6, Q-much may prioritize exploitation compared to the (Sub-)trajectory level methods and perform less effectively on challenging tasks with well-separated modes. To verify this, we conducted hypergrid tasks with batch sizes of 16 and 128 (our default setting) on both a $20 \times 20$ grid and a $128 \times 128$ grid. The rationale is as follows. For a grid with height $N$ and dimension $D$, the minimum distance between reward modes depends only on the height $N$. With moderate $D$ and small $N$, even a uniform policy can still reach the modes with considerable probability. Therefore, $N$ plays a more significant role in determining task difficulty. The experimental results are presented in Fig. 5. On the $20 \times 20$ grid with batch size 16, Q-much performs better than Sub-TB. However, this advantage disappears when using batch size 128. For $128 \times 128$ grid with batch size 128, the performance of Q-much is considerably worse than Sub-TB. When the batch size is reduced to 16, neither method converges to a satisfactory solution; nevertheless, Q-Much still underperforms Sub-TB.

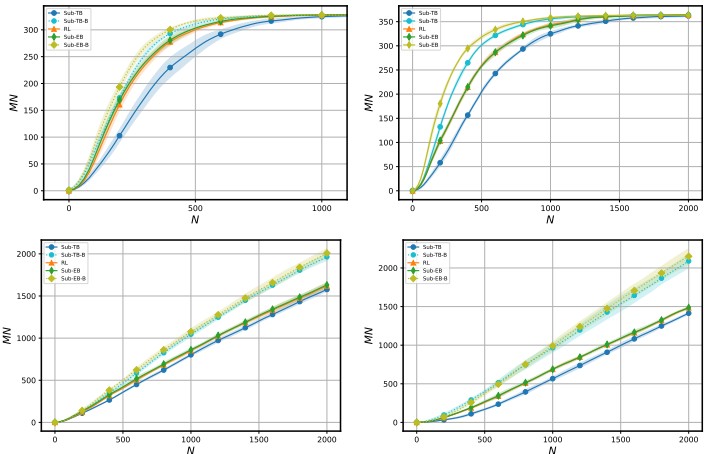

Figure 7: Plots of the mean and standard deviation values (represented by the shaded area) of MN for the SIX6 (top left), QM9 (top right), PHO4 (bottom left), and sEH (bottom right) dataset, based on five randomly started runs for each method.

**Ablation study on** $\lambda$    For the $128 \times 128$ grid, we conduct an ablation study on the hyperparameter $\lambda$ of the Sub-EB weights $w_{i,j} (= \lambda^{j-i} / \sum_{i<j \in [H+1]} \lambda^{j-i})$ to investigate its effect on policy-based GFlowNet training. We run Sub-EB methods with $\lambda$ equal to $0.1, 0.2, \ldots, 0.90$ and $0.99$. The experimental results are depicted in Fig. 6. It can be observed that the Sub-EB method with $\lambda = 0.9$ achieves the best performance. Although convergence rates and final performances differ, Sub-EB methods under all configurations exhibit good stability, demonstrating that the proposed Sub-EB objective enables reliable learning of the evaluation function $V$.

## B.2 SEQUENCE DESIGN

The generative process of this set of experiments is defined as follows. The state space excluding the final state $\mathcal{S}\backslash\{s_0, s_f\}$ is equal to $\bigcup_{t=1}^{H}\{0, \ldots, N-1\}^t$, where each state is a sequence composed of integers ranging from 0 to $N-1$. The set $\{0, \ldots, N-1\}$ corresponds to the $N$ types of building blocks. The process begins at the initial state $s_0 = \emptyset$, that is, the empty sequence. For any intermediate state $s_h \in \mathcal{S}_h$, it contains $h$ elements drawn from $\{0, \ldots, N-1\}$. There are $N \times 2$ possible transitions for $s_t$ with $t < D$, corresponding to either appending or prepending one element from $\{0, \ldots, N-1\}$ to the current sequence. The procedure continues until sequences reach length $D$, yielding terminal states in $\mathcal{X} = \{0, \ldots, N-1\}^D$, and transitioning to $s_f := (N, \ldots, N)$. According to the definition of the generative process, the GFlowNet DAG $\mathcal{G}$ is *graded* and $\mathcal{S}_D = \mathcal{X}$. In practice, each state $s_t$ is represented as a sequence of fixed length $D$, where the first $t$ elements are integers from $\{0, \ldots, N-1\}$, and the others are equal to $-1$. In particular, the initial state $s_0$ is represented as $(-1, \ldots, -1)$. We use nucleotide sequence datasets (SIX6 and PHO4) and molecular sequence datasets (QM9 and sEH ) from Shen et al. (2023). Rewards are defined as the exponents of raw scores from the datasets, $R(x) = \text{score}^\beta(x)$. The hyperparameter $\beta$ is set to 3, 5, 3 and 6 for SIX6, QM9, PHO4 and sEH, respectively, and the rewards are normalized to $[10^{-3}, 10]$, $[10^{-3}, 10]$, $[0, 10]$ and $[10^{-3}, 10]$, respectively. In this experimental setting, we consider an additional metric introduced by Shen et al. (2023). It is Mode Accuracy (MA) of $P_F(x)$ w.r.t. $P^*(x)$, defined as:

$$\text{MA}(P_F(x), P^*(x)) = \min\left(\frac{\mathbb{E}_{P_F(x)}[R(x)]}{\mathbb{E}_{P^*(x)}[R(x)]}, 1\right). \tag{66}$$

We use dynamic programming (Malkin et al., 2022a) to compute $P_F(x)$ and the exact MA, $D_{\text{TV}}$ and $D_{\text{JSD}}$ between $P_F(x)$ and $P^*(x)$.

In this set of experiments, we focus not only on distribution modeling but also on mode discovery, where the goal is to uncover high-reward terminating states. In addition to RL, Sub-EB, and Sub-TB methods, we consider Sub-TB-B and Sub-EB-B, which explicitly promote the exploration of high-reward states during trajectory sampling.

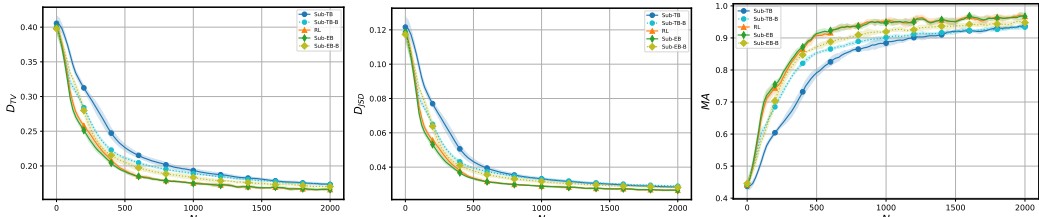

Figure 8: Plots of the mean and standard deviation values (represented by the shaded area) of $D_{\text{TV}}$ (left), $D_{\text{JSD}}$ (middle), and MA (right) for the SIX6 dataset, based on five randomly started runs for each method.

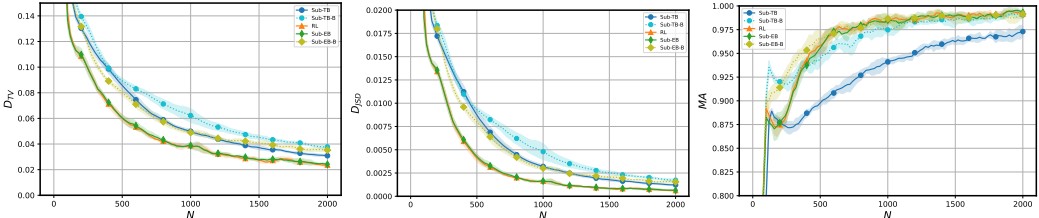

Figure 9: Plots of the mean and standard deviation values (represented by the shaded area) of $D_{\text{TV}}$ (left), $D_{\text{JSD}}$ (middle), and MA (right) for the QM9 dataset, based on five randomly started runs for each method.

**Distribution modeling Results** We compare different training methods by their performance measured by MA, $D_{\text{TV}}$ and $D_{\text{JSD}}$, as shown in Figs. 8, 9, 10, and 11 for SIX, QM9, PHO4, and sEH datasets respectively. It can be seen that Sub-EB performs slightly better than RL. This can be ascribed to the sufficient stability of RL in these experiments, rendering the advantages brought by the Sub-TB objective less obvious. Nevertheless, Sub-EB and RL outperform Sub-TB in terms of both convergence rate and final performance. For the offline variants, it can be observed that Sub-EB-B performs slightly better than Sub-TB-B, but performs worse than Sub-EB for all datasets except PHO4. The experiment results validate that the proposed Sub-EB objective can be applied to the policy-based training workflow for learning the valuation function $V$ effectively.

**Mode discovery Results** We measure performance using Mode Number (MN), defined as the number of unique high-reward terminating states discovered during training. A terminating state is regarded as highly rewarded if its reward falls within the top $0.5\%$ for the SIX6, QM9, and PHO4 datasets, and the top $0.01\%$ for the sEH dataset. As depicted in Fig. 7, Sub-EB-B and Sub-TB-B can find a fixed number of unique modes faster and discover more unique modes within a fixed number of optimization iterations, despite a decline in distribution modeling performance on all datasets except PHO4. These results validate our offline policy-based training workflow and support our claim that the proposed Sub-EB objective enables the integration of offline sampling techniques.

Combined with the mode discovery results in Fig. 7, this indicates that offline techniques that encourage high-reward terminating states are helpful for mode discovery, while they may hinder accurate distribution modeling.

### B.3 BN STRUCTURE LEARNING

A Bayesian Network is a probabilistic model, representing the joint distribution of $N$ random variables, whose factorization is determined by the network structure $x$, which is a DAG graph. Accordingly, the distribution can be written as $P(y_1, \ldots, y_N) = \prod_{n=1}^{N} P(y_n | Pa_x(y_n))$, where $Pa_x(y_n)$ denotes the parent nodes of $y_n$ in graph $x$. Since any graph structure can be encoded as an adjacency matrix, the state space excluding the final state is defined as $\mathcal{S} \setminus \{s_f\} := \left\{ s | \mathcal{C}(s) = 0, s \in \{0, 1\}^{N \times N} \right\}$ where $\mathcal{C}$ corresponds to the acyclic graph constraint introduced by Deleu et al. (2022). It should be noted that each BN DAG $x \in \mathcal{X}$ corresponds to a state $s \in \mathcal{S}$ in

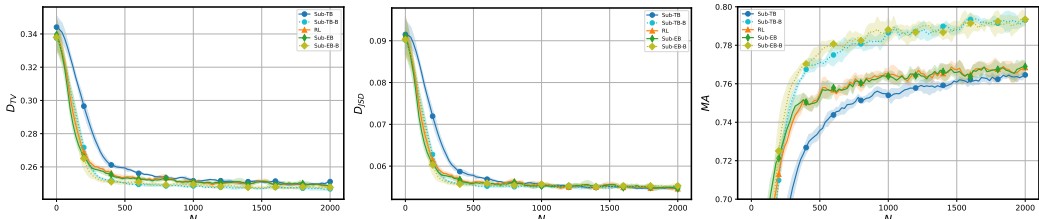

Figure 10: Plots of the mean and standard deviation values (represented by the shaded area) of $D_{\text{TV}}$ (left), $D_{\text{JSD}}$ (middle), and MA (right) for the PHO4 dataset, based on five randomly started runs for each method.

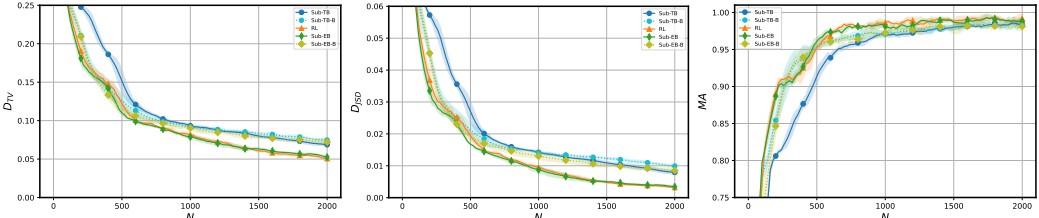

Figure 11: Plots of the mean and standard deviation values (represented by the shaded area) of $D_{\text{TV}}$ (left), $D_{\text{JSD}}$ (middle), and MA (right) for the sEH dataset, based on five randomly started runs for each method.

the GFlowNet DAG $\mathcal{G}$. The generative process begins from the initial state $s_0 = 0^{N \times N}$, representing a graph without edges, and ends at $s_f := -1^{N \times N}$. For any $s \in \mathcal{S} \backslash s_f$, a transition $(s \rightarrow s')$ corresponds to adding an edge by flipping a zero entry in the adjacency matrix to one, provided that the resulting state $s'$ remains acyclic. Alternatively, the generative process is stopped by transitioning to $s_f$ and returning $s$ as the terminating graph structure. According to the definition of the generative process, $\mathcal{G}$ is **not** *graded* for this set of experiments. Given an observed sample set $\mathcal{D}_y$ of $y_{1:N}$, the goal of structure learning is to approximate the posterior distribution $P(x|\mathcal{D}_y) \propto P(\mathcal{D}_y|x)P(x)$. In the absence of prior knowledge about $x$, the prior distribution $P(x)$ is often assumed to be uniform, reducing the task to maximizing the likelihood, $P(\mathcal{D}_y|x)(\propto P(x|\mathcal{D}_y))$.

The ground-truth graph structure $x^*$ and the corresponding dataset $\mathcal{D}_y$ of size $|\mathcal{D}_y| = 10^3$ are simulated from Erdős–Rényi model (Deleu et al., 2022). We use BGe score (Kuipers et al., 2014) to assess generated graph structures, and define the reward $R(x) = \left(\frac{\Delta(x;\mathcal{D}_y)}{C}\right)^{\beta}$, where $\Delta(x;\mathcal{D}_y) = \text{BGe}(x;\mathcal{D}_y) - \text{BGe}(s_0;\mathcal{D}_y)$, $\beta = 10$ sharpens the reward function toward high-score structures, and $C := \Delta(x^*;\mathcal{D}_y)$ normalizes the reward so that $R(x^*) = 1$. The exact number of DAGs on $n$ nodes, denoted as $a(n)$ satisfies (Robinson, 2006):

$$a(n) = \sum_{k=1}^{n}(-1)^{k+1}\binom{n}{k} 2^{k(n-k)} a(n-k), \tag{67}$$

with $a(0) = 1$. For the ease of accessing distribution modeling performance, Malkin et al. (2022b) set the number of nodes to 5, resulting in about $2.92 \times 10^4$ possible DAGs. In this setup, the ground-truth DAG contains 5 edges. In addition to this small-scale case, we also consider two much larger cases with 10 and 15 nodes, corresponding to about $4.18 \times 10^{18}$ and $2.38 \times 10^{35}$ possible DAGs, respectively. We set the ground-truth DAGs to contain 10 and 15 edges in the two respective cases.

**Additional experimental results**  For the small-scale case, we use dynamic programming (Malkin et al., 2022a) to explicitly compute $P_F(x)$ for learned $\pi_F$, and compute the exact $D_{\text{TV}}$ and $D_{\text{JSD}}$ between $P_F(x)$ and $P^*(x)$. In Fig. 13, the mean and standard deviation values of $D_{\text{TV}}$ and $D_{\text{JSD}}$ are plotted for five runs of Sub-TB, RL, Sub-EB, and Q-Much, respectively. It can be seen that Sub-EB performs better than the RL, Sub-TB and Q-Much. These results confirm our conclusion that the Sub-EB objective enables more reliable learning of the evaluation function $V$ compared to the $\lambda$-TD

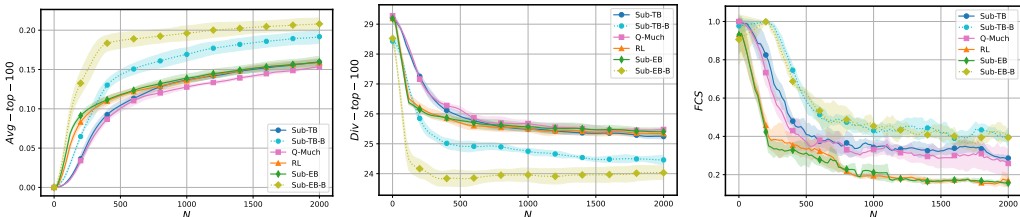

Figure 12: Plots of the mean and standard deviation values (represented by the shaded area) of average reward (left), diversity (middle) and FCS (right) of the top 100 unique candidate graphs over 15 nodes, based on five randomly started runs for each method.

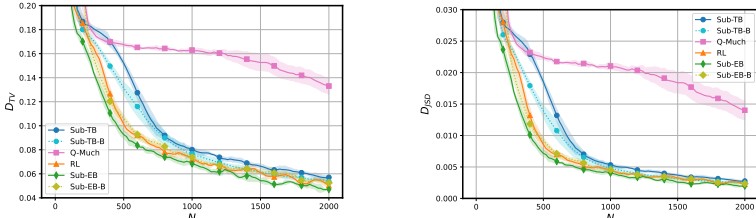

Figure 13: Plots of the mean and standard deviation values (represented by the shaded area) of $D_{\mathrm{TV}}$ (left) and $D_{\mathrm{JSD}}$ (right) for the BN learning structure learning task over 5 nodes, based on five randomly started runs for each method.

objective, thereby improving the performance of policy-based methods. We also include results for Sub-EB-B and Sub-TB-B. It can be seen that they both perform worse than their original versions. These results further support our conclusion that explicitly encouraging exploration of high-reward regions may not be beneficial for distributional modeling performance.

## B.4 MOLECULAR GRAPH DESIGN

Each molecule (graph) is a composition of at most 8 building blocks, selected from $N = 105$ predefined molecular substructures provided by Bengio et al. (2021). Each block (graph node) has a list that contains at most $M$ available atom indices for forming bonds (graph edges) with other blocks. Each block's list contains exactly one target atom and 0 to $M - 1$ source atoms. A bond is formed by connecting a source atom in one block to a target atom in another. A graphical illustration of 4 exemplary blocks is shown in Fig. 14. Following Bengio et al. (2021), we restrict the maximum number of blocks in a molecular graph to be $H = 8$. Each state $s \in \mathcal{S}$ is represented as a $H \times 2$ matrix. The generative process begins from the initial state $s_0 = -1^{H \times 2}$, representing an empty graph, and ends at $s_f = 105^{H \times 2}$. For any state $s_h \in \mathcal{S}_h$:

1. The first column contains $h$ integers drawn from $\{0, \ldots, N - 1\}$, representing the indices of the building blocks present in the molecule.

2. The second column contains $h - 1$ integers drawn from $\{0, \ldots, H \times M - 1\}$, representing the indices of bonding sites that encode the connectivity structures. Specifically, if an integer is equal to $(i - 1) \times H + j$, this indicates that the target atom of the newly added block will be connected to the $j$-th source atom of the $i$-th block.

For any $s \in \mathcal{S} \setminus (s_f, s_0)$, a transition $(s \to s')$ corresponds to a two-level action:

1. Selecting a building block to add, represented by an integer in $\{0, \ldots, N - 1\}$.

2. Selecting a bonding site represented by an integer $k \in \{0, \ldots, H \times M - 1\}$.

The generative process stops when we make the transition $(x \to s_f)$, when no available source atoms remain for forming bonds, or when the state reaches the limit of $H$ blocks. According to the definition of the generative process, the GFlowNet DAG $\mathcal{G}$ is **not** *graded*, meaning that any state

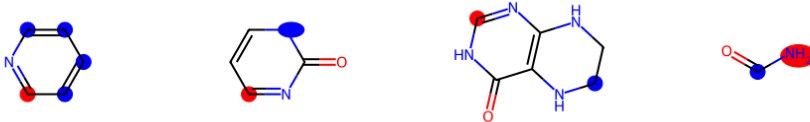

Figure 14: Four exemplary building blocks. Red and blue dots indicate the available atoms for bonding, where each bonding edge originates from a red dot and terminates at a blue dot.

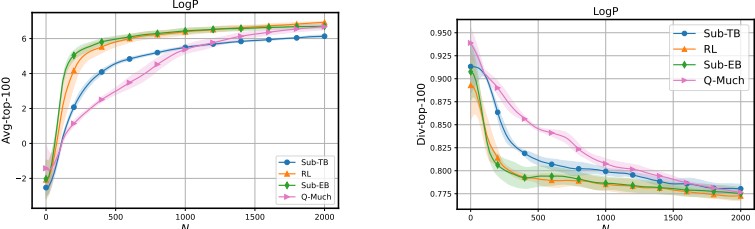

Figure 15: Plots of the mean and standard deviation values of average reward (left) and diversity (right, Tanimoto diversity of molecular Morgan fingerprints) of top 100 unique molecules for the LogP task, based on five randomly started runs for each method.

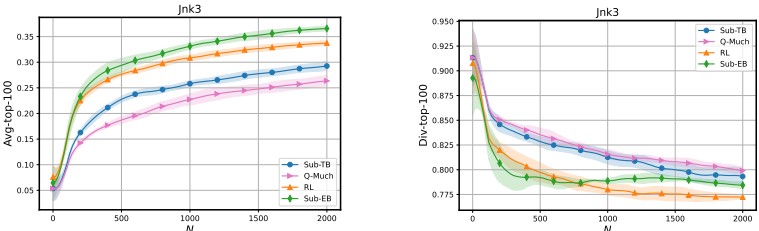

Figure 16: Plots of the mean and standard deviation values of average reward (left) and diversity (right, Tanimoto diversity of molecular Morgan fingerprints) of top 100 unique molecules for the JNK3 task, based on five randomly started runs for each method.

except $s_f$ can be a terminating state $x \in \mathcal{X}$. Since there may be multiple types of bonds between a given pair of blocks, the size of the terminating state space is greater than $\frac{105!}{(105-8)!} \approx 10^{16}$. The Octanol–Water Partition Coefficient (LogP) and c-Jun N-terminal Kinase 3 (JNK3) scores provided in the *pyTDC* package are directly used as the reward functions $R(x)$. Since actions in this generative process involve two levels, we use separate neural networks to represent the policy for each action component. The log-probability of an action is computed as the sum of the log-probabilities produced by the corresponding networks for the two action levels. Finally, following Bengio et al. (2021), we choose parameterized $\pi_B$ and use a batch size of 4 to emulate scenarios where querying the real-world molecule oracle is computationally expensive.

The experimental results are presented in Figs. 15 and 16. We report the average reward and diversity of the top 100 unique graphs discovered during the training process. The diversity of a set of molecules is computed as the average pairwise dissimilarity based on Tanimoto distance of Morgan fingerprints via the *pyTDC* package. For the LogP task, RL, Sub-EB, and Q-Much achieve similar average rewards, all outperforming Sub-TB. Moreover, RL and Sub-EB converge much faster than Sub-TB and Q-Much. All four methods exhibit comparable diversity. For the JNK3 task, Sub-EB achieves the highest average reward and demonstrates the fastest convergence among all methods. Its diversity is comparable to that of Sub-TB and higher than that of RL. Although Q-Much achieves the highest diversity, its average reward is almost the lowest. Overall, these results indicate that

Sub-EB achieves the best performance in terms of average reward and convergence rate while maintaining reasonable diversity, confirming its effectiveness for large-scale tasks.

## B.5 RUNTIME REPORT

| Method | Mean | Std | Method | Mean | Std | Method | Mean | Std |
|--------|------|-----|--------|------|-----|--------|------|-----|
| Sub-TB | 1h 08m | 1.09 | Sub-TB | 0h 52m | 0.27 | Sub-TB | 1h 59m | 1.53 |
| Sub-TB-P | 1h 14m | 10.36 | Sub-TB-P | 0h 58m | 7.92 | Sub-TB-P | 2h 19m | 4.40 |
| **RL** | **1h 03m** | 0.92 | RL | **0h 44m** | 1.43 | RL | **1h 27m** | 7.37 |
| RL-P | 1h 27m | 0.57 | RL-P | 1h 06m | 2.21 | RL-P | 2h 03m | 10.68 |
| RL-MLE | 1h 09m | 1.78 | RL-MLE | 0h 58m | 3.27 | RL-MLE | 2h 28m | 9.58 |
| Sub-EB | 1h 17m | 2.31 | Sub-EB | 0h 54m | 2.66 | Sub-EB | 2h 04m | 4.11 |
| Sub-EB-P | 1h 17m | 1.49 | Sub-EB-P | 1h 11m | 0.82 | Sub-EB-P | 2h 41m | 15.46 |

Table 1: The mean and standard deviation (std) of the total runtimes for each method on the $64 \times 64 \times 64$ (left), $128 \times 128 \times 128$ (middle), and $256 \times 256 \times 256$ (right) grids. Here, 'h' denotes hours and 'm' denotes minutes, and all std values are reported in minutes.

| Method | Mean | Std | Method | Mean | Std |
|--------|------|-----|--------|------|-----|
| Sub-TB | 0h 05m | 0.21 | Sub-TB | **0h 04m** | 1.10 |
| Sub-TB-B | 0h 16m | 0.67 | Sub-TB-B | 0h 12m | 0.57 |
| RL | **0h 04m** | 0.06 | RL | **0h 04m** | 0.12 |
| Sub-EB | **0h 04m** | 0.34 | Sub-EB | **0h 04m** | 0.06 |
| Sub-EB-B | 0h 17m | 1.63 | Sub-EB-B | 0h 13m | 0.71 |

Table 2: The mean and standard deviation (std) of the total runtimes for each method on the SIX6 and QM9 datasets. Here, 'h' denotes hours and 'm' denotes minutes, and all std values are reported in minutes.

| Method | Mean Time | Std (min) | Method | Mean Time | Std (min) |
|--------|-----------|-----------|--------|-----------|-----------|
| Sub-TB | 0h 19m | 2.05 | **Sub-TB** | **3h 30m** | 10.23 |
| Sub-TB-B | 0h 27m | 0.42 | Sub-TB-B | 3h 46m | 11.59 |
| **RL** | **0h 18m** | 0.10 | RL | 3h 42m | 13.78 |
| **Sub-EB** | **0h 18m** | 0.15 | Sub-EB | 3h 41m | 13.97 |
| Sub-EB-B | 0h 30m | 1.09 | Sub-EB-B | 3h 58m | 7.57 |

Table 3: The mean and standard deviation (std) of the total runtimes for each method on the PHO4 and sEH datasets. Here, 'h' denotes hours and 'm' denotes minutes, and all std values are reported in minutes.

| Method | Mean | Std | Method | Mean | Std | Method | Mean | Std |
|---|---|---|---|---|---|---|---|---|
| Sub-TB | 0h 57m | 1.11 | **Sub-TB** | **0h 10m** | 0.80 | Sub-TB | 0h 16m | 1.28 |
| Sub-TB-B | 1h 23m | 4.27 | Sub-TB-B | 0h 36m | 1.82 | Sub-TB-B | 0h 42m | 3.82 |
| RL | 0h 56m | 0.46 | RL | 0h 11m | 3.58 | **RL** | **0h 12m** | 1.79 |
| **Sub-EB** | **0h 49m** | 0.87 | **Sub-EB** | **0h 10m** | 0.91 | Sub-EB | 0h 14m | 0.28 |
| Sub-EB-B | 1h 07m | 1.43 | Sub-EB-B | 0h 31m | 2.27 | Sub-EB-B | 0h 39m | 1.13 |
| Q-Much | 0h 59m | 3.57 | Q-Much | 0h 12m | 1.34 | Q-Much | 0h 19 m | 0.62 |

Table 4: The mean and standard deviation (std) of the total runtimes for each method on the 5-node, 10-node and 15-node BN tasks. Here, 'h' denotes hours and 'm' denotes minutes, and all std values are reported in minutes. The runtimes for the 5-node cases are the largest due to the explicit computation of $D_{TV}$ for performance comparison during training.

| Method | Mean Time | Std (min) | Method | Mean Time | Std (min) |
|---|---|---|---|---|---|
| **RL** | **0h 9m** | 0.36 | RL | 0h 12m | 1.33 |
| RLEval | 0h 13m | 0.17 | **RLEval** | **0h 9m** | 0.08 |
| Sub-TB | 0h 13m | 0.93 | **Sub-TB** | **0h 9m** | 0.29 |
| Q-Much | 0h 19 m | 3.50 | Q-Much | 0 h 10 m | 0.26 |

Table 5: The mean and standard deviation (std) of the total runtimes for each method on the LogP and JNK3 tasks. Here, 'h' denotes hours and 'm' denotes minutes, and all std values are reported in minutes.

