# OpenReview forum: "Evaluating GFlowNet from partial episodes for stable and flexible policy-based training"
_ICLR.cc/2026/Conference — ICLR 2026 Poster_

### Official Review · Reviewer_C8bA · 2025-10-23

**Soundness:** 3
**Presentation:** 2
**Contribution:** 2
**Rating:** 2
**Confidence:** 4

**Summary:**

The focus of this paper is policy gradient based learning of GFlowNets. The paper presents a sub-trajectory based loss for learning the evaluation function of a GFlowNet forward policy. The evaluation function is also theoretically connected to flow functions of GFlowNets. The learnable evaluation function is then utilized in the policy gradient algorithm to train the GFlowNet forward policy. The algorithm also allows for learning the backward policy. The approach is experimentally evaluated on hypergrids, biological sequence environments and bayesian structure learning tasks.

**Strengths:**

The paper studies an interesting and important direction of applying policy gradient methods to GFlowNets. I also found the presented connections between evaluation functions and flows to be interesting and insightful (although not entirely new, see Weaknesses).

**Weaknesses:**

The main weakness of this work in my opinion is its limited novelty. The definition of evaluation function (Equation 3) coincides with the negative entropy-regularized value in the construction that frames GFlowNet learning as an entropy-regularized RL problem from [1]. The connections between entropy-regularized values and flows were also previosly presented in [1] (see Theorem 1 and Proposition 1 of [1]). Theorem 3.1 in the present paper can be seen as a generalization of results connecting flows and values from Theorem 1 and Proposition 1 of [1], however, from what I understand, this can be shown by a simple modification of the proof of Proposition 1 from [1]. In addition, such contributions and their novelty must be correctly framed with respect to the previous literature, while the work [1] is neither discussed nor cited in this paper.

The proposed SubEB objective (Equation 8) exactly coincides with SubTB loss [2], which is already well-known and widely studied in the GFlowNet literature. The only difference is that this loss is used to learn only flows/values in the proposed algorithm for the current policy, while the policy uses a different expression for gradient update. The policy gradient training of GFlowNets itself is also not new and was previously studied in [3]. From what I understand, the difference of the proposed algorithm from the algorithm in [3] is a different way to evaluate states/trajectories (by learning the evaluation function via SubEB/SubTB).

The writing in this work seems unclear in a number of places. In line 46 it is stated that the direct optimization of distributional divergences between forward and backward trajectory distribution is not possible since the normalizing constant is unkown. However, for example, the normalizing constant does not affect the optimization of KL divergence, and directly optimizing it is possble (see [4]). Moreover, in line 345 the authors themselves acknowledge that the are works that directly optimize this KL divergence. I also found the writing regarding backward policy optimization in Section 3.2 and Section 3.3 to be a bit jumbled up and hard to get through (e.g. $W$ is used in the text prior to its definition), so the clarity could be improved here in my opinion. However, since this is mostly a retelling of the results from the prior work [3], I think this is a minor point. In addition, Algorithm 1 does not specify how the gradients wrt forward policy are calculated. Does it use the expression from Equation 4?

Another minor point, but since the paper partially focuses on the backward policy optimization, it would be good to cite and discuss prior works on this topic in the text [5, 6].

Next, the presented empirical results on sequence design tasks, which are the only non-synthetic experiments in the main text, are unconvincing. The improvements are very marginal in comparison to SubTB in Figure 3 in the case when the local search is used. When it is not used, the improvements are notable only in the first two plots.

In addition, all the experiments in the main text are on environments of a very small scale: 256 × 256 grid (65 thousand terminal states), 64 x 64 x 64 grid (262 thousand terminal states), SIX6 (65 thousand terminal states), QM9 (58 thousand terminal states), PHO4 (1 million terminal states), sEH (34 million terminal states). All of these environments can be considered toy since the space of objects we work with is small enough that it is possible to iterate over all objects in reasonable time, thus the task of training a GFlowNet policy for sampling is artificial and redundant. There are experiments on a larger-scale structure learning task in Appendix, so I would suggest moving them to the main text and putting more emphasis on them.

Overall, I found this to be an interesting paper, but the presented contributions in my opinion are not enough to recommend acceptance to a conference of this level.

References:\
[1] Tiapkin et al. Generative Flow Networks as Entropy-Regularized RL. AISTATS 2024\
[2] Madan et al. Learning GFlowNets from partial episodes for improved convergence and stability. ICML 2023\
[3] Niu et al. GFlowNet Training by Policy Gradients. ICML 2024\
[4] Malkin et al. GFlowNets and variational inference. ICLR 2023\
[5] Jang et al. Pessimistic Backward Policy for GFlowNets. NeurIPS 2024\
[6] Gritsaev et al. Optimizing Backward Policies in GFlowNets via Trajectory Likelihood Maximization. ICLR 2025

**Questions:**

0) See Weaknesses.

1) Why are there expected values in Equations 1 and 5? Subtrajectory balance conditions mean that the forward and backward subtrajectory flows should be equal for all possible subtrajectories, not only on average (see [1]).

References:\
[1] Madan et al. Learning GFlowNets from partial episodes for improved convergence and stability. ICML 2023

---

> ### Author Response · Authors · 2025-11-19
> **Response 1**
>
> > "Overall, I found this to be an interesting paper... ".
>
>    We appreciate your recognition of our work. In response to the weaknesses you pointed out, we provide detailed, point-by-point replies below.
>
> 1. > “The main weakness of this work... Work [1] is neither discussed nor cited in this paper.”
>
>    Thanks for pointing out this related work. We have added a detailed discussion of [1] ( Appendix A.6) and its experiment results on BN task and molecular graph tasks in the updated manuscript. However, we would like to respectfully remark that the focus on Proposition [1] and our Theorem 3.1 is **distinct**. Our Theorem 1 is **not** proving the expression of $V^\dagger(s_0)$ (or generally $V^\dagger(s_h)$) can be rewritten in the form KL divergences, which is proved in Proposition 1 of [1]. Instead, our theorem 3.1 show that any paramterized function $V$ that satisfies the condition in our Theorem 3.1 is guaranteed to match $V^\dagger$, thereby allowing us to avoid explicitly computing the expectations inside $V^\dagger$. The detailed explanation is provided below:
>     * Firstly, for your reference, we generalize the proof of Proposition 1 for $V^\dagger(s_0)$ in [1] to general $V^\dagger(s_h)$ as follows.  The proof follows directly from the definition of $V^\dagger(s_h)$ for  $h \in [0,H]$ in [3]:
>
>       $V^\dagger(s_h):=E_{P_F(\tau_h|s_h)}\left[\sum_{i=h}^{H-1} \log \frac{\pi_F(s_{i+1}|s_i)}{\pi_B(s_i|s_{i+1})}+\log \pi_F(s_f|s_H)-\log R(x)\right]$
>
>       $=E_{P_F(\tau)}\left[\log \frac{\prod_{h=1}^H \pi_F(s_i|s_{i-1})P_B(s_h)Z^\ast\}{\prod_{h=1}^H \pi_B(s_i|s_{i-1})P_B(s_h)Z^\ast R(x)}\right]=D_{KL}(P_F(\tau)||P_B(\tau))-\log P_B(s_h)Z^\ast$
>
>       Then it reduces to proposition 1 for $V^\dagger(s_0)$ in [1] as  $P_B(s_0)=1$.
>
>    *  However, this KL expression does not tell us how to practically evaluate $V^\dagger(s_t)$ since it is computationally infeasible to enumerate all possible trajectories $\tau$ and explicitly compute the expectation inside $V^\dagger$. Therefore, in the RL literature, it is standard to derive various learning objectives based on the Temporal Difference (TD) formulation of the **Bellman equation** $V(s)=R(s \rightarrow s')s+E[V(s')])$ so that any parameterized function $V$ can directly approximate $V^\dagger$  without computing the full expectation (Chapter 6 in [7]). Correspondingly, we present the Sub-EB objective and corresponding proof, which is grounded in the **evaluation balance equation**.
>
>    *   In the RL literature[7], methods are commonly categorized into (soft) Q-learning approaches[8] and (soft) actor-critic approaches[9]. Works [1,2] follow the first direction,  while our work and [3] follow the latter direction. To better explain this point, we respectfully refer you to our response to Weakness 2, where we provide a detailed explanation.
>
>
> [7] Richard S. Sutton and Andrew G. Barto. Reinforcement learning: An introduction. MIT press, 2018.
>
> [8] Tuomas Haarnoja, Haoran Tang, Pieter Abbeel, and Sergey Levine. Reinforcement learning with deep energy-based policies. In International conference on machine learning, pp. 1352–1361. PMLR, 2017
>
> [9] Tuomas Haarnoja, Aurick Zhou, Pieter Abbeel, and Sergey Levine. Soft actor-critic: Off-policy maximum entropy deep reinforcement learning with a stochastic actor. In International conference on machine learning, pp. 1861–1870. PMLR, 2018.
>
> [10] Tristan Deleu, Padideh Nouri, Nikolay Malkin, Doina Precup, and Yoshua Bengio. Discrete probabilistic inference as control in multi-path environments. In Uncertainty in Artificial Intelligence, pp. 997–1021. PMLR, 2024

---

> ### Author Response · Authors · 2025-11-19
> **Response 2**
>
> 2. > "The proposed SubEB objective (Equation 8) exactly coincides with SubTB loss [2]...."
>
>     * We agree with your comments that they share the almost same form. However, we would like to humbly remark that the they play **distinct** role play for two different RL framework, where [1,2] and all flow-balancing objective (like  DB, TB,  Sub-TB) follows the framework of  (Soft-)Q-learning, while our work and [3], follow the (Soft) actor-critic framework.
>
>       * The logic between the two frameworks is distinct:  In the Soft Q-learning,  $V$ and $\pi_F$ are optimized jointly (via some objective) towards the **minimal** evaluation function $V^\ast=\log P_B(s_h)Z^\ast$ and optimal policy $\pi_F^\ast$. While  in the actor-critic,  we first learn a critic $V$ toward $V^\dagger$ (via some objective) that judge how far actor $\pi_F$ is from optimal actor $\pi_F^\ast$, and then minimize the policy gap.
>
>        * Tradition methods in the second direction requires a fixed $\pi_B$ and operates in an on-policy manner [7], while our Sub-EB enable **non-fixed** $\pi_B$ and **off-policy** learning. We modestly believe this represents a advancement which has not been achieved before.
>
>    * The detail explanation about the two framework is presented as follows (also provided in Appendix A.6 in the updated manuscript):
>
>       *  For the first direction, the core idea of Soft Q-learning [8] is to learn a function $Q$ that minimizes the mismatch of the Bellman equation (can be reduced from  **Sub-TB and [1]** ), which is $E_{P_\mathcal{D}(s\rightarrow s')}[\left(\log \textcolor{red}{\pi_B(s|s')+V(s')}-Q(s, s')\right)^2]$ for the GFlowNet transition environment $\mathcal{G}$ with reward $\log \pi_B$ and $\log \pi_B(x|s_f)+V(s_f):=\log R(x)$. Here,  $\pi_F(s'|s):= \frac{\exp Q(s,s')}{\exp V(s)}$ and $V(s):=\log \sum_{s'}\exp Q(s,s')$.  Any function $Q$ that achieves zero mismatch is guaranteed to equal the optimal soft Q-function $Q^\ast$ with the corresponding $V=V^\ast$ and $\pi_F=\pi_F^\ast$.
>          * The main contribution of [1] is that shows that if we treat $Q(s,s')$ as $\log F(s\rightarrow s')$ so that $V(s)=\log \sum_s' F(s\rightarrow s')=\log F(s)$, then the optimal solutions of the Detail Balance (DB) objective coincide exactly with the optimal solutions of the above Bellman objective. The distinction lies only in parameterization: one may parameterize $(\pi_F, V)$ directly and represent $Q(s,s')$ as $V(s)\pi_F(s'|s)$, instead of parameterizing $Q(s,s')$ and deriving $V$ and $\pi_F$ from it. As acknowledged by [1], their proof only applies to the DB objective with fixed $\pi_B$.
>          * To address this limitation, [10] established an equivalence between path-consistency learning (a generalized form of soft Q-learning) and the Sub-TB objective from a gradient-based perspective.
>          * Compared to [10], Our **Theorem 3.2**  offers a more direct and explicit connection along this RL direction.
>
>      * Our work and [3] is based on the theory of policy-gradients[11] which operates under the actor-critic framework[9], which operates under the actor–critic framework. In the framework, we learn a function $V$ that minimizes (but not necessarily) the Bellman objective (can be reduced from  **Sub-EB and TD-$\lambda$**), $ E_{P_F(s\rightarrow s')}[\left(\log \textcolor{red}{\pi_F(s'|s)+V(s)}-Q(s, s')\right)^2]$ with $Q(s,s'):=\log \pi_B(s|s')+V(s')$[9,12]. **At this point, the two RL directions begin to diverge.** When the $V$ achieve the optimal solution of the Bellman objective (denoted as $V^\dagger$):
>
>        $V(s)=E_{\pi_F(s'|s)}[Q(s,s')-\log \pi_F(s'|s)]$
>
>        $     =E_{\pi_F(s'|s)}\left[\log \frac{\exp Q(s,s')}{ \sum_{s'}\exp Q(s,s')    }-\log \pi_F(s'|s)+\log \sum_{s'}\exp Q(s,s')  \right  ]$
>
>        $     =- D_{KL}\left(\pi_F(s'|s)||\pi_Q(s'|s) \right) +\log \sum_{s'}\exp Q(s,s'), \quad  \pi_Q(s'|s):= \frac{\exp Q(s,s')}{\sum_{s'} \exp Q(s,s')}  $
>
>        Rewriting $\exp Q$ as $F$ yields expressions that coincide with those used in the main text. This clarifies why $V^\dagger$ (and its learned approximate $V$) serves as a critic: it evaluates how far the $\pi_F$ is from the (locally) optimal policy. The divergence is then minimized w.r.t. actor $\pi_F$ using learned critic $V$, so that true critic $V^\dagger$ moves closer to the optimal one $V^\ast$. However, traditional methods in this policy-gradient direction operate in a **on-policy** manner as indicated by the Bellman objective above and also require that $\pi_B$ is fixed.  Therefore, Sub-EB objective based on our Theorem 3.1 is proposed specifically to overcome these limitations.
>
>
> [11] Alekh Agarwal, Sham M Kakade, Jason D Lee, and Gaurav Mahajan. On the theory of policy gradient methods: Optimality, approximation, and distribution shift. Journal of Machine Learning Research, 22(98):1–76, 2021.
>
> [12] John Schulman, Xi Chen, and Pieter Abbeel. Equivalence between policy gradients and soft q-learning. arXiv preprint arXiv:1704.06440, 2017.

---

> ### Author Response · Authors · 2025-11-19
> **Responses 3,4, and 5**
>
> 3. >  "The writing in this work seems unclear in a number of places...."
>        Thank you for your careful examination. We have made the following improvements, which are highlighted blue in the updated manuscript:
>
>     * We have rewrite line 345 (line 323 in the updated manuscript) as " optimizing the KL divergence between $P_F(\tau)$ and the unnormalized distribution $P_B(\tau|x)R(x)(=P_B(\tau)Z^*)$, which has the gradient equivalence to the divergence between $P_F(\tau)$ and $P_B(\tau)$" .
>
>       * Indeed, due to this gradient equivalence, it is still possible to optimize the true divergence with the unknown value of $Z^*$. This observation is also central to the methods in [3, 4], upon which our work builds.
>
>     *  We have added the clarification “The definitions of $W^\dagger$ and $W$ are detailed in  Section 3.3.” at the first occurrence of $W^\dagger$ and $W$.
>
>     * Yes, Algorithm 1 use the sample-based approximation of Equation 4, where the expectation $E_{P_F(\tau)}\left[\sum_{h=0}^{H}A^\gamma(s_h,s_{h+1})\nabla_\theta\log \pi_F(s_{h+1}|s_h;\theta)\right]$  are approximated by  $\frac{1}{K}\sum_{k=0}^K \left[\sum_{h=0}^{H}A^\gamma(s_h^k,s_{h+1}^k)\nabla_\theta\log \pi_F(s_{h+1}^k|s_h^k;\theta)\right]$ for a batch of $K$ samples, ${\tau^k=(s_0^k,\ldots, s^k_f)}_{k=1}^K$. This is just like the approximated gradients of Sub-TB, explained in line 149/136 in the original/updated manuscripts.
>
>
>  4. > " Another minor point, but since the paper partially focuses on the backward policy optimization, it would be good to cite and discuss prior works on this topic in the text [5, 6]".
>
>     Thank you for your insightful suggestions. We have added a discussion of [5, 6] to Section 3.2  and included corresponding experimental results in our ablation study on $\pi_B$. (Both are highlighted in blue color). For your convenience, we also provide a detailed discussion below:
>
>        * [6] also focus on solving the limiated of prior RL methods [1,3] that required fixed $\pi_B$. Their approach alternates between optimizing the forward and backward policies. Applying their method to the policy-based framework, the workflow process as follows:  given a batch of trajectory samples $\\{\tau^k\\}_{k=1}^K$ from $\pi_F$, we should first freeze the parameters of $\pi_F$ and optimize $\pi_B$ by minimizing its negative log-likelihood
>
>          $-\sum_{k=1}^K \log P_B(\tau^k|x^k)$ on the sample batch. Then, we freeze $\pi_B$ and optimized $V$ and $\pi_F$ using TD-$\lambda$ and Policy-Gadient (PG), respectively.
>
>        * [5] has the similar idea, but use an additional replay buffer that stores all forward trajectories sampled in the past 20 training iterations. Due to its similarity with [6] and its focus on the TB method rather than RL methods, we did not include experimental results for this method. However, we are happy to include them later in the rebuttal period.
>
>       *  In contrast, our Sub-EB objective, due to its similarity to Sub-TB, allows $\pi_B$ to be freely chosen and jointly optimized with $V$, just as $\pi_B$ and state flow $F$ are jointly updated in Sub-TB.  Therefore, **no alternating / additional** optimization is required.
>
> 5. > " The improvements are very marginal in comparison to SubTB in Figure 3 in the case when the local search is used. When it is not used, the improvements are notable only in the first two plots."
>
>     * We apologize if there is no unclear presentation. The message, we want to convey in Figure 3 (Figure 7 in the update manuscript) is that Sub-EB-B with local search achieves comparable mode-discovery performance as Sub-TB-B with local search, and outperforms RL.  At the same time, Sub-EB can achieve comparable distribution-modeling performance to RL and outperform Sub-TB as indicated in Figures 6-9 (Figures 8-11 in the updated manuscripts).   Therefore, Sub-EB  provides good performance across both tasks, whereas Sub-TB and RL each excel in one task. Finally, we believe it is reasonable that similar performance between Sub-TB(-B) and Sub-EB(-B) appears in the two small-scale tasks.
>
>     * Besides, we have included the results of Sub-EB and Sub-TB with local search (i.e., Sub-EB-B and Sub-TB-B) for the large-scale BN tasks. The advantages of Sub-EB-B over Sub-TB-B are more pronounced in these setting.  Besides, 10-node cases (Figure 2 in the updated manuscript), we also add the 15 nodes case (Figure 12 in the updated manuscript).

---

> ### Author Response · Authors · 2025-11-19
> **Response 6 and Q1**
>
> 6. > "In addition, all the experiments in the main text are on environments of a very small scale.....There are experiments on a larger-scale structure learning task in Appendix, so I would suggest moving them to the main text and putting more emphasis on them."
>     * We have reorganized the paper as your kind suggestions. In the updated manuscripts, the sequence designs results and discussion are move into Appendix B.2, and depicted in Figures 7-11.  Meanwhile, BN structure learning tasks are now discussed in the main text.
>    *  In addition to the 10-node BN case (with approximately $4.18 \times 10^{18}$ states), we have added experiments on 15-node BN cases, which involve around $2.38 \times 10^{35}$ states.
>
>    * We also added results of  molecule graph design experiments [13] (terminating state space is larger than $10^{16}$) in the Appendix B.4 of the updated manuscript. The sEH  experiment is  simplified from this. For your convenience, we provide a brief summary of this experiment below:
>
>       * There are 105 predefined building blocks. Each block is a subgraph composed of several atoms. Within each block, 1-6 atoms are designated as bonding sites, allowing the block to be attached to other blocks during the molecule construction process. The generative process terminates when no available bonding sites are remaining or when the number of blocks in the molecule exceeds 8. The Octanol–Water Partition Coefficient (LogP) and c-Jun N-terminal Kinase 3 (JNK3) scores of pyTDC package are used as the reward function $R$ . For this experiment, the size of the terminating state space is clearly $> \frac{105!}{(105-8)!}\approx 10^{16}$.
>
> Q1. Thanks for your insightful question. There are two valid interpretations of the Sub-TB objective. The first way is:
>         $\forall \tau_{i,j}\in \tau \in \mathcal{T},   \left(\log \frac{ F(s_i)P_F(\tau_{i,j}|s_i)}{ F(s_j)P_B(\tau_{i,j}|s_j)}\right)^2=0$.   Equivalently, papers [14,15] interpret it as there is a **off-policy** [16] distribution $P_{\mathcal{D}}(\tau)$ such that $\forall \tau \in \mathcal{T}: P_{\mathcal{D}}(\tau)>0$ . Then the above expression can be rewritten as $E_{P_{\mathcal{D}}}[\left(\log \frac{ F(s_i)P_F(\tau_{i,j}|s_i)}{ F(s_j)P_B(\tau_{i,j}|s_j)}\right)^2]=0$. As $\forall \tau \in \mathcal{T}: P_{\mathcal{D}}(\tau)>0$ , the solutions to these two formulations are equivalent. Existing proofs for the Sub-TB objective/condition follow the first interpretation, while our Theorem 3.2 follows the second way, thereby having closer relationship to RL. This is also the partial reason that we claim 'Theorem 3.2 offers an alternative perspective' in line 213/203 of the original/updated manuscript.
>
>
>
> [13] Emmanuel Bengio, Moksh Jain, Maksym Korablyov, Doina Precup, and Yoshua Bengio. Flow network based generative models for non-iterative diverse candidate generation. Advances in Neural Information Processing Systems, 34:27381–27394, 2021.
>
> [14] Minsu Kim, Joohwan Ko, Dinghuai Zhang, Ling Pan, Taeyoung Yun, Woo Chang Kim, Jinkyoo Park, and Yoshua Bengio. Learning to Scale Logits for Temperature-Conditional GFlowNets. In NeurIPS 2023 AI for Science Workshop, 2023a.
>
> [15] Minsu Kim, Taeyoung Yun, Emmanuel Bengio, Dinghuai Zhang, Yoshua Bengio, Sungsoo Ahn, and Jinkyoo Park. Local search gflownets. In The Twelfth International Conference on Learning Representations, 2023b.
>
> [16] Nikolay Malkin, Salem Lahlou, Tristan Deleu, Xu Ji, Edward J Hu, Katie E Everett, Dinghuai Zhang, and Yoshua Bengio. Gflownets and variational inference. In The Eleventh International Conference on Learning Representations, 2022b.

---

### Official Review · Reviewer_P5HE · 2025-10-25

**Soundness:** 2
**Presentation:** 2
**Contribution:** 1
**Rating:** 2
**Confidence:** 4

**Summary:**

The authors proposed a novel algorithm for GFlowNet training, based on a policy-iteration-style framework, where the SubTB/PCL-style loss is used only to optimize the value and backward policy. In contrast, a policy gradient objective is applied to optimize the forward (generative) policy with an additional GAE-style variance reduction. The authors provided experimental validation of their method across several tasks, including hypergrids, sequence design, and Bayesian network learning.

**Strengths:**

- The first (to my knowledge) policy-iteration style approach for GFlowNet training, which shows a consistent performance on various benchmarks, including a high-dimensional one such as 10-vertex BNs and SEH;

**Weaknesses:**

- The theoretical results automatically seem to follow from the GFlowNet-RL equivalence, described in the works (Deleu et al. 2024, Tiapkin et al. 2024). In particular, thanks to the graded DAG structure, Theorems 3.1 and 3.2 follow from Proposition 1 of Tiapkin et al. (2024), applied to a sub-DAG that is rooted at the vertex $s_h$, as well as the interpretation of a value function as negative KL-divergence up to a log-normalizing constant. Also, the paper does not cite a related work by Tiapkin et al. (2024).
- The value loss itself is equal to on-policy SubTB loss, which also does not bring any novelty there: in many practical cases, the SubTB loss is applied in an on-policy manner, which is exactly the same as SubEB loss up to the lack of backpropagation through the forward policy.
- Lack of RL-inspired baseline comparison, such as Rectified Policy Evaluation (He et al. 2025) and Muchnausen DQN (see Vieillard et al. (2020) and its application to GFlowNets in Tiapkin et al. 2024).
- No discussion or comparison with the gradient variance reduction techniques of da Silva et al. (2024); if I understand correctly, the main difference between Sub-EB loss and Sub-TB loss is the usage of GAE-style variance-reduced objective for the forward policy, whereas the on-policy optimization through Sub-TB will result in a weaker variance reduction; thus, the comparison with other gradient variance reduction techniques applied in GFlowNets seems to be important.
- On the large-scale task of 10-vertex Bayesian Network generation, the presented metrics do not show how the method actually solves the sampling problem. I would recommend computing the reward correlation using a test set of randomly generated graphs, or using the FCS metric proposed by da Silva et al. (2025).

- Minor remark: Due to the on-policy nature of the algorithm, it's impossible to use a simple exploration technique such as $\varepsilon$-greedy exploration or replay buffers;

### References

Deleu, T., Nouri, P., Malkin, N., Precup, D., & Bengio, Y. (2024). Discrete probabilistic inference as control in multi-path environments. arXiv preprint UAI-2024;

Tiapkin, D., Morozov, N., Naumov, A., & Vetrov, D. P. (2024, April). Generative flow networks as entropy-regularized RL. In International Conference on Artificial Intelligence and Statistics (pp. 4213-4221). PMLR.

He, H., Bengio, E., Cai, Q., & Pan, L. (2024). Random policy evaluation uncovers policies of generative flow networks. ICML-2025

Vieillard, N., Pietquin, O., & Geist, M. (2020). Munchausen reinforcement learning. Advances in Neural Information Processing Systems, 33, 4235-4246.

Silva, T., de Souza da Silva, E., & Mesquita, D. (2024). On divergence measures for training gflownets. Advances in Neural Information Processing Systems, 37, 75883-75913.

Silva, T., Alves, R. B., da Silva, E. D. S., Souza, A. H., Garg, V., Kaski, S., & Mesquita, D. (2025). When do GFlowNets learn the right distribution?. In The Thirteenth International Conference on Learning Representations.

**Questions:**

- Is it possible to provide a simple toy experiment that shows that indeed using the SubTB-style loss provides a faster convergence of the value function to an actual regularized value of the current policy?
- Is it possible to provide a direct ablation on the variance reduction quality for the gradient of the forward policy?
- Do you use a shared backbone for your policy and flow estimators?
- How is it possible to introduce additional exploration into this algorithm?

---

> ### Author Response · Authors · 2025-11-26
> **Response 1**
>
> 1. * Thanks for providing these paper, we omit [Tiapkin et al. 2024] in our original submission asit is generalized by [Deleu et al. 2024]. We have added a detail discussion of these two works in the Appendix A.6 of the updated manuscript. However, we would like to humblely remark that our Theorem 3.1 is  **not** proving the expression of $V^\dagger(s_0)$ (or generally $V^\dagger(s_h)$) can be rewritten in the form KL divergences. Instead,  Theorem 3.1 show that  any paramterized function $V$ that satisfies the condition in Theorem 3.1 is guaranteed to match $V^\dagger$, thereby allowing us to avoid explicitly computing the expectations inside $V^\dagger$. The detailed explanation is provided below:
>
>      * Firstly, for your reference, we combine and restate the proof of Proposition 1 for $V^\dagger(s_0)$ in  [Tiapkin et al, 2024] to general $V^\dagger(s_h)$ as follows.  The proof follows directly from the definition of $V^\dagger(s_h)$ for  $h \in [0,H]$ in [1]:
>
>         $V^\dagger(s_h):=E_{P_F(\tau_h|s_h)}\left[\sum_{i=h}^{H-1} \log \frac{\pi_F(s_{i+1}|s_i)}{\pi_B(s_i|s_{i+1})}+\log \pi_F(s_f|s_H)-\log R(x)\right]$
>
>         $=E_{P_F(\tau)}\left[\log \frac{\prod_{h=1}^H \pi_F(s_i|s_{i-1})P_B(s_h)Z^\ast\}{\prod_{h=1}^H \pi_B(s_i|s_{i-1})P_B(s_h)Z^\ast R(x)}\right]=D_{KL}(P_F(\tau)||P_B(\tau))-\log P_B(s_h)Z^\ast$
>
>         Then it reduces to proposition 1 for $V^\dagger(s_0)$ in [1] as  $P_B(s_0)=1$.
>
>      *  This KL expression, however does not tell us how to practically evaluate $V^\dagger(s_t)$ since it is computationally infeasible to enumerate all possible trajectories $\tau$ and explicitly compute the expectation inside $V^\dagger$. Therefore, in the RL literature, it is standard to derive various learning objectives based on the Temporal Difference (TD) formulation of the **Bellman equation** $V(s)=R(s \rightarrow s')s+E[V(s')])$ so that any parameterized function $V$ can directly approximate $V^\dagger$  without computing the full expectation (Chapter 6 in [2]). Correspondingly, we introduce the Sub-EB objective, which is grounded in the **evaluation balance equation**.
>
>
>     *  Yes, RL methods focus on optimizing the $V(s_0)$[2]. They are commonly categorized into (soft) Q-learning approaches[3] and (soft) actor-critic approaches[4]. Works [Deleu et al. 2024, Tiapkin et al. 2024] and Sub-TB follow the first direction, while our work and [1] follow the latter direction. To better explain this point, we respectfully refer you to our response to Weakness 2, where we provide a detailed explanation.
>
>
> [1] Puhua Niu, Shili Wu, Mingzhou Fan, and Xiaoning Qian. Gflownet training by policy gradients. In Forty-first International Conference on Machine Learning, 2024
>
> [2] Richard S. Sutton and Andrew G. Barto. Reinforcement learning: An introduction. MIT press, 2018.
>
> [3] Tuomas Haarnoja, Haoran Tang, Pieter Abbeel, and Sergey Levine. Reinforcement learning with deep energy-based policies. In International conference on machine learning, pp. 1352–1361. PMLR, 2017
>
> [4] Tuomas Haarnoja, Aurick Zhou, Pieter Abbeel, and Sergey Levine. Soft actor-critic: Off-policy maximum entropy deep reinforcement learning with a stochastic actor. In International conference on machine learning, pp. 1861–1870. PMLR, 2018.

---

> ### Author Response · Authors · 2025-11-26
> **Response 2**
>
> 2. * We would like to humble point out that [Deleu et al. 2024], [Tiapkin et al. 2024] and Sub-TB follows the (soft-)Q learning framework, while our work follows the famous actor-critic framework[1]. In the Q-learning framework, both online or offline Sub-TB objective and their gradient are used for **optimizing** the policy, while in the actor-critic, the Sub-EB objective is used only for **evaluating** the gap between policy and desired policy, where policies are not updated based on its gradients.
>
>
>     * For your convenience, a detailed explanation of the two framework is provided below (and is also included in Appendix A.6 of the updated manuscript):
>
>        *  Under the first framework, the core idea of Soft Q-learning [4] is to learn a function $Q$ that minimizes the mismatch of the Bellman equation ( can be reduced from  **Sub-TB and [Tiapkin et al, 2024]**), which is $E_{P_\mathcal{D}(s\rightarrow s')}[\left(\log \textcolor{red}{\pi_B(s|s')+V(s')}-Q(s, s')\right)^2]$ for the GFlowNet transition environment $\mathcal{G}$ with reward $\log \pi_B$ and $\log\pi_B(x|s_f)+V(s_f):=\log R(x)$. Here,  $\pi_F(s'|s):= \frac{\exp Q(s,s')}{\exp V(s)}$ and $V(s):=\log \sum_{s'}\exp Q(s,s')$.  Any function $Q$ that achieves zero mismatch is guaranteed to equal the optimal soft Q-function $Q^\ast$ with the corresponding $V=V^\ast$ and $\pi_F=\pi_F^\ast$.
>
>            - (i) The main contribution of [Tiapkin et al, 2024]  is that shows that if we treat $Q(s,s')$ as $\log F(s\rightarrow s')$ so that $V(s)=\log \sum_s' F(s\rightarrow s')=\log F(s)$, then the optimal solutions of the Detail Balance (DB) objective coincide exactly with the optimal solutions of the above Bellman objective. The distinction lies only in parameterization: one may parameterize $(\pi_F, V)$ directly and represent $Q(s,s')$ as $V(s)\pi_F(s'|s)$, instead of parameterizing $Q(s,s')$ and deriving $V$ and $\pi_F$ from it. As acknowledged by [Tiapkin et al, 2024], their proof only applies to the DB objective with fixed $\pi_B$.
>            - (ii) To address this limitation, [Deleu et al. 2024] established an equivalence between path-consistency learning (a generalized form of soft Q-learning) and the Sub-TB objective from a gradient-based perspective.
>            - (iii) Compared to [Deleu et al. 2024], **our Theorem 3.2** aims at offering a more direct and explicit connection along this RL direction.
>
>         * Our work and all policy-gradients methods[1] operates under the actor–critic framework[4]. In the framework, we learn a function $V$ that minimizes (but not necessarily) the Bellman objective ( can be reduced from  **TD-$\lambda$ or Sub-EB**), $ E_{P_F(s\rightarrow s')}[\left(\log \textcolor{red}{\pi_F(s'|s)+V(s)}-Q(s, s')\right)^2]$ with $Q(s,s'):=\log \pi_B(s|s')+V(s')$[4,5]. **At this point, the two RL directions begin to diverge.** When the $V$ achieve the optimal solution of the Bellman objective (denoted as $V^\dagger$):
>
>            $V(s)=E_{\pi_F(s'|s)}[Q(s,s')-\log \pi_F(s'|s)]$
>
>            $     =E_{\pi_F(s'|s)}\left[\log \frac{\exp Q(s,s')}{ \sum_{s'}\exp Q(s,s')    }-\log \pi_F(s'|s)+\log \sum_{s'}\exp Q(s,s')  \right  ]$
>
>            $     =- D_{KL}\left(\pi_F(s'|s)||\pi_Q(s'|s) \right) +\log \sum_{s'}\exp Q(s,s'), \quad  \pi_Q(s'|s):= \frac{\exp Q(s,s')}{\sum_{s'} \exp Q(s,s')}  $
>
>            Rewriting $\exp Q$ as $F$ yields expressions that coincide with those used in the main text. This clarifies why $V^\dagger$ (and its learned approximate $V$) serves as a critic: it evaluates how far the $\pi_F$ is from the (locally) optimal policy. The divergence is then minimized w.r.t. actor $\pi_F$ using critic $V$, so that it $V^\dagger$ of $\pi_F$ moves closer to the optimal one $V^\ast$.
>
>             *  However, traditional actor-critic methods based on policy-gradient operate in a **on-policy** manner as indicated by the Bellman objective above and also require that $\pi_B$ is fixed.  Therefore, Sub-EB objective derived from our Theorem 3.1 is proposed specifically to overcome these limitations.
>
>
> [5] John Schulman, Xi Chen, and Pieter Abbeel. Equivalence between policy gradients and soft q- learning. arXiv preprint arXiv:1704.06440, 2017.

---

> ### Author Response · Authors · 2025-11-26
> **Responses 3-6**
>
> 3. We have added discussion and corresponding experiment results of Muchnausen DQN for  [Tiapkin et al. 2024] for both 10-vertex  and 15-vertex BN (Figures 2 and 12 in the updated manuscript ).  For Rectified Policy Evaluation [He et al. 2025], we have added the discussion in related work section, but we found it can not be applied to [Tiapkin et al. 2024] for the following reasons:
>
>     *  Rectified Policy Evaluation (He et al. 2025) is proposed for improving Flow Matching objective. In this setting, policy $\pi_F(s'|s)$ is rectified as $\frac{F(s')P_B(s|s')}{F(s)}$, and the state-wise mismatches (expected edge-wise mismatches) are computed.
>
>    *   [Tiapkin et al. 2024] takes the similar form to Detail Balance objective. It compute the edge-wise mismatch between $ F(s\rightarrow s')= \exp Q(s \rightarrow s')$ and $ \pi_B(s|s')F(s')$,  and $\pi_F(s'|s):= \frac{\exp Q(s\rightarrow s')}{\sum_{s''} \exp Q(s\rightarrow s'')}$.  If we rewrite $F(s\rightarrow s')$ as $\pi_F(s'|s) F(s)$ as indicated in  [He et al. 2025],   the learning objective becomes identically zero, eliminating the learning signal.
>
>
>
> 4. We have add the results and discussion of Silva et al. (2024) in both hyper-grids (Figure 1 and 5 ) and BN tasks (Figure 2 and 12) in the updated manuscript.
>
>
> 5. We appreciate your very useful suggestion!  We have add the FCS metric for both 10-vertex and 15 vertex BN tasks  (Figure 2 and 12 in the updated manuscript). The results show that only RL and our Sub-EB can achieve good distribution modeling performance. Following Silva et al. (2025), we sample a batch of 32 terminating states, and for each terminating state we sample 128 trajectories ending at that state.
>
>
> 6. We apologize if there is any unclear presentation. One of the main contributions of Sub-EB is that it enables the use of **offline samplers and replay buffers**, as described in Section 3.3 and Algorithm 2 in our original submission.  Accordingly, in our sequence-design experiments, we compared Sub-EB and Sub-TB when they are equipped with the local-search technique[6], and denoted as Sub-EB-B and Sub-TB-B, respectively.  The technique relies on backward–forward offline sampling and a prioritized replay buffer to explore high-reward states. The results are presented in Figures 7-11 in the updated manuscript (Figure 3, 6-9 in the original manuscript).
>
>     *  To make results more convincing, we have also added the results of Sub-EB-B and Sub-TB-B for 10-vertex and 15-vertex BN tasks in Figures 2 and 12 of the updated manuscript.
>
>
> [6] Minsu Kim, Taeyoung Yun, Emmanuel Bengio, Dinghuai Zhang, Yoshua Bengio, Sungsoo Ahn, and Jinkyoo Park. Local search gflownets. In The Twelfth International Conference on Learning Representations, 2023b.

---

> ### Author Response · Authors · 2025-11-26
> **Responses Q1-Q4**
>
> Q1. We understood your concern as asking whether the Sub-EB improve TD-$\lambda$ solely for learning
>  towards the true evaluation function. We respectfully suggest that this kind of comparison is not appropriate, and we manage to provide an alternative comparision:
>
> * Unlike Sub-TB, where making $F$ match $F^\ast$ is part of the ultimate goal,  V^\dagger is only a intermediate quantity in RL literature[1,2]. It continuously changes whenever the parameters of the forward policy $\pi_F$ are updated.
>
> * In this sense, given enough training data and fixed $\pi_F$, whether $V$ matches $V^\dagger$ is not important. What matters is, given limited data, whether $V$ can effectively guide the learning process of $\pi_F$. As discussed in our 'Related work' section, trajectory-wise objectives of $V$ bias to global exploration during the learning process of $\pi_F$, while edge-wise versions bias to local exploitation. The challenge lies in the trade-off between exploration and exploitation.
> * Therefore, both the original TD-$\lambda$ objective paper [7] and Chapter 6 (Temporal-Difference Learning) of book [2] emphasize how the choice of objective for  impact the learning process of , rather than whether  is close to  at a fixed.
>
> * Instead, we report the counts of visited states for each reward value (0.01, 0.5, 2.5)  during the training process of the $256\times256$ grid experiment. As shown below, the results indicate that state-visitation distribution of Sub-EB is more balanced than that of RL, as Sub‑EB visits low‑rewarded and moderately‑rewarded states more frequently than RL.
>
>
> | **Step** | **RL (0.01)**   | **RL (0.50)**    | **RL (2.50)**    | **Sub-EB (0.01)** | **Sub-EB (0.50)** | **Sub-EB (2.50)** |
> | -------: | --------------- | ---------------- | ---------------- | ----------------- | ----------------- | ----------------- |
> |  **200** | 3272.2 ± 424.6  | 12651.6 ± 444.2  | **9676.2 ± 816.9**   | **4851.8 ± 275.1**    | **13066.4 ± 144.3**   | 7681.8 ± 331.4    |
> |  **400** | 6138.6 ± 746.7  | 26430.0 ± 648.8  | **18631.4 ± 1358.0** | **9360.4 ± 278.4**    | **27137.8 ± 179.0**   | 14701.8 ± 343.4   |
> |  **600** | 7706.0 ± 722.8  | 39594.8 ± 746.1  | **29499.2 ± 1363.6** | **11972.2 ± 463.9**   | **40660.6 ± 217.3**   | 24167.2 ± 606.3   |
> |  **800** | 8979.0 ± 714.4  | 52849.8 ± 734.7  | **40571.2 ± 1309.8** | **13967.0 ± 530.1**   | **53997.0 ± 207.1**   | 34436.0 ± 683.6   |
> | **1000** | 10047.2 ± 657.6 | 65934.6 ± 807.8  | **52018.2 ± 1319.8** | **15695.4 ± 561.7**   | **67212.0 ± 251.6**   | 45092.6 ± 779.9   |
> | **1200** | 11114.8 ± 446.0 | 78986.0 ± 811.7  | **63499.2 ± 1044.5** | **17231.0 ± 579.7**   | **80342.6 ± 325.2**   | 56026.4 ± 868.3   |
> | **1400** | 12194.0 ± 479.7 | 92247.2 ± 591.4  | **74758.8 ± 455.7**  | **18580.4 ± 602.6**  | **93489.8 ± 309.6**   | 67129.8 ± 856.8   |
> | **1600** | 13071.4 ± 534.9 | 105358.4 ± 546.4 | **86370.2 ± 428.4**  | **19853.8 ± 613.1**   | **106625.6 ± 315.5**  | 78320.6 ± 852.6   |
> | **1800** | 13934.6 ± 542.8 | 118365.8 ± 583.1 | **98099.6 ± 449.2**  | **21060.2 ± 604.9**   | **119758.0 ± 342.1**  | 89581.8 ± 907.4   |
> | **2000** | 14779.8 ± 546.7 | 131419.2 ± 581.8 | **109801.0 ± 510.1** | **22180.8 ± 626.8**   | **132856.6 ± 357.0**  | 100962.6 ± 940.8  |
>
>
>
> Q2.   We have examined da Silva et al. (2024) as well as the prior work [8] on which it is built. Both papers discuss variance-reduction techniques primarily in terms of performance, which is consistent with our treatment. Therefore, we are unsure which specific factor your requested ablation refers to. Could you please clarify which component or setting you would like us to ablate? Additionally, we would appreciate guidance on which policy-variance quality metric you would like to see.
>
> Q3. No, our code is built upon [9], and follows its default setups, where policy and state flow estimator does not share the same structure.
>
> Q4. We modestly refer you to our response to Weakness 6.
>
>
>
>
>
> [7] John Schulman, Philipp Moritz, Sergey Levine, Michael Jordan, and Pieter Abbeel. High-dimensional continuous control using generalized advantage estimation. In Proceedings of the International Conference on Learning Representations (ICLR), 2016.
>
>
> [8] Malkin, N., Lahlou, S., Deleu, T., Ji, X., Hu, E. J., Everett, K. E., Zhang, D., and Bengio, Y. Gflownets and variational inference. In The Eleventh International Conference on Learning Representations, 2023
>
> [9] Salem Lahlou, Joseph D Viviano, and Victor Schmidt. torchgfn: A pytorch gflownet library. arXiv preprint arXiv:2305.14594, 2023.

---

> > ### Comment · Reviewer_P5HE · 2025-11-27
> >
> > I would like to thank the authors for their response and for resolving many of my concerns. Indeed, the positions of Theorem 3.1. become much clearer to me right now, and I am happy to increase my score.
> >
> > W3. Thank you very much for adding one of the baselines and an additional discussion on the difference between these approaches. However, it would be very beneficial to add the second one, despite the ideological differences between the methods.
> >
> > W6. Indeed, there was a misunderstanding. The current presentation in Section 3.3 seemed to me to be a completely offline method that requires a fixed dataset of terminal states and thus is not applicable to the usual GFlowNet training scenario, and also assumes $\pi_F$ to be fixed. It would be helpful if you could clarify the application of this algorithm in a usual online training of GFlowNets without a predefined dataset of terminal states.
> >
> > Q2. I am sorry for the misunderstanding. I meant two types of study: (1) performance-based comparison to other actor-critic-based approaches with variance reduction (as in da Silva et al. (2024)) and (2) study on a gradient variance similar to that presented in the SubTB paper (Madan et al. 2023).
> >
> > Madan, K., Rector-Brooks, J., Korablyov, M., Bengio, E., Jain, M., Nica, A. C., ... & Malkin, N. (2023, July). Learning gflownets from partial episodes for improved convergence and stability. In International Conference on Machine Learning (pp. 23467-23483). PMLR.

---

> ### Author Response · Authors · 2025-12-01
>
> W6. A fixed set of terminating states $x$ is **never** required.
>
> * In the usual training scenarios, trajectories $\tau$ are sampled from an offline sampler $P_\mathcal{D}(\tau)$, which is usually defined from $P_F$ (and augmented with replay buffer); and the model is updated based on these sampled trajectories.
>
> * In our method, trajectories $\tau$ are still sampled by $P_\mathcal{D}(\tau)$, but we replace sub-trajectories $\tau|x$ before $x$ by the sub-trajectory samples from $P_B(\tau|x)$, and update the model based on these modified trajectories.
>
> * The reason is that only the terminal states $x$ are informed by real-world feedback via reward $R(x)$, and therefore exploration has its greatest impact in how effectively new terminal states $x$ are discovered.
>
>
> Q2. (1) We compared the performance over $128\times 128$ grid for Silva et al. (2024), the actor critic method, and our Sub-EB. (2) We further conducted the gradient variance study following the procedure of Madan et al. (2023). The discussion is provided in Appendix B5, and Figure 17, where Sub-EB has the lowest variance (corresponding to high cosine similarity) for all $k\in \\{2,3,\ldots,9\\}$.
>
>
> W3. For He et al. (2025), their method suffers from numerical instability and therefore cannot be applied to complex or large-scale generative tasks such as the Bayesian Network (BN) structure learning tasks. For a 10-node BN with 12 edges, a sampled trajectory is provided as follows:
>
> * $[]            \rightarrow [(8, 5)]\rightarrow [\ldots, (7, 0)] \rightarrow [\ldots, (9, 3)] \rightarrow  [\ldots, (4, 7)] $
>
>      $\rightarrow  [\ldots, (5, 4)] \rightarrow  [\ldots, (3, 8)]\rightarrow  [\ldots, (8, 6)]\rightarrow  [\ldots, (1, 6)] $
>
>      $\rightarrow  [\ldots, (4, 2) ]\rightarrow  [\ldots, (1, 5)] \rightarrow  [\ldots, (6, 2)] \rightarrow  [\ldots, (5, 7)]] $
>
> * We then need to compute $g(s_t):=\prod_{i=0}^{t-1}A(S_{i})/B(s_{i+1})$ for $t \in[ 1,\ldots, 12]$, where $A(s)$ and $B(s)$ are the numbers of outgoing and incoming actions at state $s$.  In this BN example,
>    * $A(s_{0:11})= 90,   88,   87, 85,  82,  75,  64,  60,  58,  52,  46,44$,
>    * $B(s_{1:12})=  1,   2,   3,   4,   5,   6,   7,   8,   9,  10,  11,  12$, as we can only remove existing edges.
>    * $g(s_{1:12}) =90, 3.96\times 10^3, 1.1484 \times 10^5, 2.4404 \times 10^6, 4.0022 \times 10^7, 5.0027 \times 10^8,$
>      $4.5739\times 10^9, 3.4304\times 10^{10}, 2.2107\times 10^{11}, 1.1496\times 10^{12}, 4.8073\times 10^{12}, 1.7627\times 10^{13}.$ This large growth of $g(s)$ makes it numerically difficult to compute the mean-square-error objective w.r.t. $g(s)F_\theta(s)$, which quickly becomes unstable.
>
>
> Besides, according to Algo.3 in the paper $\pi_F(s'|s):=\frac{F_\theta(s')\pi_B(s|s')}{F_\theta(s)}$ and $\pi_B(s|s')=\frac{1}{B(s')}$ are uniform. While $\sum_{s'} \pi_F(s'|s)=\frac{\sum_{s'}F_\theta(s')/B(s')}{F_\theta(s)}$ clearly can **not** be guaranteed to always be $1$, and $\pi_F$ can be a invalid policy, they sample from a **uniform** $\pi_F(s'|s):=\frac{1}{A(s)}$, but this can lead to poor performance, especially for the large space size (e.g. $4.18\times 10^{18}$ possible 10-node BNs) .  Since the code for this method is not publicly available, we are unable to resolve these ambiguities or implement the necessary corrections for a fair comparison.

---

> > ### Comment · Area_Chair_y9ti · 2025-12-02
> >
> > Dear authors,
> >
> > Since the reviewer is currently unavailable for follow-up, I am stepping in to continue this discussion as there remain some concerns regarding the latest author response yesterday, to ensure clarity in the final stage. I have carefully reviewed the referenced papers and the comments.
> >
> > (1) Regarding the additional experiments where "Q-Much is shown to have the lowest performance", this appears inconsistent with the results in (Tiapkin et al. (2024)), which lacks a clear explanation. Clarifying the specific experimental setup or other differences that might explain this discrepancy is helpful.
> >
> > (2) Regarding the statement that the policy $P_F$ "does not sum to 1", this characterization appears inaccurate. $P_F$ is the final output of Algorithm 3 (He et al. (2025)) that you mentioned, which naturally holds at convergence -- based on the theorems in previous sections where F(s) represents state flows of a well-trained GFlowNet that samples proportionally from the reward. During training, similar to the standard GFlowNet formulations where edge flows act as unnormalized logits, one can simply apply normalization to convert it into probabilities at the implementation level.
> >
> > Could authors please provide clarification for this newly added baseline and refine the statements to ensure an accurate representation?

---

> ### Author Response · Authors · 2025-12-03
> **Response to (1)**
>
> Dear AC,
>
> Thank you very much for taking the time to follow up and ask for further clarification. We sincerely appreciate your careful review of our responses and the referenced papers. Below, we address each of the two points you raised.
> 1. We would like to clarify that Tiapkin et al. (2024)  (i.e. Q-much) did not conduct experiments on BN structure learning tasks. Besides,  the performance of  Q-much (red in Figures 2 and 12) is better than CV (pink in Figures 2 and 12). Nevertheless, to address potential disparity, we have conducted additional experiments on the hypergrid tasks, which was evaluated in both Sub-TB and Q-much. We obtained the following results.
>     * As explained in our Appendix A.6, Q-much prioritizes **exploitation**. Because the target Q is set to $\log F_\theta(s')P_B(s|s')$, where $F_\theta$ encodes the currently learned partial knowledge about the task, resulting in **biased** but low-variance task feedback.  In contrast, the (Sub-)trajectory level objective in our work encourages **exploration** as the target trajectory flow $ P_B (\tau|x)R(x)$ is exactly equal to optimal flow $F^\ast(\tau)$, serving as **unbiased** but high-variance feedback.  Therefore, Q-much may perform less effectively on challenging tasks that have well-separated modes.
>    * To verify this, we conducted hyper-grid tasks on two-dimensional grids with height equal to 20 (one setting in Tiapkin et al. (2024)) and 128 (one of our settings).  Hyper-grids are homogeneous across dimensions, so the environment height $N$ usually has a greater effect on the modeling difficulty level than the grid dimension $D$.  To be more specific,  the minimum distance between reward modes only depends on height $N$; thus for a grid with small $N$ and moderate $D$,  even random sampling can still identify reward modes.  The experimental results are presented in Figure 4 in Appendix B.
>       * On the 20×20 grid with batch size 16 (the configuration used in Tiapkin et al. (2024)), Q-much performs better than Sub-TB. However, this advantage disappears when using batch size 128 (our setting), as shown in Figure 4.
>      *  For $128 \times 128$ grid with the batch size equal to 128,  the performance of Q-much is considerably worse than Sub-TB.   Although both methods fail to converge to a good solution when the batch size is 16, Q-much still underperforms Sub-TB.

---

> ### Author Response · Authors · 2025-12-03
> **Response to (2)**
>
> 2.  For the optimal flow $F^\ast$[10], it is known that $\pi_B(s|s'):=\frac{F^\ast(s \rightarrow s')}{F^\ast(s')}$ and $\sum_{s'} F^\ast(s\rightarrow s')=F^\ast(s)$.
>      * We fully agree that, in the **ideal** converged point where $F_\theta(s)=F^\ast(s)$ for all states, using the above definition, we can obtain $\sum_{s'} \pi_F(s'|s):=\sum_{s'} \frac{F^\ast(s')\pi_B(s|s')}{F^\ast(s)}=\frac{\sum_{s'} F^\ast(s\rightarrow s')}{F(s')}=1$. However, in **practice**, a neural network $F_\theta$ can hardly satisfy $F_\theta=F^\ast$ exactly for all states. As a result, $\pi_F$ may fail to normalize to $1$, meaning $\pi_F$ can be an invalid probability distribution.
>      We appreciate your comments about the renormalization solution, and we did investigate this option.  However, computational issues still persist.
>      *  When sampling trajectories $\tau=(s_0\ldots s\rightarrow s' \ldots s_f)$ using $\pi_F(s'|s):=\frac{F_\theta(s')\pi_B(s|s')}{F(s)}$, we need to evaluate the neural network $F_\theta$ **once for every outgoing action**,  meaning that we require $A(s)$ forward passes through $F_\theta$. In contrast, in standard GFlowNet methods such as Sub-TB, TB, and DB [10],  the quantities $\pi_F(s'|s)$ or $F_\theta(s\rightarrow s')$ are modeled directly by neural networks, requiring only a single forward pass of the neural networks to produce all state-action probabilities.
>     * This difference gives rise to significant computational burdens. To sample the above BN trajectory, The number of forward passes for standard methods (and our methods) is simply the number of edges in the trajectory plus one terminal signal, i.e. $\\#$ forward passes **=12 +1**.  In contrast, applying the renormalization to the baseline method, $\\#$ forward passes =90 + 88+87+85+82+75+64+60+58+52+46+44 =**831 +1**.   This results in a >60 times increase in neural network evaluations during sampling.
>
>     * To mitigate this issue, we vectorized the implementation so that the neural network receives the batch of all successor states simultaneously. This avoids explicitly looping over each successor state and partially alleviates the computational burden. However, since we still need to evaluate $F_\theta$ for each outgoing action, the runtime remains significantly larger than that of standard GFlowNet methods. We found that the runtime for 10-node BN tasks is about 3h 8m, while it is only 10m for Sub-TB and Sub-EB as shown in Table 4.
>
>     * Besides,  $F(s_f)\pi_B(s|s_f)$ is not defined when constructing $\pi_F(s_f|s)$. In graded DAG environments, this issue does not arise because the terminating-state layer $X=S_H$, so we can impose  $\pi_F(s_f|s_H)=1$ and $\pi_F(s_f|s_{<H})=0$.  However, in general DAG environments (like the hyper-grid and BN tasks), $X=\cup_{h=0}^H S_h$, meaning that any $s\neq s_f $ can be a terminating state $\in X$. Therefore, $\pi_F(s_f|s)$ must be defined for all states. While one may conceptually introduce dummy layers to convert a general DAG into a graded one for theoretical analysis, this issue still needs to be addressed explicitly at the implementation level. In practice, we conjecture that the appropriate definition should be $F(s_f)\pi_B(s|s_f):=R(s)$, as adopted in our work.
>
> * Finally, we sincerely appreciate your efforts in reviewing our responses, and fully respect both the reviewer’s and your comments. As you can see in our original rebuttal, we provided very detailed derivations to address the all four reviewers' concerns, conducted all additional experiments requested, and incorporated all feasible baselines except the method by He et al. (2025) due to the unavailability of their code/implementation. Following the reviewer's request, we still exerted our best efforts to reproduce this method by ourselves and, in doing so, encountered multiple practical issues. We believe that it is our responsibility to report these findings transparently throughout the process.  Moreover, since these issues and the potential remedies are not described in the published paper, we do not consider it appropriate to further develop/modify the described method in their original paper for our rebuttal.
>
>
> [10] Yoshua Bengio, Salem Lahlou, Tristan Deleu, Edward J Hu, Mo Tiwari, and Emmanuel Bengio. Gflownet foundations. Journal of Machine Learning Research, 24(210):1–55, 2023.

---

> > ### Comment · Area_Chair_y9ti · 2025-12-03
> >
> > Dear authors,
> >
> > I appreciate the detailed illustrations.
> >
> > (1) Thanks for the detailed explanation regarding the comparison with Q-much (Tiapkin et al. (2024)) and the additional experiments; it is much clearer now.
> >
> > It will be helpful to include these results in the main paper's figures (in Fig 1 first for the hypergrids, as it is included in Fig 2 now for the BN structure learning task).
> >
> > (2) Thanks for the additional computation time comparison. Since Algorithm 3 (He et al. (2025)) is based on flow iteration (analogous to Q-iteration in RL), it can face scalability challenges as the authors mentioned.
> >
> > However, their theoretical guarantees are derived under specific assumptions tailored for certain DAG structures. Those assumptions may not hold in hypergrids as discussed in (He et al. (2025)) or BN structure learning tasks discussed here, which fall outside its intended domain of that specific algorithm.

---

> > > ### Author Response · Authors · 2025-12-04
> > >
> > > Dear AC,
> > >
> > > Thanks for your reply. We have updated the manuscript accordingly and added the results in Figure 1 for $D_{\mathrm{TV}}$ and in Figure 5 for $D_{\mathrm{JSD}}$.

---

### Official Review · Reviewer_wVfJ · 2025-10-31

**Soundness:** 4
**Presentation:** 3
**Contribution:** 3
**Rating:** 8
**Confidence:** 2

**Summary:**

The authors build upon prior work on GFlowNet training with policy gradients where an evaluation function is defined using rewards based on the log-ratio of forward and backward transitions. This formulation establishes a direct connection between minimizing the value function and minimizing the KL divergence between the forward and backward policies.

In this work, the authors propose a novel objective called Subtrajectory Evaluation Balance (Sub-EB), which enables more balanced learning compared to the conventional RL λ-TD objective. Sub-EB incorporates information from subtrajectories that both start and end at a given step whereas the conventional $\lambda$-TD objective focuses on events starting at a step and edge-wise mismatches only while also allowing for arbitrary weighting schemes, unlike λ-TD, which is constrained by an exponential λ-decay. It additionally establishes a theoretical connection between the flow function and their evaluation function.

Furthermore, the authors extend their approach to off-policy learning, in contrast to Niu et al. (2024). They introduce an evaluation function for backward trajectories, whose minimization likewise reduces the KL divergence. Because the objective is defined over backward trajectories, this formulation allows the use of a behavior policy distinct from the forward policy to generate states from which the backward rollouts are performed.

Their results demonstrate strong performance compared to the Sub-TB objective as well as the $\lambda$-TD one. Their objective allows for faster convergence with less variance when looking at total variation. Likewise, they outperform Sub-TB in mode discovery with their best performance coming from Sub-EB-p that parametrizes the backward policy.

**Strengths:**

- Strong theoretical analysis and results
- I appreciated the theoretical connections established between the evaluation function and the flow function.

**Weaknesses:**

N/A

**Questions:**

- Figure 3 is too small and hard to read. It would be better if the font is made bigger.
- It would be good to compare against LED-GFN since they learn energy decompositions that can be seen as a proxy for evaluating state-transitions.
- In theorem 3.2, it would be good to clarify in the main text what the minimum is taken over exactly.

---

> ### Author Response · Authors · 2025-11-19
> **Responses Q1, Q2, and Q3**
>
> We sincerely appreciate the positive recognition of our contributions, especially your remarks on the connections we established between the evaluation and flow functions, and the extension to off-policy learning.  Below we provide point-by-point responses to your questions/suggestions.
>
> Q1. We have enlarge the figure 3 (figure 7 in the updated manuscript) for the mode-discovery performances in sequence design and moved it to Appendix B.2, together with the figures for distribution modeling performances in sequence design. We believe this would makes the presentation more consistent.
>
> Q2. Thanks for pointing out this paper.  The idea is to approximate the terminal reward $R(x)$ by a sum of edge-wise potentials $\sum_{h=0}^H \phi(s_h \rightarrow s_{h+1})$, thereby ensuring that all edges, rather than just the final transition $(x\rightarrow s_f)$ are informed by reward signals.  We does not report the corresponding experiments results for the following reason, but we are happy to include them later in the rebuttal period.
>
>   * If we instead consider $\log R(x)\approx \sum_{h=0}^H \log \phi(s_h \rightarrow s_{h+1})$, this idea can also be naturally incorporated into our framework.. Given the policy-based optimization objective,  $V^\dagger(s_0;\theta):=E_{P_F(\tau;\theta)}\left[\sum_{i=0}^{H-1} \log \frac{\pi_F(s_{i+1}|s_i;\theta)}{\pi_B(s_i|s_{i+1})}+\log \pi_F(s_f|x)-\log R(x)\right]$, replacing $\log R(x)$ by its approximation yields $\approx E_{P_F(\tau;\theta)}\left[\sum_{i=0}^{H} \log \frac{\pi_F(s_{i+1}|s_i;\theta)}{\pi_B(s_i|s_{i+1})\phi(s_h \rightarrow s_{h+1})}\right]:=E_{P_F(\tau;\theta)}\left[\sum_{i=0}^{H} R_\phi(s_h\rightarrow s_{h+1};\theta)\right]$.
>   * The idea is very useful for graded DAG,  where the terminating state space $X$ corresponds to the last state layer $S_H$, and reward function $R$ is only defined for states in  $X$. In such cases, using edge-wise reward decomposition helps propagate reward signals to intermediate states.  However, for general non-graded DAG ( in the hyper-grid and BN tasks),  every state can be a terminating state, i.e. $X=\cup_{h=0}^H S_{h}$. Therefore, all states are already directly informed by reward signals,
>
> Q3. We have rewritten $\min D_{KL}(P_F(\tau_{h:}|s_h)\|P_B(\tau_{h:}|s_h))$ as  $D_{KL}(P_F^\ast(\tau_{h:}|s_h)\|P_B(\tau_{h:}|s_h))$, and states that $\pi_F=\pi_F^\ast$. This avoid the ambiguity regarding what the minimization is taken over.

---

> > ### Comment · Reviewer_wVfJ · 2025-11-25
> >
> > I thank the authors for their response and maintain my positive assessment of their work.

---

### Official Review · Reviewer_hiRW · 2025-11-01

**Soundness:** 2
**Presentation:** 3
**Contribution:** 2
**Rating:** 4
**Confidence:** 4

**Summary:**

This paper proposes Subtrajectory Evaluation Balance (Sub-EB), a new training principle for GFlowNets that enables learning a value function V(s) from partial trajectories, serving as a critic for forward policy improvement. The authors derive Sub-EB as a sufficient condition for flow consistency, connecting GFlowNets to reinforcement learning style. Empirical results across synthetic and real-world tasks demonstrate improved training stability, flexibility in backward policy design, and effectiveness in off-policy learning scenarios.

**Strengths:**

1. Sub-EB gives an alternative way for optimization of value-based and policy-based GFlowNet;

2. Both on-policy and off-policy training regimes are considered, which broadens the method's applicability.

**Weaknesses:**

1. The result analogous to theorems 1-2 in current submission are well-known from the point of view of entropy-regularized (soft) RL, see [Tiapkin et al, 2024]. In particular, Proposition 1 in [Tiapkin et al, 2024] is precisely the result of Theorem 1 for complete trajectories. In light of this result, novelty and technical originality of theorems 1 and 2 is limited.

2. It’s unclear how much faster the algorithm is in wall-clock time, rather than in terms of the number of steps;

3. The paper does not provide a theoretical guarantee that optimizing the Sub-EB objective leads to convergence of the target distribution, even under ideal conditions;

4. The paper does not provide sufficient comparisons with other approaches that use learnable backward policies, e.g. [Gritsaev et al, 2024], [Jang et al, 2024]. Also the paper does not include Sub-TB-B or Sub-EB-B experiments on the Bayesian network structure learning task.

References:

[Tiapkin et al, 2024] Tiapkin, D., Morozov, N., Naumov, A., and Vetrov, D. P. (2024). Generative flow networks as entropy-regularized RL. In AISTATS-2024.

[Gritsaev et al, 2024] Gritsaev, T., Morozov, N., Samsonov, S., and Tiapkin, D. (2025). Optimizing backward policies in GFlownets via trajectory likelihood maximization. In ICLR-2025.

[Jang et al, 2024] Jang, H., Jang, Y., Kim, M., Park, J., and Ahn, S. (2024a). Pessimistic backward policy for GFlowNets. In Neurips-2024.

**Questions:**

1. Is it possible to prove that Sub-EB condition imply that the learned forward policy $\pi_F$ is optimal?

2. Is it possible to perform ablation studies on the update frequency of the evaluator V?

---

> ### Author Response · Authors · 2025-11-19
> **Response 1**
>
> 1. * We appreciate your insightful comments. We have added a detailed discussion of [Tiapkin et al, 2024] (Appendix A.6) and its experiment results on BN tasks in the updated manuscript. However, we would like to respectfully remark that our Theorem 3.1 is  **not** proving the expression of $V^\dagger(s_0)$ (or generally $V^\dagger(s_h)$) can be rewritten in the form KL divergences. Instead,  Theorem 3.1 show that  any paramterized function $V$ that satisfies the condition in Theorem 3.1 is guaranteed to match $V^\dagger$, thereby allowing us to avoid explicitly computing the expectations inside $V^\dagger$. The detailed explanation is provided below:
>
>      * Firstly, for your reference, we combine and restate the proof of Proposition 1 for $V^\dagger(s_0)$ in  [Tiapkin et al, 2024] to general $V^\dagger(s_h)$ as follows.  The proof follows directly from the definition of $V^\dagger(s_h)$ for  $h \in [0,H]$ in [1]:
>
>         $V^\dagger(s_h):=E_{P_F(\tau_h|s_h)}\left[\sum_{i=h}^{H-1} \log \frac{\pi_F(s_{i+1}|s_i)}{\pi_B(s_i|s_{i+1})}+\log \pi_F(s_f|s_H)-\log R(x)\right]$
>
>         $=E_{P_F(\tau)}\left[\log \frac{\prod_{h=1}^H \pi_F(s_i|s_{i-1})P_B(s_h)Z^\ast\}{\prod_{h=1}^H \pi_B(s_i|s_{i-1})P_B(s_h)Z^\ast R(x)}\right]=D_{KL}(P_F(\tau)||P_B(\tau))-\log P_B(s_h)Z^\ast$
>
>         Then it reduces to proposition 1 for $V^\dagger(s_0)$ in [1] as  $P_B(s_0)=1$.
>
>      *  This KL expression, however does not tell us how to practically evaluate $V^\dagger(s_t)$ since it is computationally infeasible to enumerate all possible trajectories $\tau$ and explicitly compute the expectation inside $V^\dagger$. Therefore, in the RL literature, it is standard to derive various learning objectives based on the Temporal Difference (TD) formulation of the **Bellman equation** $V(s)=R(s \rightarrow s')s+E[V(s')])$ so that any parameterized function $V$ can directly approximate $V^\dagger$  without computing the full expectation (Chapter 6 in [2]). Correspondingly, we introduce the Sub-EB objective, which is grounded in the **evaluation balance equation**.
>
>     * In the RL literature[2], methods are commonly categorized into (soft) Q-learning approaches [4] and (soft) actor-critic approaches[5]. [Tiapkin et al, 2024], [3] and all flow-balancing objective (like  DB, TB,  Sub-TB)  follow the first framework, while our work and [1] follow the latter framework.
>       * The logic between the two frameworks is distinct:  In the Soft Q-learning,  $V$ and $\pi_F$ are optimized jointly (via some objective) towards the **minimal** evaluation function $V^\ast=\log P_B(s_h)Z^\ast$ and optimal policy $\pi_F^\ast$. While in the actor-critic,  we first learn a critic $V$ toward $V^\dagger$ (via some objective) that judge how far actor $\pi_F$ is from optimal actor $\pi_F^\ast$, and then minimize the policy gap.
>
>       * Tradition methods in the second direction requires a fixed $\pi_B$ and operates in an on-policy manner [2], while our Sub-EB enable **non-fixed** $\pi_B$ and **off-policy** learning. We modestly believe this represents a advancement which has not been achieved before.
>
>
> [1] Puhua Niu, Shili Wu, Mingzhou Fan, and Xiaoning Qian. Gflownet training by policy gradients. In Forty-first International Conference on Machine Learning, 2024
>
> [2] Richard S. Sutton and Andrew G. Barto. Reinforcement learning: An introduction. MIT press, 2018.
>
> [3] Tristan Deleu, Padideh Nouri, Nikolay Malkin, Doina Precup, and Yoshua Bengio. Discrete probabilistic inference as control in multi-path environments. In Uncertainty in Artificial Intelligence, pp. 997–1021. PMLR, 2024
>
> [4] Tuomas Haarnoja, Haoran Tang, Pieter Abbeel, and Sergey Levine. Reinforcement learning with deep energy-based policies. In International conference on machine learning, pp. 1352–1361. PMLR, 2017
>
> [5] Tuomas Haarnoja, Aurick Zhou, Pieter Abbeel, and Sergey Levine. Soft actor-critic: Off-policy maximum entropy deep reinforcement learning with a stochastic actor. In International conference on machine learning, pp. 1861–1870. PMLR, 2018.

---

> ### Author Response · Authors · 2025-11-19
> **Responses 1 (continued) and 2**
>
> 1. (continue)
>
>    * For your convenience, a detailed explanation of the two RL framework is provided below (and is also included in Appendix A.6 of the updated manuscript):
>
>      *  In the first framework, the core idea of Soft Q-learning [4] is to learn a function $Q$ that minimizes the mismatch of the Bellman equation ( can be reduced from  **Sub-TB and [Tiapkin et al, 2024]**), which is $E_{P_\mathcal{D}(s\rightarrow s')}[\left(\log \textcolor{red}{\pi_B(s|s')+V(s')}-Q(s, s')\right)^2]$ for the GFlowNet transition environment $\mathcal{G}$ with reward $\log \pi_B$ and $\log\pi_B(x|s_f)+V(s_f):=\log R(x)$. Here,  $\pi_F(s'|s):= \frac{\exp Q(s,s')}{\exp V(s)}$ and $V(s):=\log \sum_{s'}\exp Q(s,s')$.  Any function $Q$ that achieves zero mismatch is guaranteed to equal the optimal soft Q-function $Q^\ast$ with the corresponding $V=V^\ast$ and $\pi_F=\pi_F^\ast$.
>
>          - (i) The main contribution of [Tiapkin et al, 2024]  is that shows that if we treat $Q(s,s')$ as $\log F(s\rightarrow s')$ so that $V(s)=\log \sum_s' F(s\rightarrow s')=\log F(s)$, then the optimal solutions of the Detail Balance (DB) objective coincide exactly with the optimal solutions of the above Bellman objective. The distinction lies only in parameterization: one may parameterize $(\pi_F, V)$ directly and represent $Q(s,s')$ as $V(s)\pi_F(s'|s)$, instead of parameterizing $Q(s,s')$ and deriving $V$ and $\pi_F$ from it. As acknowledged by [Tiapkin et al, 2024], their proof only applies to the DB objective with fixed $\pi_B$.
>          - (ii) To address this limitation, [3] established an equivalence between path-consistency learning (a generalized form of soft Q-learning) and the Sub-TB objective from a gradient-based perspective.
>          - (iii) Compared to [3], **our Theorem 3.2** aims at offering a more direct and explicit connection along this RL direction.
>
>      * Our work and [1] is based on the theory of policy-gradients[6] which operates under the actor-critic framework[5]. In the framework, we learn a function $V$ that minimizes (but not necessarily) the Bellman objective ( can be reduced from  **TD-$\lambda$ or Sub-EB**), $ E_{P_F(s\rightarrow s')}[\left(\log \textcolor{red}{\pi_F(s'|s)+V(s)}-Q(s, s')\right)^2]$ with $Q(s,s'):=\log \pi_B(s|s')+V(s')$[5,7]. **At this point, the two RL directions begin to diverge.** When the $V$ achieve the optimal solution of the Bellman objective (denoted as $V^\dagger$):
>
>        $V(s)=E_{\pi_F(s'|s)}[Q(s,s')-\log \pi_F(s'|s)]$
>
>        $     =E_{\pi_F(s'|s)}\left[\log \frac{\exp Q(s,s')}{ \sum_{s'}\exp Q(s,s')    }-\log \pi_F(s'|s)+\log \sum_{s'}\exp Q(s,s')  \right  ]$
>
>        $     =- D_{KL}\left(\pi_F(s'|s)||\pi_Q(s'|s) \right) +\log \sum_{s'}\exp Q(s,s'), \quad  \pi_Q(s'|s):= \frac{\exp Q(s,s')}{\sum_{s'} \exp Q(s,s')}  $
>
>        Rewriting $\exp Q$ as $F$ yields expressions that coincide with those used in the main text. This clarifies why $V^\dagger$ (and its learned approximate $V$) serves as a critic: it evaluates how far the $\pi_F$ is from the (locally) optimal policy. The divergence is then minimized w.r.t. actor $\pi_F$ using learned critic $V$, so that true critic $V^\dagger$ moves closer to the optimal one $V^\ast$. However, traditional methods in this policy-gradient direction operate in a **on-policy** manner as indicated by the Bellman objective above and also require that $\pi_B$ is fixed.  Therefore, Sub-EB objective derived from our Theorem 3.1 is proposed specifically to overcome these limitations.
>
>
> 2. Thanks for your suggestion, the running times for the RL method with Sub-EB objective is similar to that of Sub-TB method due to their similarity. We have provide detailed running times reports for all experiments (Table.1-4 in the appendix B of the updated manuscript.)
>
> [6] Alekh Agarwal, Sham M Kakade, Jason D Lee, and Gaurav Mahajan. On the theory of policy gradient methods: Optimality, approximation, and distribution shift. Journal of Machine Learning Research, 22(98):1–76, 2021.
>
> [7] John Schulman, Xi Chen, and Pieter Abbeel. Equivalence between policy gradients and soft q-
> learning. arXiv preprint arXiv:1704.06440, 2017.

---

> ### Author Response · Authors · 2025-11-19
> **Response 3**
>
> 3. We apologize for the potentially unclear presentation.   We provide two angles of analysis below to clarify the convergence of our method based on policy gradients.
>
>      *  Firstly, in the idealized setting, convergence to the optimal policy can be established directly as follows (also provided in Appendix A.7 of the updated manuscript).  Keep setting $\pi_F(\cdot|s)$ to $\pi_Q(\cdot|s)$ for every sampled state $s$.  When all states is visited through sampling and the Bellman objective for $V$ is zero, meaning $\forall s\in S: \pi_F(\cdot|s)=\pi_Q(\cdot|s)$ and $V=V^\dagger$, we have:
>
>         * $V(x):=- D_{KL}\left(\pi_F(\cdot|x)\| \pi_{Q}(\cdot|x)\right)+ Q(x,s_f)= Q(x, s_f):=\log R(x)$  as $R(x)$ is not optimized.
>
>         * $Q(s_{H-1},x):=\log \pi_B(s_{H-1}|x)+V^\ast(x)=\log \frac{F^\ast(s_{H-1}\rightarrow x)}{F^\ast(x)}+\log R(x)=\log F^\ast(s_{H-1}\rightarrow x)$, where we use the definition of $\pi_B$. Next, we have
>
>            $V(s_{H-1}):=- D_{KL}\left(\pi_F(\cdot|s_{H-1})\|\pi_{Q}(\cdot|s_{H-1})\right)+\log\sum_x\exp  Q(s_{H-1},x)=\log\sum_x F^\ast(s_{H-1},x)=\log F^\ast(s_{H-1})$
>
>            $ \pi_F(x|s_{H-1})=\pi_{Q}(x|s_{H-1}):=\frac{\exp Q(s_{H-1},x)}{\sum_x \exp Q(s_{H-1},x)}=\frac{F^\ast(s_{H-1}\rightarrow x)}{F^\ast(s_{H-1})}:=\pi_F^\ast(x|s_{H-1}).$
>
>         *  Continuing this recursion, we will arrive at $V(s)=\log F^\ast(s)$, $Q(s\rightarrow s')=\log F^\ast(s\rightarrow s')$, and $\pi_F(s'|s)=\pi_F^\ast(s'|s)$ for all $s$ and $(s\rightarrow s')$. In practice, it is not necessary for $V$ to match $V^\dagger$ every time we update $\pi_F$. Optimality is still attainable as long as we continually set the current $\pi_F$ to ​$\pi_Q$ and reduce the Bellman objective of $V$.
>
>         Returning to our work, we generalized the ideal policy-optimization operator by the theories of policy gradients for practical implementation. Here optimizing the sub-trajectory level divergence of $P_F(\tau_{h:}|s_h)$ to $P_B(\tau_{h:}|s_h)$  implies the edge-wise divergence minimization.
>
>     *  Second, the convergence of PG can be also guaranteed in the same manner as in the basic Trajectory-Balance (TB) method [8].  We provide detail explanations for both cases as follows:
>
>         *  The TB aiming at reducing $E_{\tau\sim P_F} \left[\left(\log P_F(\tau)Z- \log P(\tau|x)R(x)\right)^2\right]$. Since both $\pi_F$ and $\pi_B$ are markovian, according to Proposition.23 in [6],  it is guaranteed that there always exists a unique $\pi_F$ such that $F(\tau):=P_F(\tau)Z$ equal to the desired $F^\ast(\tau):=P_B(\tau_x|x)R(x)$ for every $\tau$.   This means $\forall (s\rightarrow s'): F(s\rightarrow s')=F^\ast(s\rightarrow s')$. Since $\pi_F(s'|s):=F(s\rightarrow s')/F(s)$, we also have $\pi_F=\pi_F^\ast$.
>        *   The gradient of the TB objective can be written as (Appendix B.4 in  [1]):
>
>            $E_{\tau\sim P_F} \left[\sum_{h=0}^H\nabla_\theta  \log \pi_F(s_{h+1}|s_{h};\theta)\log \frac{P_F(\tau)}{P_B(\tau|x)R(x)}\right]=E_{\tau \sim P_F} \left[\sum_{h=0}^H  \nabla_\theta  \log \pi_F(s_{h+1}|s_{h};\theta)  E_{\tau_h|s_h\rightarrow s_{h+1} \sim P_F}\left[ \log \frac{P_F(\tau_h|s_h)}{P_B(\tau_h|x)R(x)}\right]\right].$  Since expectations are all practically approximated by  sample batches, the idea of policy-gradient methods (Chapter 13 of [2]) is that we can use $A^\gamma$ based on $V$ to represent the inner expectation rather than by $\sum_{k=1}^K  \log \frac{P_F(\tau_h^k|s_h^k)}{P_B(\tau_h^k|x^k)R(x^k)}$.
>        *   We further use $\gamma\in [0,1]$ to control the approximation quality. When $\gamma=1$,  the gradients of the PG and TB methods are identical, and therefore the convergence of PG is guaranteed.  When $\gamma\neq1$, gradient equivalence and convergence are guaranteed only when when $V=V^\dagger$. Nevertheless, prior works [1,9] has shown that even an imperfect $V (\neq V^\dagger)$ still substantially improve performance by appropriately tuning $\gamma$.  Since the approximation quality primarily depends on how well $V$ is learnt, our work is mainly devoted to addressing this challenge.
>
>
> [8] Yoshua Bengio, Salem Lahlou, Tristan Deleu, Edward J Hu, Mo Tiwari, and Emmanuel Bengio. Gflownet foundations. Journal of Machine Learning Research, 24(210):1–55, 2023.
>
> [9] Schulman, John, et al. "High-dimensional continuous control using generalized advantage estimation." arXiv preprint arXiv:1506.02438 (2015).

---

> ### Author Response · Authors · 2025-11-19
> **Response 4, Q1 and Q2**
>
> 4.  * We have add the results of Sub-EB-B and Sub-TB-B for 5 nodes and 10 nodes cases (Figures 13 and 2 in the updated manuscript ). We further conducted experiments and included results for the 15-node case (Figure 12  in the updated manuscript). The discussions of the 10-node and 15-node cases have been moved to the main text and are highlighted in blue.
>
>     * Thank you very much for pointing out the two closely related works.  We have added a discussion of the two works to Section 3.2 and included corresponding experimental results in our ablation study on $\pi_B$ (Both are highlighted in blue color).  For your convenience, we also provide a detailed discussion below:
>
>        * [Gritsaev et al., 2024] solves the limitation of prior RL methods [1] that required fixed $\pi_B$ by an approach, which alternates between optimizing the forward and backward policies. Applying their method to the policy-based framework, the workflow process as follows:  given a batch of trajectory samples $\\{\tau^k\\}_{k=1}^K$ from $\pi_F$, we should first freeze the parameters of $\pi_F$ and optimize $\pi_B$ by minimizing its negative log-likelihood
>
>           $-\sum_{k=1}^K \log P_B(\tau^k|x^k)$ on the sample batch. Then, we freeze $\pi_B$ and optimized $V$ and $\pi_F$ using TD-$\lambda$ and Policy-Gadient (PG), respectively.
>
>        * [Jang et al, 2024] has the similar idea. follows a similar idea, but use an additional replay buffer that stores all forward trajectories sampled in the past 20 training iterations. Due to its similarity with [Gritsaev et al., 2024] and its focus on the TB method rather than RL methods, we did not include experimental results for this method. However, we are happy to include them later in the rebuttal period.
>        *  [1] also **alternates** between optimizing the forward and backward policies: They first sample a batch of **forward** trajectories from $\pi_F$, freeze the parameters of $\pi_B$ and optimize $V$ and $\pi_F$. Next, they sample a batch of **backward** trajectories starting conditioning on the previously sampled terminal states, freeze $\pi_F$, and optimized $\pi_B$ and $W$ based these backward samples.
>        *  In contrast, our Sub-EB objective, due to its similarity to Sub-TB, allows $\pi_B$ to be freely chosen and jointly optimized with $V$, just as $\pi_B$ and state flow $F$ are jointly updated in Sub-TB.  Therefore, **no alternating / additional** optimization is required.
>
>
> Q1.  We kindly refer you to our response to Weakness 3.
>
> Q2. Sorry, we are not sure what you mean by *frequency*. Do you mean the number of times that  $V$ is updated by gradient descent for each sampled batch in an iteration?

---

### Author Response · Authors · 2025-12-04
**Rebuttal Summary**

Dear Area Chairs,

We sincerely appreciate the time and constructive feedback provided by the reviewers, as well as your follow-up comments. To best support your evaluation, we have briefly summarized our responses to the reviewers’ concerns and questions below. For ease of reference, we refer to the reviewers  in **order of appearance (from top to bottom) as Reviewers $1–4$**.

First, we note that Reviewer $2$ raised no concerns about our work, and the main concerns of Reviewer $1, 3$, and $4$ happened to all focus on the distinction between  [Tiapkin et al, 2024] (Q-much)/ Sub-TB and  our work, corresponding to Weaknesses $1-2$ for each reviewer.  Since Reviewer $3$ kindly replied and acknowledged the distinctions we summarized below, we believe our response also addresses the main concerns of Reviewers $1$ and $4$, who did not provide replies.

  * The Q-much and Sub-TB, follows the Soft **Q-learning** framework, where policy $\pi_F$ and $F/V$ are **optimized** jointly towards optimal quantities $\pi_F^\ast$ and $F^\ast/V^\ast$.

  * In contrast, our Sub-EB method follows the (Soft) Actor–Critic framework. Here, the critic $V^\dagger$, approximated by $V$, first evaluates the divergence gap between $\pi_F$ and $\pi_F^\ast$. Then under the guidance of $V^\dagger$, actor $\pi_F$ is further optimized toward $\pi_F^\ast$.

 * Prior works including Q-much give the definitions of $V^\ast$ and $V^\dagger$, whereas our contribution focuses on deriving a principled objective for learning $V^\dagger$ via function $V$, which has not been explored previously.

Second, the remaining concerns primarily relate to additional experimental results and clarification about presented method. We provide a summary table below outlining our efforts and responses. For brevity, we refer to **Reviewer as R., Weakness as W., and Question as Q.**.


| ID | Type | Responses |
| :---                        | :--- | :--- |
| **R.1**- W.3, Q.1 | Clarification | Convergence analysis in two perspectives is provided in our response comments. |
| **R.1**- W.4         | Experiment | We added the experimental results of other work about learnable backward policies (Figures 1 and 5).  |
|                              |                    |    We conducted additional experiments on 15-node BNs, and added the previously missing Sub-EB-B results for 5-, 10- and 15- nodes cases (Figures $2$, $12$, and $13$).

| ID | Type | Responses |
| :---                        | :--- | :--- |
|  **R.2** - Q.1, Q.3     | Presentation|  The corresponding unclear presentations in main text have been revised.  |
|  **R.2** - Q.2        | Clarification|  It is clarified that the suggested LED approach is orthogonal to our method and can be integrated into our framework.|

| ID | Type | Responses |
| :---                        | :--- | :--- |
|  **R.3** - W.3, W.4     | Experiment | We added the results of [Tiapkin et al. 2024] (Q-much) and Silva et al. (2024)  (CV) for Hypergrid and BN tasks (Figures 1, 5, 2, and 12). |
| **R.3**- W5 | Experiment |  The suggested  FCS metric for performance comparison on BN tasks has been added in Figures 2 and 12.|
|  **R.3** - W.6, Q.4        | Clarification| One of our main contributions is just enabling policy-based methods for offline samplers and replay buffers to explicitly enhance exploration (i.e. Sub-EB-B for all experiments).
|  **R.3** - Q.1        | Experiment | A table of reward-type counts during training is provided in the response comments to show that Sub-EB achieves more balanced learning of $V$ compared with traditional TD–$\lambda$.
|  **R.3** - Q.2        | Experiment | We conducted the gradient variance study in the suggested manner and reported the results in Figure 17.


| ID | Type | Responses |
| :---                        | :--- | :--- |
|  **R.4** - W.3        | Presentation |  The corresponding unclear presentations in main text have been revised.
|  **R.4** - W.4        | Experiment |  We have discussed other backward-policy optimization methods in Section 3.2 and added the experimental results in Figures 1 and 5.
|  **R.4** - W.5        | Clarification | Figure 7 focuses on verifying that Sub-EB-B ( with the offline sampler), improves the mode-discovery performance of Sub-EB. Since the first two tasks in Figure 7 are small-scale, observing similar performance across methods is reasonable. The performance gap between Sub-EB-B and Sub-TB-B becomes much more pronounced in the large-scale BN tasks. |
| **R.4** - W.6      | Experiment  |  The small-scale sequence design experiments are simplified from the large-scale molecular graph design experiments, which is provided in Appendix B.5, and experimental results are shown in Figures 15 and 16.|
| **R.4**-Q.1     |  Clarification | We have shown that the expectation and summation forms of Sub-TB are equivalent.

---

### Meta-Review · Area_Chair_y9ti · 2026-01-05

**Summary:**

**Summary:**

The paper introduces the Subtrajectory Evaluation Balance (Sub-EB) objective, a method for learning the evaluation function within a policy-based GFlowNet framework. By framing the problem through the lens of soft actor-critic (rather than the traditional value-based soft Q-learning perspective), the authors aim to improve training stability and flexibility.

**Strenghs and improvements during the rebuttal:**

During the review process, the primary concerns raised by reviewers focused on the scalability of the method, the absence of comparisons against recent baselines, and the precise theoretical novelty compared to existing entropy-regularized RL formulations (e.g., Tiapkin et al., 2024).

In response to reviewer feedback, the authors expanded the experimental section, including a 15-node Bayesian network structure learning and molecular graph design tasks, showing that the method scales to larger spaces. The proposed Sub-EB method also demonstrates greater stability and faster convergence compared to standard policy-based baselines and remains competitive with or stronger than state-of-the-art methods (Q-much). The rebuttal addressed the majority of the reviewers' concerns.

**Recommendation & Critical instructions for camera-ready version:**

The paper presents a stable and flexible training objective for GFlowNets. The authors have also mitigated the main concerns regarding scalability. The resulting method is empirically strong and offers a different perspective to understand the training of GFlowNets. The AC recommends acceptance based on the paper's empirical merits and the new perspective it offers on GFlowNets training.

However, while the AC acknowledges the paper's strengths and recommends acceptance, there is a remaining concern regarding the accuracy of certain claims during the discussion phase regarding He et al., 2025 and Tiapkin et al., 2024: (i) Discussions of He et al., 2025: P_F naturally holds at convergence or during training via standard normalization practices in the implementation level; it should also be contextualized within the scope where He et al., 2025 applies (specific DAG structures). The discussion should instead precisely frame the comparison as a computational scalability challenge. (ii) Comparisons with Tiapkin et al., 2024: The experimental configuration differences like a larger batch size which leads to experimental results differences should be clarified to ensuring fairness of the comparison.

The authors are advised to revise the statements/avoid inaccurate representations to ensure rigorousness and strengthen the clarity in the updated version.

**Reviewer Concerns:**

Concerns addressed by the rebuttal:

1. The reviewers initially questioned the lack of comparison against recent value-based methods (e.g., Q-much from Tiapkin et al., 2024).
- The authors included new comparisons against Q-much and CV across hypergrid and Bayesian network tasks, demonstrating that Sub-EB offers greater stability and convergence.
2. A major concern was that the experiments were limited to small-scale or simulated environments (Hypergrids, small BNs), which did not sufficiently prove the method's utility for complex problems.
- The authors added experiments on 15-node Bayesian networks and molecular graph design, showing that the method scales to larger spaces.
3. (Major; Partly addressed) Several reviewers questioned whether the proposed soft actor-critic-like approach was fundamentally different from soft Q-learning-like approach (specifically the Q-much baseline) or just a re-parameterization.
- The authors clarified the structural difference: soft Q-learning (Sub-TB/Q-much) jointly optimizes the policy and flows, while the proposed Sub-EB (soft actor-critic style) separates the evaluation from the policy update.
4. The reviewer raised doubts regarding the rigorousness of the convergence analysis.
- The authors provided convergence analysis from two perspectives in their response, strengthening the theoretical grounding.

Outstanding concerns: There remain two outstanding concerns, but can be fixed in the updated version (which is about clarity and ).
1. Framing of theoretical novelty: While the distinction between actor-critic and Q-learning is valid in implementation, the mathematical objectives in entropy-regularized RL are deeply connected.
- Please do a careful phrasing in the final version to ensure the relationship with Q-much is presented with appropriate clarification.
2. Characterization of prior work: During the rebuttal process, the authors provided responses distinguishing their work from He et al., 2025 and Tiapkin et al., 2024 (e.g., regarding the handling of backward policies and specific experimental comparisons). It was noted that some of these characterizations contained some inaccuracies. While the core contribution of the proposed method is recognized, it is important to ensure the accuracy of the statements in the updated manuscript.

**Reviewer Scores:**

- Reviewer hiRW: likely a marginal attitude (maintain 4 or increase to 6) -- most of the concerns are addressed
- Reviewer wVfJ: maintain positive (8) -- the reviewer was satisfied early on, and the rebuttal reinforced the positive impression
- Reviewer P5HE: likely a marginal attitude (increase to 4 or 6) -- the reviewer has partly acknowledged the rebuttal
- Reviewer C8bA: likely increase to 4 -- the rebuttal clarified the theoretical contribution, which was this reviewer's main concern, justifying a move away from a strong rejection.

---

### Decision · Program_Chairs · 2026-01-26

Accept (Poster)